



# Exhumation and erosion of the Northern Apennines, Italy: new insights from low-temperature thermochronometers

Erica D. Erlanger[1,2], Maria Giuditta Fellin[2], Sean D. Willett[2]

[1]GFZ Potsdam, Potsdam, 14473, Germany
[2]ETH Zürich, Zürich, 8092, Switzerland

*Correspondence to*: Erica D. Erlanger (ederlanger@gmail.com)

**Abstract.** Analysis of new detrital apatite fission-track (AFT) ages from modern river sands, published bedrock and detrital AFT ages, and bedrock apatite (U-Th)/He (AHe) ages from the Northern Apennines provide new insights into the spatial and temporal pattern of erosion rates through time across the orogen. The pattern of time-averaged erosion rates derived from AHe
ages from the Ligurian side of the orogen illustrates slower erosion rates relative to AFT rates from the Ligurian side and relative to AHe rates from the Adriatic side. These results are corroborated by an analysis of paired AFT and AHe thermochronometer samples, which illustrate that erosion rates have generally increased through time on the Adriatic side, but have decreased through time on the Ligurian side. Using an updated kinematic model of an asymmetric orogenic wedge, with imposed erosion rates on the Ligurian side that are a factor of two slower relative to the Adriatic side, we demonstrate that
cooling ages and maximum burial depths are able to replicate the pattern of measured cooling ages across the orogen and estimates of burial depth from vitrinite reflectance data. These results suggest that horizontal motion is an important component of the overall rock motion in the wedge, and that the asymmetry of the orogen has existed for at least several million years.

## 1 Introduction

The Apennine mountains of Italy are an active orogen characterized by contemporaneous extensional and compressional
tectonics. In the Northern Apennines, these features are linked to rollback of the Adriatic slab beneath Eurasia, suggested to be active since the Oligocene (Malinverno and Ryan, 1986). The interplay between extension and compression has affected the overall tectonic evolution of the Northern Apennines and, in particular, its exhumational and topographic evolution. Low-temperature bedrock and detrital thermochronology studies have constrained the timing and rates of exhumation at the orogen-scale (e.g. Thomson et al., 2010; Malusà and Balestrieri, 2012), and at the regional scale along the extensional retrowedge
(Ligurian side) of the orogen (e.g. Fellin et al., 2007) and in the frontal fold-and-thrust belt (Adriatic side) (Balestrieri et al., 1996; Carlini et al., 2013; Zattin et al., 2002). Age-elevation profiles and multiple thermochronometers have revealed spatially variable exhumation across and along strike of the orogen, and temporal variability in exhumation rates. While these findings have improved our understanding of the evolution of the Northern Apennines, a comparison between basement and detrital





data and between exhumation patterns across the primary drainage divide within the frame of an orogenic-scale analysis is
30 lacking.

In this paper, we use an updated analysis method that derives long-term erosion rates from cooling ages. We derive time-averaged erosion rates for individual samples, using two different methods for constraining the initial and final geothermal gradients. We additionally calculate erosion rates through time for existing paired AFT and AHe samples, to compare with
35 results from age-elevation transects (Thomson et al., 2010) that illustrate a change in erosion rates at 4 Ma and were interpreted to reflect an orogen-wide increase in erosion at this time. Our results suggest that the increase in exhumation is restricted to the Adriatic side of the orogen, and may have occurred later (~1–3 Ma), whereas exhumation rates decreased on the Ligurian side at ~1–5 Ma. To understand how this pattern of regional erosion rates relates to orogen-scale kinematics of the Northern Apennines, we propose an updated kinematic model that allows for crustal accretion from both frontal accretion and
40 underplating, and variable temperature at the base of the crust. We find that the pattern of AFT, AHe, and ZHe cooling ages, and the pattern of vitrinite reflectance across the orogen are broadly consistent with the wedge kinematics for an asymmetric orogen that is dominated by frontal accretion and has slower erosion rates on the Ligurian side by a factor of two.

### 1.1 Structural evolution

Development of the Apenninic wedge began at ~30 Ma, due to convergence and southwest-directed subduction of the Adriatic
microplate beneath Eurasia. From the late Oligocene, sediments supplied largely by the Alps were deposited as turbidite sequences into a series of northward-migrating foredeep basins (Macigno, Cervarola, and Marnoso-Arenacea Basins) (Fig. 1), which were eventually deformed and thrust during the Neogene (Ricci Lucchi, 1986). Until the Pliocene, these Tertiary foredeep basins were overridden by the Ligurian Unit (Fig. 1), a non-metamorphosed, allochthonous accretionary complex that was thrust upon the Tertiary foredeep deposits as a surficial nappe (Merla, 1952; Pini, 1999). Eocene-to-Pliocene basins
formed on top of the Ligurian Unit (Epi-Ligurian Unit) (Ori and Friend, 1984; Cibin et al., 2001), which record discontinuous deposition of shallow-marine and continental sediments (Ricci Lucchi, 1986), and presently exist as denudational remnants above the Ligurian Unit. Today, the Ligurian and Epi-Ligurian Units are the highest structural units exposed in the Northern Apennines.





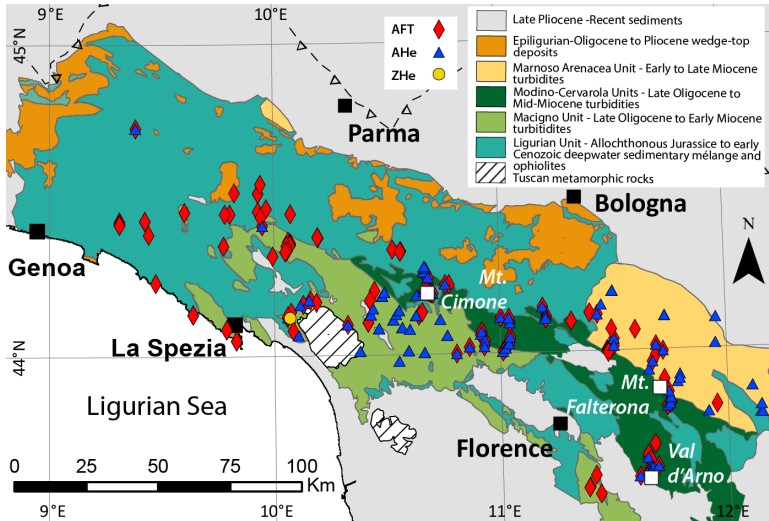

**Figure 1 Simplified geologic map of the Northern Apennines and locations of bedrock AFT samples (diamonds), AHe samples (triangles), and ZHe sample (circle). Dashed, sawtooth lines represent the thrust front buried beneath Po Plain sediments. The following chronostratigraphic divisions are used as minimum depositional ages for the Tertiary foredeep units: Macigno Unit (Chattian-Aquitanian) (Cita Sironi et al., 2006), Cervarola Unit (Aquitanian-Langhian) (Delfrati et al., 2002), and Marnoso Arenacea Unit (late Burdigalian-Tortonian) (Pialli et al., 2000).**

The onset of near-surface exhumation is constrained by the present extent and depositional ages of the Epi-Ligurian Units on the Adriatic side (Fig. 1), which are commonly not younger than the Tortonian in the NW of the study area; however, to the ESE, near Bologna, they can be as young as the Pliocene (Cibin et al., 2001). The onset of near-surface exhumation in the Northern Apennines is suggested to have begun earlier than 14 Ma, during the Tortonian (Ventura et al., 2001), although the

timing of the onset is debated. However, it is clear that rapid exhumation began at 8–9 Ma on the Ligurian side (Balestrieri et al., 1996), and at 4–7 Ma near the divide between the Ligurian and Adriatic sides, based on the ages and younging trend in AFT and AHe thermochronometers <10 Ma (Fig. 2) (Thomson et al., 2010).





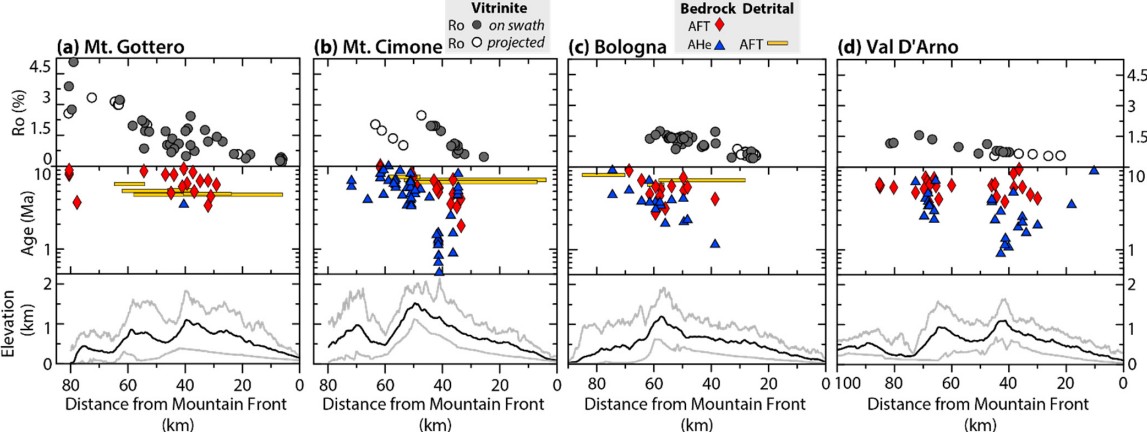

**Figure 2 Vitrinite reflectance, cooling ages, and topography plotted along (a) Mt. Gottero, (b) Mt. Cimone, (c) Bologna, and (d) Val D'Arno swath profiles. Profile locations are shown in Fig. 3. (Top Row) Filled circles are vitrinite reflectance samples located within the 30 km-wide swath profile; empty circles are located outside of swath profile line and were projected onto the line. (Middle Row) Cooling ages corrected for topography for bedrock AFT (red diamonds), and AHe (blue triangles). Detrital AFT samples (yellow rectangles) were not corrected for topography. (Bottom Row) Mean elevation (thick black line), and minimum and maximum elevation (light gray lines).**

The first evidence for emergent topography in the Northern Apennines is documented in the Early Pliocene, both by lacustrine deposits in an intermontane extensional basin located within the Magra River catchment (Fig. 3) (Bertoldi, 1988; Balestrieri et al., 2003), and by the exhumation of the Alpi Apuane metamorphic dome (white, hatched area in Fig. 3a) to the surface (Fellin et al., 2007). The onset of topographic relief then migrated eastward (Abbate et al., 1999; Carlini et al., 2013; Thomson et al., 2010), recorded by Pleistocene surface uplift of rocks at the drainage divide (Balestrieri et al., 2003), and the formation of Pleistocene-to-Holocene deformed fluvial terraces near the Adriatic mountain front (Picotti and Pazzaglia, 2008; Wegmann and Pazzaglia, 2009; Wilson et al., 2009).





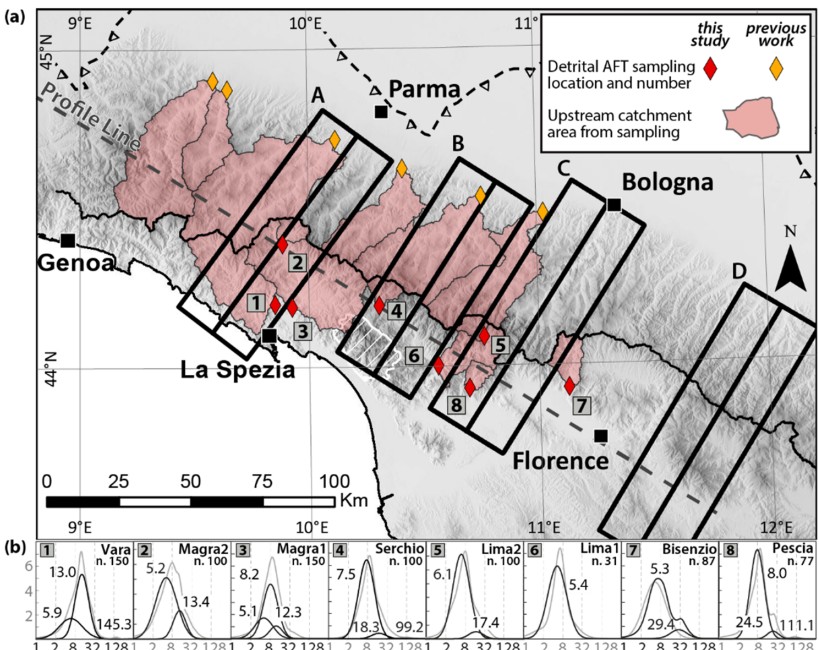

**Figure 3 (a) Location map for detrital samples. Detrital AFT samples from this study are illustrated as red diamonds, and published detrital AFT samples are shown as yellow diamonds. Numbers in the gray squares correspond to the age population plots shown in (b). (b) Peak distribution curves (black curves), total PDFs (gray curves), and peak ages for all sampled Ligurian catchments.**

### 1.2 Thermochronology data compilation

Cooling ages in the Northern Apennines are primarily limited to apatite fission track (AFT) and apatite (U-Th)/He (AHe) methods (Balestrieri et al., 1996; Ventura et al., 2001; Zattin et al., 2002; Balestrieri et al., 2003; Fellin et al., 2007; Thomson et al., 2010; Malusà and Balestrieri, 2012; Carlini et al., 2013; Balestrieri et al., 2018), as the region is dominated by sedimentary rocks (Fig. 1) that have experienced relatively low burial temperatures of less than 200–250°C (Reutter et al., 1983). Non-reset clastic rocks in the Northern Apennines are commonly exposed close to the frontal thrust zone, near the

mountain front, and at high elevations (e.g. Zattin et al., 2002; Thomson et al., 2010; Carlini et al., 2013).

Maximum burial depths of rock across the Northern Apennines are constrained most commonly from vitrinite reflectance data (Fig. 2; Reutter et al., 1983; Ventura et al., 2001; Botti et al., 2004; Carlini et al., 2013), which is a proxy for burial temperatures recorded by organic particles, and is expressed as Ro (%). Higher Ro values generally reflect higher burial temperatures; Ro

increases steadily from NE to SW in the Northern Apennines, with maximum Ro values near the Ligurian coastline, as shown along swath profiles in Fig. 2. This pattern of Ro values was interpreted to reflect NE-directed Miocene thrusting of the Ligurian Unit, which buried the underlying Tertiary foredeep deposits (Reutter et al., 1983).



## 2 Methods

### 2.1 Detrital AFT thermochronology

Bulk samples of modern sand were collected from six rivers on the Ligurian side of the Northern Apennines (Fig. 3a) and are representative of the Macigno, Alpi Apuane, and Ligurian Units (Fig. 1). As some Ligurian catchments (Magra and Serchio Rivers) contain basins with Pliocene sediments, additional samples were collected in tributaries above these basins to avoid sampling the younger, post-orogenic sediments.

Samples were processed according to the external detector method for AFT dating, using standard methods. Bulk samples were sieved and heavy minerals were separated using standard techniques, involving the use of the Wilfley table, heavy liquids, and the Frantz magnetic separator. Apatites were mounted in epoxy and were subsequently polished to expose the internal surfaces of the apatite grains. Multiple mounts per sample were produced to maximize the number of datable grains, and we aimed to date at least 100 grains per sample. However, only 37, 87, and 77 apatites were countable in samples Lima1 (6),

Bisenzio (7), and Pescia (8), respectively, whereas the high number of countable apatites in samples Vara (1) and Magra1 (3) allowed us to date 150 grains in each sample.

Apatites were etched in 5.5 N HNO$_3$ for 20 seconds at 21 °C. AFT ages were measured and calculated using the external-detector and the zeta-calibration methods (Hurford and Green, 1983) with IUGS age standards (Durango and Fish Canyon

apatites) (Hurford, 1990). The analyses were subjected to the $\chi^2$ test (Galbraith, 1981) to assess whether the sample age distributions were over-dispersed; a probability of less than 5% denotes mixed distributions.

We determined age populations for detrital samples based on dominant age peaks identified with the Binomfit program (Brandon, 2002), which is well suited for AFT data with low spontaneous track density. In order to estimate the degree of

125 resetting of the detrital age populations relative to the Apenninic orogenic event, we compared the detrital cooling ages with minimum depositional ages of the Tertiary foredeep units exposed in the drainage areas (Fig. 1).

### 2.2 Erosion rate analysis

We compiled ages from new and existing detrital AFT samples (23), bedrock AFT samples (139), AHe samples (135), and ZHe samples (26) (Tables 1–4) (Abbate et al., 1994; Balestrieri et al., 1996; Ventura et al., 2001; Zattin et al., 2002; Balestrieri

et al., 2003; Fellin et al., 2007; Thomson et al., 2010; Malusà and Balestrieri, 2012; Carlini et al., 2013). As the Alpi Apuane have an erosional history different from the rest of the Northern Apennines (Balestrieri et al., 2003; Fellin et al., 2007), we removed these samples in our compilation of thermochronometric ages.

**Table 1 Compilation of bedrock AHe cooling ages and sample descriptions.**



| ID | Method | Lithology | Latitude | Longitude | Sample Elevation (km) | Mean Elevation (km) | Age (Ma) | Error (2σ) | Surface Temperature (°C) | Reference | Measured Modern Geothermal Gradient (°C/km) | Uncertainty (°C/km) | Reference |
|---|---|---|---|---|---|---|---|---|---|---|---|---|---|
| 020620-3 | AHe | Macigno Unit | 44.122 | 10.068 | 0.756 | 0.383 | 3.66 | 0.22 | 12.19 | Fellin et al. (2007) | 40.0 | 5 | della Vedova et al. (2001) |
| 03AP34 | AHe | Macigno Unit | 44.066 | 10.107 | 0.285 | 0.340 | 6.89 | 1.22 | 12.40 | Fellin et al. (2007) | 40.0 | 7.5 | della Vedova et al. (2001) |
| 03AP47 | AHe | PseudoMacigno Unit/Apuan autoch.* | 44.128 | 10.259 | 0.890 | 0.870 | 3.60 | 0.22 | 9.75 | Fellin et al. (2007) | nd | nd | della Vedova et al. (2001) |
| 03AP51 | AHe | Macigno Unit | 44.014 | 10.380 | 1.060 | 0.688 | 6.85 | 0.41 | 10.66 | Fellin et al. (2007) | 40.0 | 2.5 | della Vedova et al. (2001) |
| 03AP58 | AHe | PseudoMacigno Unit/Apuan autoch.* | 44.003 | 10.308 | 0.305 | 0.665 | 5.86 | 0.35 | 10.77 | Fellin et al. (2007) | nd | nd | della Vedova et al. (2001) |
| 03GB04 | AHe | PseudoMacigno Unit/Apuan autoch.* | 43.974 | 10.277 | 0.600 | 0.440 | 5.45 | 0.33 | 11.90 | Fellin et al. (2007) | nd | nd | della Vedova et al. (2001) |
| 03GB07 | AHe | Macigno Unit | 44.124 | 10.059 | 0.675 | 0.356 | 5.10 | 0.31 | 12.32 | Fellin et al. (2007) | 40.0 | 7.5 | della Vedova et al. (2001) |
| 03GB09 | AHe | Macigno Unit | 44.162 | 10.115 | 0.335 | 0.546 | 3.51 | 0.21 | 11.37 | Fellin et al. (2007) | 42.5 | 7.5 | della Vedova et al. (2001) |
| 03GB10 | AHe | Macigno Unit | 44.177 | 10.156 | 0.530 | 0.656 | 6.32 | 0.38 | 10.82 | Fellin et al. (2007) | 42.5 | 7.5 | della Vedova et al. (2001) |
| 03RE19 | AHe | PseudoMacigno Unit/Apuan autoch.* | 44.009 | 10.315 | 0.270 | 0.704 | 4.04 | 0.24 | 10.58 | Fellin et al. (2007) | nd | nd | della Vedova et al. (2001) |
| 03RE20 | AHe | Macigno Unit | 44.098 | 10.326 | 1.055 | 0.808 | 4.74 | 0.28 | 10.06 | Fellin et al. (2007) | 40.0 | 5 | della Vedova et al. (2001) |
| 03AP08AB | AHe | Macigno Unit | 44.190 | 10.632 | 0.880 | 1.295 | 3.36 | 0.20 | 7.63 | Thomson et al. (2010) | 42.5 | 3.5 | della Vedova et al. (2001) |
| 03AP12A | AHe | Macigno Unit | 44.110 | 10.735 | 0.815 | 1.155 | 4.65 | 0.28 | 8.33 | Thomson et al. (2010) | 45.0 | 2.5 | della Vedova et al. (2001) |
| 03AP23A | AHe | Macigno Unit | 44.129 | 10.429 | 0.425 | 0.709 | 9.80 | 0.59 | 10.56 | Thomson et al. (2010) | 40.0 | 2.5 | della Vedova et al. (2001) |
| 03AP23B | AHe | Macigno Unit | 44.129 | 10.429 | 0.425 | 0.709 | 6.27 | 0.38 | 10.56 | Thomson et al. (2010) | 40.0 | 2.5 | della Vedova et al. (2001) |
| 03AP28A | AHe | Macigno Unit | 44.111 | 10.529 | 1.035 | 0.899 | 6.60 | 0.40 | 9.60 | Thomson et al. (2010) | 40.0 | 2.5 | della Vedova et al. (2001) |
| 03AP28C | AHe | Macigno Unit | 44.111 | 10.529 | 1.035 | 0.899 | 7.66 | 0.46 | 9.60 | Thomson et al. (2010) | 40.0 | 2.5 | della Vedova et al. (2001) |
| 03AP28D | AHe | Macigno Unit | 44.111 | 10.529 | 1.035 | 0.899 | 6.12 | 0.37 | 9.60 | Thomson et al. (2010) | 40.0 | 2.5 | della Vedova et al. (2001) |
| 03AP29A | AHe | Macigno Unit | 44.130 | 10.542 | 1.320 | 1.050 | 5.33 | 0.32 | 8.85 | Thomson et al. (2010) | 40.0 | 2.5 | della Vedova et al. (2001) |
| 03AP31A | AHe | Macigno Unit | 44.142 | 10.553 | 1.815 | 1.131 | 7.40 | 0.44 | 8.44 | Thomson et al. (2010) | 40.0 | 2.5 | della Vedova et al. (2001) |
| 03AP31B | AHe | Macigno Unit | 44.142 | 10.553 | 1.815 | 1.131 | 5.22 | 0.31 | 8.44 | Thomson et al. (2010) | 42.5 | 5 | della Vedova et al. (2001) |
| 03AP51C | AHe | Macigno Unit | 44.014 | 10.380 | 1.060 | 0.688 | 7.92 | 0.48 | 10.66 | Thomson et al. (2010) | 40.0 | 2.5 | della Vedova et al. (2001) |
| 03AP52A | AHe | Macigno Unit | 44.084 | 10.463 | 0.370 | 0.582 | 7.04 | 0.42 | 11.19 | Thomson et al. (2010) | 40.0 | 7.5 | della Vedova et al. (2001) |
| 03AP52B | AHe | Macigno Unit | 44.084 | 10.463 | 0.370 | 0.582 | 7.00 | 0.42 | 11.19 | Thomson et al. (2010) | 40.0 | 7.5 | della Vedova et al. (2001) |
| 03AP52C | AHe | Macigno Unit | 44.084 | 10.463 | 0.370 | 0.582 | 8.01 | 0.48 | 11.19 | Thomson et al. (2010) | 40.0 | 7.5 | della Vedova et al. (2001) |
| 03RE02 | AHe | Macigno Unit | 44.148 | 10.438 | 0.765 | 0.836 | 6.85 | 0.41 | 9.92 | Thomson et al. (2010) | 40.0 | 5 | della Vedova et al. (2001) |
| 03RE05A | AHe | Macigno Unit | 44.188 | 10.480 | 1.495 | 1.138 | 6.73 | 0.40 | 8.41 | Thomson et al. (2010) | 40.0 | 2.5 | della Vedova et al. (2001) |
| 03RE05B | AHe | Macigno Unit | 44.188 | 10.480 | 1.495 | 1.138 | 5.71 | 0.34 | 8.41 | Thomson et al. (2010) | 40.0 | 2.5 | della Vedova et al. (2001) |
| 03RE05C | AHe | Macigno Unit | 44.188 | 10.480 | 1.495 | 0.582 | 8.10 | 0.49 | 11.19 | Thomson et al. (2010) | 42.5 | 7.5 | della Vedova et al. (2001) |
| 03RE05CD | AHe | Macigno Unit | 44.188 | 10.480 | 1.495 | 1.138 | 9.41 | 0.56 | 8.41 | Thomson et al. (2010) | 40.0 | 2.5 | della Vedova et al. (2001) |
| 03RE05D | AHe | Macigno Unit | 44.188 | 10.480 | 1.495 | 0.582 | 6.37 | 0.38 | 11.19 | Thomson et al. (2010) | 42.5 | 7.5 | della Vedova et al. (2001) |
| 03RE06A | AHe | Macigno Unit | 44.201 | 10.488 | 1.640 | 1.194 | 5.93 | 0.36 | 8.13 | Thomson et al. (2010) | 42.5 | 7.5 | della Vedova et al. (2001) |
| 03RE06B | AHe | Macigno Unit | 44.201 | 10.488 | 1.640 | 1.194 | 6.43 | 0.39 | 8.13 | Thomson et al. (2010) | 42.5 | 7.5 | della Vedova et al. (2001) |
| 03RE12A | AHe | Macigno Unit | 44.059 | 10.767 | 0.460 | 0.942 | 5.87 | 0.35 | 9.39 | Thomson et al. (2010) | 45.0 | 7.5 | della Vedova et al. (2001) |
| 03RE12B | AHe | Macigno Unit | 44.059 | 10.767 | 0.460 | 0.942 | 5.55 | 0.33 | 9.39 | Thomson et al. (2010) | 45.0 | 7.5 | della Vedova et al. (2001) |
| 03RE14A | AHe | Macigno Unit | 44.005 | 10.665 | 0.840 | 0.635 | 5.28 | 0.32 | 10.92 | Thomson et al. (2010) | 40.0 | 5 | della Vedova et al. (2001) |
| 03RE14B | AHe | Macigno Unit | 44.005 | 10.665 | 0.840 | 0.635 | 9.95 | 0.60 | 10.92 | Thomson et al. (2010) | 40.0 | 5 | della Vedova et al. (2001) |
| 03RE7 | AHe | Macigno Unit | 44.200 | 10.676 | 1.600 | 1.214 | 2.88 | 0.17 | 8.03 | Thomson et al. (2010) | 45.0 | 1.5 | della Vedova et al. (2001) |
| 03RE7R1 | AHe | Macigno Unit | 44.200 | 10.676 | 1.600 | 1.214 | 3.14 | 0.19 | 8.03 | Thomson et al. (2010) | 45.0 | 1.5 | della Vedova et al. (2001) |
| 03TH02 | AHe | Macigno Unit | 44.086 | 10.568 | 0.979 | 0.881 | 6.70 | 0.40 | 9.70 | Thomson et al. (2010) | 42.5 | 5 | della Vedova et al. (2001) |
| 03TH02B | AHe | Macigno Unit | 44.086 | 10.568 | 0.979 | 0.881 | 7.51 | 0.45 | 9.70 | Thomson et al. (2010) | 42.5 | 5 | della Vedova et al. (2001) |
| 03TH12B | AHe | Macigno Unit | 44.080 | 10.600 | 0.678 | 0.904 | 4.27 | 0.26 | 9.58 | Thomson et al. (2010) | 40.0 | 5 | della Vedova et al. (2001) |
| 03TH13A | AHe | Macigno Unit | 44.013 | 10.593 | 0.153 | 0.578 | 8.18 | 0.49 | 11.21 | Thomson et al. (2010) | 40.0 | 5 | della Vedova et al. (2001) |
| 03TH13C | AHe | Macigno Unit | 44.013 | 10.593 | 0.153 | 0.578 | 7.27 | 0.44 | 11.21 | Thomson et al. (2010) | 40.0 | 5 | della Vedova et al. (2001) |
| 03TH18A | AHe | Macigno Unit | 43.980 | 10.552 | 0.047 | 0.449 | 6.74 | 0.40 | 11.85 | Thomson et al. (2010) | 40.0 | 5 | della Vedova et al. (2001) |
| 03TH23A | AHe | Macigno Unit | 44.124 | 10.628 | 1.645 | 1.205 | 4.73 | 0.28 | 8.08 | Thomson et al. (2010) | 42.5 | 2.5 | della Vedova et al. (2001) |
| 03TH23BD | AHe | Macigno Unit | 44.124 | 10.628 | 1.645 | 1.205 | 5.38 | 0.32 | 8.08 | Thomson et al. (2010) | 42.5 | 2.5 | della Vedova et al. (2001) |
| 03TH23C | AHe | Macigno Unit | 44.124 | 10.628 | 1.645 | 1.205 | 4.55 | 0.27 | 8.08 | Thomson et al. (2010) | 42.5 | 2.5 | della Vedova et al. (2001) |
| 050320-1C | AHe | Helminthoid Flysch | 44.263 | 10.664 | 1.112 | 1.024 | 1.15 | 0.11 | 8.98 | Thomson et al. (2010) | 42.5 | 3 | della Vedova et al. (2001) |
| 050320-1D | AHe | Helminthoid Flysch | 44.263 | 10.664 | 1.112 | 1.024 | 1.84 | 0.11 | 8.98 | Thomson et al. (2010) | 42.5 | 3.5 | della Vedova et al. (2001) |
| 050320-2B | AHe | Helminthoid Flysch | 44.276 | 10.674 | 1.239 | 0.965 | 5.28 | 0.32 | 9.27 | Thomson et al. (2010) | 42.5 | 4 | della Vedova et al. (2001) |
| 050320-2C | AHe | Helminthoid Flysch | 44.276 | 10.674 | 1.239 | 0.965 | 5.42 | 0.33 | 9.27 | Thomson et al. (2010) | 42.5 | 4.5 | della Vedova et al. (2001) |
| 050320-3A | AHe | Helminthoid Flysch | 44.280 | 10.668 | 1.272 | 0.957 | 9.29 | 0.56 | 9.31 | Thomson et al. (2010) | 42.5 | 5 | della Vedova et al. (2001) |
| 050320-3B | AHe | Helminthoid Flysch | 44.280 | 10.668 | 1.272 | 0.957 | 6.04 | 0.66 | 9.31 | Thomson et al. (2010) | 42.5 | 5.5 | della Vedova et al. (2001) |
| 050320-3C | AHe | Helminthoid Flysch | 44.280 | 10.668 | 1.272 | 0.957 | 6.61 | 0.40 | 9.31 | Thomson et al. (2010) | 42.5 | 6 | della Vedova et al. (2001) |





| 1926 | AHe | Marnoso Arenacea Unit | 44.107 | 11.729 | 0.250 | 0.471 | 6.14 | 0.37 | 11.75 | Thomson et al. (2010) | 25.5 | 0.5 | Pauselli et al. (2019) |
|---|---|---|---|---|---|---|---|---|---|---|---|---|---|
| 1926B | AHe | Marnoso Arenacea Unit | 44.107 | 11.729 | 0.250 | 0.471 | 3.16 | 0.19 | 11.75 | Thomson et al. (2010) | 25.5 | 0.5 | Pauselli et al. (2019) |
| 1926C | AHe | Marnoso Arenacea Unit | 44.107 | 11.729 | 0.250 | 0.471 | 5.88 | 0.35 | 11.75 | Thomson et al. (2010) | 25.5 | 0.5 | Pauselli et al. (2019) |
| 1926D | AHe | Marnoso Arenacea Unit | 44.107 | 11.729 | 0.250 | 0.471 | 2.89 | 0.17 | 11.75 | Thomson et al. (2010) | 25.5 | 0.5 | Pauselli et al. (2019) |
| 1929 | AHe | Marnoso Arenacea Unit | 44.037 | 11.504 | 0.700 | 0.702 | 1.65 | 0.10 | 10.59 | Thomson et al. (2010) | 28.5 | 2.5 | Pauselli et al. (2019) |
| AP1 | AHe | Marnoso Arenacea Unit | 43.790 | 12.146 | 0.700 | 0.776 | 1.34 | 0.08 | 10.22 | Thomson et al. (2010) | 23.5 | 3 | Pauselli et al. (2019) |
| AP17 | AHe | Marnoso Arenacea Unit | 43.876 | 12.110 | 0.600 | 0.605 | 1.94 | 0.12 | 11.07 | Thomson et al. (2010) | 22.5 | 2.5 | Pauselli et al. (2019) |
| AP2 | AHe | Marnoso Arenacea Unit | 43.79 | 12.15 | 0.60 | 0.77 | 2.41 | 0.14 | 10.25 | Thomson et al. (2010) | 24.0 | 1.5 | Pauselli et al. (2019) |
| AP3 | AHe | Marnoso Arenacea Unit | 43.815 | 12.149 | 0.900 | 0.727 | 3.27 | 0.20 | 10.46 | Thomson et al. (2010) | 23.5 | 1.5 | Pauselli et al. (2019) |
| AP30 | AHe | Marnoso Arenacea Unit | 43.895 | 11.779 | 0.750 | 0.879 | 1.62 | 0.10 | 9.70 | Thomson et al. (2010) | 27.5 | 1.5 | Pauselli et al. (2019) |
| AP33 | AHe | Marnoso Arenacea Unit | 43.919 | 11.792 | 0.650 | 0.801 | 1.29 | 0.08 | 10.09 | Thomson et al. (2010) | 27.0 | 1.5 | Pauselli et al. (2019) |
| AP36E | AHe | Marnoso Arenacea Unit | 44.097 | 11.955 | 0.370 | 0.271 | 9.54 | 0.77 | 12.74 | Thomson et al. (2010) | 22.5 | 0.5 | Pauselli et al. (2019) |
| AP37 | AHe | Marnoso Arenacea Unit | 44.015 | 11.951 | 0.150 | 0.423 | 2.95 | 0.18 | 11.99 | Thomson et al. (2010) | 21.0 | 1.5 | Pauselli et al. (2019) |
| AP38 | AHe | Marnoso Arenacea Unit | 43.797 | 11.914 | 1.200 | 0.858 | 1.97 | 0.12 | 9.81 | Thomson et al. (2010) | 26.0 | 6.5 | Pauselli et al. (2019) |
| AP43R1 | AHe | Marnoso Arenacea Unit | 43.818 | 11.733 | 0.515 | 0.860 | 3.08 | 0.20 | 9.80 | Thomson et al. (2010) | 29.5 | 2.5 | Pauselli et al. (2019) |
| AP43R2 | AHe | Marnoso Arenacea Unit | 43.818 | 11.733 | 0.515 | 0.860 | 3.35 | 0.20 | 9.80 | Thomson et al. (2010) | 29.5 | 2.5 | Pauselli et al. (2019) |
| AP44R1 | AHe | Marnoso Arenacea Unit | 43.824 | 11.746 | 0.725 | 0.868 | 2.02 | 0.12 | 9.76 | Thomson et al. (2010) | 29.5 | 2.5 | Pauselli et al. (2019) |
| AP45R1 | AHe | Marnoso Arenacea Unit | 43.844 | 11.749 | 0.940 | 0.897 | 3.12 | 0.22 | 9.62 | Thomson et al. (2010) | 29.5 | 2.5 | Pauselli et al. (2019) |
| AP45R2 | AHe | Marnoso Arenacea Unit | 43.844 | 11.749 | 0.940 | 0.897 | 1.04 | 0.06 | 9.62 | Thomson et al. (2010) | 29.0 | 3.5 | Pauselli et al. (2019) |
| AP47R1 | AHe | Marnoso Arenacea Unit | 43.864 | 11.739 | 1.365 | 0.906 | 2.33 | 0.14 | 9.57 | Thomson et al. (2010) | 29.0 | 3.5 | Pauselli et al. (2019) |
| AP48R1 | AHe | Marnoso Arenacea Unit | 43.879 | 11.711 | 1.655 | 0.907 | 3.29 | 0.20 | 9.56 | Thomson et al. (2010) | 30.0 | 1.25 | Pauselli et al. (2019) |
| AP48R2 | AHe | Marnoso Arenacea Unit | 43.879 | 11.711 | 1.655 | 0.907 | 3.29 | 0.20 | 9.56 | Thomson et al. (2010) | 30.0 | 1.25 | Pauselli et al. (2019) |
| AP5 | AHe | Marnoso Arenacea Unit | 44.189 | 11.501 | 0.200 | 0.521 | 5.96 | 0.36 | 11.50 | Thomson et al. (2010) | 28.5 | 1.5 | della Vedova et al. (2001) |
| AP52 | AHe | Marnoso Arenacea Unit | 43.905 | 11.791 | 0.565 | 0.836 | 1.92 | 0.12 | 9.92 | Thomson et al. (2010) | 26.5 | 1.5 | Pauselli et al. (2019) |
| AP53 | AHe | Marnoso Arenacea Unit | 43.934 | 11.656 | 0.907 | 0.830 | 5.33 | 0.32 | 9.95 | Thomson et al. (2010) | 29.0 | 1 | Pauselli et al. (2019) |
| AP54 | AHe | Marnoso Arenacea Unit | 43.961 | 11.670 | 0.690 | 0.811 | 1.93 | 0.12 | 10.05 | Thomson et al. (2010) | 27.5 | 1.5 | Pauselli et al. (2019) |
| AP55 | AHe | Marnoso Arenacea Unit | 44.013 | 11.687 | 1.070 | 0.722 | 2.08 | 0.12 | 10.11 | Thomson et al. (2010) | 27.5 | 1.5 | Pauselli et al. (2019) |
| AP57 | AHe | Marnoso Arenacea Unit | 43.995 | 11.719 | 0.450 | 0.746 | 1.32 | 0.08 | 10.37 | Thomson et al. (2010) | 27.5 | 0 | Pauselli et al. (2019) |
| AP5B | AHe | Marnoso Arenacea Unit | 44.189 | 11.501 | 0.200 | 0.521 | 2.15 | 0.13 | 11.50 | Thomson et al. (2010) | 28.5 | 1.5 | della Vedova et al. (2001) |
| AP5C | AHe | Marnoso Arenacea Unit | 44.189 | 11.501 | 0.200 | 0.521 | 1.01 | 0.06 | 11.50 | Thomson et al. (2010) | 28.5 | 1.5 | della Vedova et al. (2001) |
| AP5D | AHe | Marnoso Arenacea Unit | 44.189 | 11.501 | 0.200 | 0.521 | 2.72 | 0.16 | 11.50 | Thomson et al. (2010) | 28.5 | 2 | della Vedova et al. (2001) |
| AP8 | AHe | Marnoso Arenacea Unit | 44.147 | 11.449 | 0.300 | 0.629 | 1.28 | 0.08 | 10.95 | Thomson et al. (2010) | 30.0 | 2.5 | della Vedova et al. (2001) |
| AP9 | AHe | Marnoso Arenacea Unit | 44.115 | 11.431 | 0.400 | 0.674 | 1.37 | 0.08 | 10.73 | Thomson et al. (2010) | 27.5 | 3.75 | Pauselli et al. (2019) |
| C1 | AHe | Cervarola Unit | 44.113 | 11.002 | 0.500 | 0.765 | 3.62 | 0.22 | 10.28 | Thomson et al. (2010) | 40.0 | 6 | della Vedova et al. (2001) |
| C10 | AHe | Cervarola Unit | 44.143 | 11.191 | 0.605 | 0.766 | 0.79 | 0.05 | 10.27 | Thomson et al. (2010) | 35.0 | 5 | della Vedova et al. (2001) |
| C11 | AHe | Cervarola Unit | 44.001 | 10.807 | 0.950 | 0.667 | 6.11 | 0.37 | 10.77 | Thomson et al. (2010) | 45.0 | 7.5 | della Vedova et al. (2001) |
| C13 | AHe | Cervarola Unit | 44.021 | 10.864 | 0.700 | 0.747 | 3.87 | 0.23 | 10.37 | Thomson et al. (2010) | 47.5 | 10 | della Vedova et al. (2001) |
| C16 | AHe | Cervarola Unit | 44.060 | 10.913 | 0.630 | 0.909 | 2.86 | 0.17 | 9.55 | Thomson et al. (2010) | 45.0 | 8.5 | della Vedova et al. (2001) |
| C17 | AHe | Cervarola Unit | 44.068 | 10.919 | 0.625 | 0.921 | 2.80 | 0.17 | 9.49 | Thomson et al. (2010) | 45.0 | 8.5 | della Vedova et al. (2001) |
| C2 | AHe | Cervarola Unit | 44.004 | 11.012 | 0.830 | 0.660 | 3.62 | 0.22 | 10.80 | Thomson et al. (2010) | 45.0 | 8.5 | della Vedova et al. (2001) |
| C22 | AHe | Cervarola Unit | 44.041 | 10.932 | 0.850 | 0.821 | 3.97 | 0.24 | 10.00 | Thomson et al. (2010) | 45.0 | 8.5 | della Vedova et al. (2001) |
| C23 | AHe | Cervarola Unit | 44.021 | 10.929 | 0.884 | 0.719 | 4.27 | 0.26 | 10.51 | Thomson et al. (2010) | 45.0 | 7.5 | della Vedova et al. (2001) |
| C29 | AHe | Cervarola Unit | 44.731 | 9.386 | 0.320 | 0.775 | 2.84 | 0.17 | 10.23 | Thomson et al. (2010) | 35.0 | 0.5 | della Vedova et al. (2001) |
| C3 | AHe | Cervarola Unit | 44.014 | 11.025 | 0.780 | 0.706 | 4.05 | 0.24 | 10.57 | Thomson et al. (2010) | 45.0 | 8 | della Vedova et al. (2001) |
| C34 | AHe | Cervarola Unit | 44.417 | 9.949 | 0.510 | 0.855 | 2.73 | 0.16 | 9.83 | Thomson et al. (2010) | 40.0 | 2.5 | della Vedova et al. (2001) |
| C37 | AHe | Cervarola Unit | 44.246 | 10.683 | 0.641 | 1.064 | 1.59 | 0.10 | 8.78 | Thomson et al. (2010) | 45.0 | 2.5 | della Vedova et al. (2001) |
| C4 | AHe | Cervarola Unit | 44.028 | 11.038 | 0.680 | 0.753 | 1.96 | 0.12 | 10.34 | Thomson et al. (2010) | 43.5 | 8 | della Vedova et al. (2001) |
| C40 | AHe | Cervarola Unit | 44.223 | 10.758 | 1.250 | 0.980 | 1.73 | 0.10 | 9.20 | Thomson et al. (2010) | 46.0 | 1 | della Vedova et al. (2001) |
| C5 | AHe | Cervarola Unit | 44.049 | 11.044 | 0.610 | 0.798 | 3.64 | 0.22 | 10.11 | Thomson et al. (2010) | 42.5 | 7.5 | della Vedova et al. (2001) |
| C52A | AHe | Cervarola Unit | 44.013 | 11.503 | 0.360 | 0.661 | 3.43 | 0.21 | 10.80 | Thomson et al. (2010) | 28.5 | 4 | Pauselli et al. (2019) |
| C6 | AHe | Cervarola Unit | 44.095 | 11.044 | 0.650 | 0.779 | 1.92 | 0.12 | 10.21 | Thomson et al. (2010) | 40.0 | 2.5 | della Vedova et al. (2001) |
| C7 | AHe | Cervarola Unit | 44.111 | 11.039 | 0.625 | 0.755 | 2.05 | 0.12 | 10.33 | Thomson et al. (2010) | 40.0 | 2.5 | della Vedova et al. (2001) |
| C8 | AHe | Cervarola Unit | 44.115 | 11.204 | 0.700 | 0.757 | 1.89 | 0.11 | 10.32 | Thomson et al. (2010) | 37.5 | 2.5 | della Vedova et al. (2001) |
| C9 | AHe | Cervarola Unit | 44.106 | 11.204 | 1.000 | 0.738 | 1.62 | 0.10 | 10.41 | Thomson et al. (2010) | 37.5 | 2.5 | della Vedova et al. (2001) |
| CIM1 | AHe | Cervarola Unit (Modino) | 44.194 | 10.699 | 2.165 | 1.174 | 3.35 | 0.20 | 8.23 | Thomson et al. (2010) | 45.0 | 1.5 | della Vedova et al. (2001) |
| CIM1R1 | AHe | Cervarola Unit (Modino) | 44.194 | 10.699 | 2.165 | 1.174 | 3.34 | 0.20 | 8.23 | Thomson et al. (2010) | 45.0 | 1.5 | della Vedova et al. (2001) |
| CIM2 | AHe | Cervarola Unit (Modino) | 44.194 | 10.704 | 2.045 | 1.165 | 2.74 | 0.16 | 8.27 | Thomson et al. (2010) | 45.0 | 1.5 | della Vedova et al. (2001) |
| CIM3 | AHe | Cervarola Unit (Modino) | 44.196 | 10.692 | 1.950 | 1.184 | 2.63 | 0.16 | 8.18 | Thomson et al. (2010) | 45.0 | 1.5 | della Vedova et al. (2001) |
| CIM3R1 | AHe | Cervarola Unit (Modino) | 44.196 | 10.692 | 1.950 | 1.184 | 3.55 | 0.21 | 8.18 | Thomson et al. (2010) | 45.0 | 1.5 | della Vedova et al. (2001) |



| | | | | | | | | | | | |
|---|---|---|---|---|---|---|---|---|---|---|---|
| CIM4 | AHe | Cervarola Unit (Modino) | 44.200 | 10.684 | 1.830 | 1.196 | 2.84 | 0.17 | 8.12 | Thomson et al. (2010) | 45.0 | 1.5 | della Vedova et al. (2001) |
| CIM4R1 | AHe | Cervarola Unit (Modino) | 44.200 | 10.684 | 1.830 | 1.196 | 2.98 | 0.18 | 8.12 | Thomson et al. (2010) | 45.0 | 1.5 | della Vedova et al. (2001) |
| CIM5 | AHe | Cervarola Unit (Modino) | 44.202 | 10.677 | 1.750 | 1.208 | 2.98 | 0.18 | 8.06 | Thomson et al. (2010) | 45.0 | 1.5 | della Vedova et al. (2001) |
| CIM5A | AHe | Cervarola Unit (Modino) | 44.202 | 10.677 | 1.750 | 1.208 | 2.69 | 0.16 | 8.06 | Thomson et al. (2010) | 45.0 | 1.5 | della Vedova et al. (2001) |
| CIM5R1 | AHe | Cervarola Unit (Modino) | 44.202 | 10.677 | 1.750 | 1.208 | 2.53 | 0.15 | 8.06 | Thomson et al. (2010) | 45.0 | 1.5 | della Vedova et al. (2001) |
| CIM6 | AHe | Cervarola Unit (Modino) | 44.201 | 10.666 | 1.660 | 1.230 | 2.62 | 0.16 | 7.95 | Thomson et al. (2010) | 45.0 | 1.5 | della Vedova et al. (2001) |
| CIM6R1 | AHe | Cervarola Unit (Modino) | 44.201 | 10.666 | 1.660 | 1.230 | 2.68 | 0.16 | 7.95 | Thomson et al. (2010) | 45.0 | 1.5 | della Vedova et al. (2001) |
| VALD10a | AHe | Macigno Unit | 43.653 | 11.640 | 1.4 | 0.836 | 3.49 | 0.21 | 9.92 | Thomson et al. (2010) | 28.5 | 1 | Pauselli et al. (2019) |
| VALD1a | AHe | Macigno Unit | 43.594 | 11.603 | 0.497 | 0.471 | 6.94 | 0.42 | 11.75 | Thomson et al. (2010) | 29.0 | 1 | Pauselli et al. (2019) |
| VALD2a | AHe | Macigno Unit | 43.612 | 11.645 | 0.5 | 0.685 | 3.89 | 0.23 | 10.67 | Thomson et al. (2010) | 30.0 | 0.5 | Pauselli et al. (2019) |
| VALD2R1 | AHe | Macigno Unit | 43.612 | 11.645 | 0.5 | 0.685 | 3.87 | 0.23 | 10.67 | Thomson et al. (2010) | 30.0 | 0.5 | Pauselli et al. (2019) |
| VALD4a1 | AHe | Macigno Unit | 43.620 | 11.648 | 0.74 | 0.730 | 3.52 | 0.21 | 10.45 | Thomson et al. (2010) | 29.5 | 0.5 | Pauselli et al. (2019) |
| VALD4a2 | AHe | Macigno Unit | 43.620 | 11.648 | 0.74 | 0.730 | 5.03 | 0.3 | 10.45 | Thomson et al. (2010) | 29.5 | 0.5 | Pauselli et al. (2019) |
| VALD4R1 | AHe | Macigno Unit | 43.620 | 11.648 | 0.74 | 0.730 | 4.01 | 0.24 | 10.45 | Thomson et al. (2010) | 29.5 | 0.5 | Pauselli et al. (2019) |
| VALD5a | AHe | Macigno Unit | 43.621 | 11.656 | 0.85 | 0.747 | 3.96 | 0.24 | 10.36 | Thomson et al. (2010) | 30.0 | 0.5 | Pauselli et al. (2019) |
| VALD5R1 | AHe | Macigno Unit | 43.621 | 11.656 | 0.85 | 0.747 | 3.92 | 0.23 | 10.36 | Thomson et al. (2010) | 30.0 | 0.5 | Pauselli et al. (2019) |
| VALD6a | AHe | Macigno Unit | 43.620 | 11.659 | 0.88 | 0.746 | 4.09 | 0.25 | 10.37 | Thomson et al. (2010) | 30.0 | 0.5 | Pauselli et al. (2019) |
| VALD6R1 | AHe | Macigno Unit | 43.620 | 11.659 | 0.88 | 0.746 | 3.86 | 0.23 | 10.37 | Thomson et al. (2010) | 30.0 | 0.5 | Pauselli et al. (2019) |
| VALD7a | AHe | Macigno Unit | 43.604 | 11.651 | 1.1 | 0.658 | 3.75 | 0.22 | 10.81 | Thomson et al. (2010) | 31.0 | 0.5 | Pauselli et al. (2019) |
| VALD8R1 | AHe | Macigno Unit | 43.626 | 11.684 | 1.2 | 0.782 | 4.11 | 0.25 | 10.19 | Thomson et al. (2010) | 30.5 | 0.5 | Pauselli et al. (2019) |
| VALD8R2 | AHe | Macigno Unit | 43.626 | 11.684 | 1.2 | 0.782 | 3.46 | 0.21 | 10.19 | Thomson et al. (2010) | 30.5 | 0.5 | Pauselli et al. (2019) |

*Lithologies exposed in the Alpi Apuane metamorphic dome. These samples were excluded from the erosion rate inversions

**Table 2 Compilation of bedrock AFT cooling ages and sample descriptions.**



| ID | Method | Lithology | Latitude | Longitude | Sample Elevation (km) | Mean Elevation (km) | Age (Ma) | Error (1σ) | Surface Temperature (°C) | Reference |
|---|---|---|---|---|---|---|---|---|---|---|
| AR1 | AFT | Pseudomacigno Apuan autochthon* | 44.058 | 10.226 | 0.840 | 0.607 | 4.71 | 0.59 | 11.06 | Abbate et al. (1994) |
| (AR2A)a | AFT | Pseudomacigno Apuan autochthon* | 44.055 | 10.256 | 0.840 | 0.634 | 5.24 | 0.63 | 10.93 | Abbate et al. (1994) |
| AR3 | AFT | Pseudomacigno Apuan autochthon* | 44.055 | 10.256 | 0.840 | 0.634 | 4.95 | 0.58 | 10.93 | Abbate et al. (1994) |
| BT1 | AFT | graywacke | 44.085 | 9.787 | 0.475 | 0.104 | 4.58 | 0.78 | 13.58 | Abbate et al. (1994) |
| BT2 | AFT | graywacke | 44.085 | 9.787 | 0.525 | 0.104 | 4.73 | 0.55 | 13.58 | Abbate et al. (1994) |
| CB3 | AFT | granite cgl | 44.051 | 9.830 | 0.000 | 0.079 | 8.11 | 1.19 | 13.70 | Abbate et al. (1994) |
| CB4 | AFT | gneiss cgl | 44.051 | 9.830 | 0.000 | 0.079 | 8.46 | 0.42 | 13.70 | Abbate et al. (1994) |
| CB5 | AFT | graywacke | 44.051 | 9.830 | 0.000 | 0.079 | 7.13 | 0.67 | 13.70 | Abbate et al. (1994) |
| CP1 | AFT | Hercynian Basement* | 44.028 | 10.264 | 0.675 | 0.554 | 3.93 | 0.36 | 11.33 | Abbate et al. (1994) |
| CP3 | AFT | Apuan autochthon* | 44.017 | 10.264 | 0.650 | 0.554 | 3.64 | 0.71 | 11.33 | Abbate et al. (1994) |
| FC3 | AFT | Pseudomacigno Apuan autochthon* | 44.078 | 10.267 | 1.620 | 0.701 | 5.59 | 0.61 | 10.59 | Abbate et al. (1994) |
| FC5 | AFT | Pseudomacigno Apuan autochthon* | 44.078 | 10.267 | 1.620 | 0.701 | 5.95 | 0.59 | 10.59 | Abbate et al. (1994) |
| FO1 | AFT | Pseudomacigno Apuan autochthon* | 44.036 | 10.375 | 0.450 | 0.619 | 1.96 | 0.56 | 11.00 | Abbate et al. (1994) |
| FO4 | AFT | Pseudomacigno Apuan autochthon* | 44.033 | 10.375 | 0.425 | 0.610 | 1.91 | 0.31 | 11.05 | Abbate et al. (1994) |
| FO5 | AFT | Pseudomacigno Apuan autochthon* | 44.044 | 10.388 | 0.460 | 0.636 | 1.63 | 0.25 | 10.92 | Abbate et al. (1994) |
| G2 | AFT | Hercynian Basement Apuan autochthon* | 44.069 | 10.193 | 0.170 | 0.571 | 3.96 | 0.36 | 11.24 | Abbate et al. (1994) |
| MD1 (MAD1) | AFT | Hercynian Basement* | 44.040 | 10.192 | 0.787 | 0.502 | 3.86 | 0.77 | 11.59 | Abbate et al. (1994) |
| ROM1 | AFT | Marnoso Arenacea Unit | 44.002 | 11.472 | 0.890 | 0.562 | 5.50 | 1.10 | 11.29 | Balesterieri et al. (2018) |
| ROM2 | AFT | Marnoso Arenacea Unit | 44.001 | 11.472 | 0.760 | 0.560 | 5.00 | 0.70 | 11.30 | Balesterieri et al. (2018) |
| ROM3 | AFT | Marnoso Arenacea Unit | 44.000 | 11.475 | 0.675 | 0.560 | 3.90 | 1.00 | 11.30 | Balesterieri et al. (2018) |
| ROM4 | AFT | Marnoso Arenacea Unit | 43.998 | 11.477 | 0.575 | 0.563 | 4.00 | 1.00 | 11.28 | Balesterieri et al. (2018) |
| ROM5 | AFT | Marnoso Arenacea Unit | 43.994 | 11.477 | 0.480 | 0.562 | 6.60 | 1.40 | 11.29 | Balesterieri et al. (2018) |
| TCGA | AFT | Marnoso Arenacea Unit | 43.993 | 11.476 | 0.360 | 0.560 | 5.00 | 1.60 | 11.30 | Balesterieri et al. (2018) |
| CAS1 | AFT | Macigno Unit | 44.206 | 10.446 | 1.300 | 1.061 | 8.93 | 1.34 | 8.79 | Balestrieri (2000) |
| CAS2 | AFT | Macigno Unit | 44.174 | 10.424 | 0.965 | 0.980 | 9.19 | 1.43 | 9.20 | Balestrieri (2000) |
| CAST2 | AFT | Macigno Unit | 44.105 | 10.415 | 0.270 | 0.811 | 8.91 | 1.30 | 10.04 | Balestrieri (2000) |
| CAST3 | AFT | Macigno Unit | 44.105 | 10.415 | 0.240 | 0.811 | 8.21 | 1.00 | 10.04 | Balestrieri (2000) |
| GOM2 | AFT | Macigno Unit | 44.125 | 10.642 | 1.850 | 1.061 | 9.51 | 1.40 | 8.79 | Balestrieri (2000) |
| GOM3 | AFT | Macigno Unit | 44.134 | 10.656 | 1.300 | 1.096 | 6.19 | 0.86 | 8.62 | Balestrieri (2000) |
| BOR2 | AFT | Gottero Sandstone | 44.4352 | 9.425 | 0.452 | 0.779 | 7.50 | 1.00 | 10.20 | Balestrieri et al. (1996) |
| BORI | AFT | Gottero Sandstone | 44.4352 | 9.425 | 0.450 | 0.779 | 6.40 | 1.00 | 10.20 | Balestrieri et al. (1996) |
| MG3 | AFT | Gottero Sandstone | 44.2345 | 9.472 | 0.000 | 0.193 | 9.70 | 1.10 | 13.13 | Balestrieri et al. (1996) |
| MS1 | AFT | Gottero Sandstone | 44.1338 | 9.638 | 0.000 | 0.136 | 9.70 | 1.10 | 13.42 | Balestrieri et al. (1996) |
| MS2 | AFT | Gottero Sandstone | 44.1338 | 9.638 | 0.000 | 0.136 | 7.60 | 0.70 | 13.42 | Balestrieri et al. (1996) |
| MS4 | AFT | Gottero Sandstone | 44.1338 | 9.638 | 0.000 | 0.136 | 8.00 | 1.20 | 13.42 | Balestrieri et al. (1996) |
| MS5 | AFT | Gottero Sandstone | 44.1338 | 9.638 | 0.000 | 0.136 | 8.70 | 1.20 | 13.42 | Balestrieri et al. (1996) |
| RAM1 | AFT | Gottero Sandstone | 44.434 | 9.311 | 1.318 | 0.632 | 8.60 | 1.10 | 10.94 | Balestrieri et al. (1996) |
| RAM3 | AFT | Gottero Sandstone | 44.4268 | 9.312 | 1.075 | 0.606 | 6.50 | 1.10 | 11.07 | Balestrieri et al. (1996) |
| RAM4 | AFT | Gottero Sandstone | 44.4268 | 9.312 | 1.075 | 0.606 | 7.50 | 1.00 | 11.07 | Balestrieri et al. (1996) |
| RAM5 | AFT | Gottero Sandstone | 44.4221 | 9.311 | 0.950 | 0.587 | 7.30 | 0.80 | 11.17 | Balestrieri et al. (1996) |
| RAM6 | AFT | Gottero Sandstone | 44.4221 | 9.311 | 0.948 | 0.587 | 6.50 | 0.70 | 11.17 | Balestrieri et al. (1996) |
| ZAT2 | AFT | Gottero Sandstone | 44.3908 | 9.442 | 1.349 | 0.637 | 9.50 | 1.30 | 10.92 | Balestrieri et al. (1996) |
| CH1 | AFT | Macigno Unit | 43.601 | 11.411 | 0.303 | 0.337 | 5.60 | 0.90 | 12.42 | Bonini et al. (2013) |
| CH2 | AFT | Macigno Unit | 43.541 | 11.430 | 0.504 | 0.379 | 6.10 | 1.00 | 12.20 | Bonini et al. (2013) |
| CH3 | AFT | Macigno Unit | 43.565 | 11.382 | 0.722 | 0.373 | 6.90 | 0.90 | 12.24 | Bonini et al. (2013) |
| CH4 | AFT | Macigno Unit | 43.562 | 11.380 | 0.857 | 0.376 | 7.40 | 1.00 | 12.22 | Bonini et al. (2013) |
| PR 11 | AFT | Subligurian | 44.463 | 9.930 | 0.880 | 0.808 | 8.70 | 1.10 | 10.06 | Carlini et al. (2013) |
| PR 12 | AFT | Tuscan Nappe | 44.353 | 9.776 | 0.668 | 0.700 | 8.70 | 1.20 | 10.60 | Carlini et al. (2013) |
| PR 15 | AFT | Ligurian | 44.472 | 9.966 | 1.085 | 0.837 | 7.30 | 1.90 | 9.92 | Carlini et al. (2013) |
| PR 17 | AFT | Ligurian | 44.379 | 10.196 | 0.860 | 1.001 | 4.10 | 0.50 | 9.10 | Carlini et al. (2013) |
| PR 18 | AFT | Subligurian | 44.380 | 10.194 | 0.780 | 1.001 | 4.60 | 0.80 | 9.10 | Carlini et al. (2013) |
| PR 20 | AFT | Tusc. | 44.338 | 10.528 | 0.500 | 0.881 | 2.30 | 0.30 | 9.69 | Carlini et al. (2013) |
| PR 22 | AFT | Subligurian | 44.331 | 10.564 | 1.082 | 0.851 | 2.50 | 0.50 | 9.84 | Carlini et al. (2013) |
| PR 23.1 | AFT | Ligurian | 44.329 | 10.562 | 1.111 | 0.861 | 4.70 | 0.90 | 9.79 | Carlini et al. (2013) |
| PR 25.1 | AFT | Tuscan Nappe | 44.320 | 9.995 | 0.248 | 0.639 | 7.00 | 0.90 | 10.91 | Carlini et al. (2013) |
| PR 26 | AFT | Tuscan Nappe | 44.463 | 9.602 | 1.135 | 0.883 | 5.40 | 0.90 | 9.68 | Carlini et al. (2013) |
| PR 27 | AFT | Epiligurian | 44.525 | 9.824 | 0.618 | 0.726 | 4.70 | 1.00 | 10.47 | Carlini et al. (2013) |
| PR 28.1 | AFT | Ligurian | 44.522 | 9.931 | 0.710 | 0.776 | 3.20 | 0.50 | 10.22 | Carlini et al. (2013) |
| PR 28.2 | AFT | Ligurian | 44.522 | 9.931 | 0.702 | 0.776 | 4.10 | 0.60 | 10.22 | Carlini et al. (2013) |
| PR 3 | AFT | Tuscan Nappe | 44.446 | 9.943 | 0.600 | 0.817 | 6.20 | 1.00 | 10.01 | Carlini et al. (2013) |





| PR 5 | AFT | Ligurian | 44.456 | 9.804 | 0.718 | 0.777 | 7.80 | 0.80 | 10.21 | Carlini et al. (2013) |
|---|---|---|---|---|---|---|---|---|---|---|
| PR 6.1 | AFT | Subligurian | 44.456 | 9.783 | 0.600 | 0.789 | 4.30 | 0.90 | 10.15 | Carlini et al. (2013) |
| PR 7 | AFT | Subligurian | 44.550 | 9.940 | 0.301 | 0.748 | 4.90 | 1.00 | 10.36 | Carlini et al. (2013) |
| 03GB07 | AFT | Macigno Unit | 44.124 | 10.059 | 0.675 | 0.356 | 7.90 | 0.90 | 12.32 | Felin et al. (2007) |
| 03RE20 | AFT | Macigno Unit | 44.098 | 10.326 | 1.055 | 0.760 | 7.50 | 1.00 | 10.30 | Felin et al. (2007) |
| MSV 2 | AFT | Pseudomacigno Apuan* | 44.106 | 10.288 | 0.654 | 0.752 | 5.70 | 0.75 | 10.34 | Felin et al. (2007) |
| S 1 | AFT | Macigno Unit | 44.178 | 10.160 | 0.546 | 0.662 | 6.50 | 0.95 | 10.79 | Felin et al. (2007) |
| S 3 | AFT | Macigno Unit | 44.139 | 10.073 | 0.494 | 0.374 | 6.60 | 0.85 | 12.23 | Felin et al. (2007) |
| S 4 | AFT | Macigno Unit | 44.128 | 10.059 | 0.636 | 0.330 | 5.10 | 0.85 | 12.45 | Felin et al. (2007) |
| SC 2 | AFT | Macigno Unit | 44.081 | 10.083 | 0.204 | 0.342 | 6.40 | 1.10 | 12.39 | Felin et al. (2007) |
| SM 3 | AFT | Macigno Unit | 44.164 | 10.129 | 0.250 | 0.558 | 8.80 | 1.40 | 11.31 | Felin et al. (2007) |
| SU 1 | AFT | Macigno Unit | 44.170 | 10.188 | 0.773 | 0.724 | 6.50 | 1.05 | 10.48 | Felin et al. (2007) |
| 050320-1a | AFT | Helminthoid Flysch | 44.263 | 10.664 | 1.112 | 0.976 | 7.30 | 2.30 | 9.22 | Thomson et al. (2010) |
| 050320-1b | AFT | Helminthoid Flysch | 44.263 | 10.664 | 1.112 | 0.976 | 5.00 | 1.30 | 9.22 | Thomson et al. (2010) |
| CIM1 | AFT | Cervarola Unit (Modino) | 44.194 | 10.699 | 2.165 | 1.121 | 7.53 | NA | 8.50 | Thomson et al. (2010) |
| CIM2 | AFT | Cervarola Unit (Modino) | 44.194 | 10.704 | 2.045 | 1.116 | 7.84 | NA | 8.52 | Thomson et al. (2010) |
| CIM3 | AFT | Cervarola Unit (Modino) | 44.196 | 10.692 | 1.950 | 1.124 | 7.44 | NA | 8.48 | Thomson et al. (2010) |
| CIM4 | AFT | Cervarola Unit (Modino) | 44.200 | 10.684 | 1.830 | 1.128 | 6.68 | NA | 8.46 | Thomson et al. (2010) |
| CIM5 | AFT | Cervarola Unit (Modino) | 44.202 | 10.677 | 1.750 | 1.134 | 7.22 | NA | 8.43 | Thomson et al. (2010) |
| CIM6 | AFT | Cervarola Unit (Modino) | 44.201 | 10.666 | 1.660 | 1.149 | 6.60 | NA | 8.35 | Thomson et al. (2010) |
| SILL1 | AFT | Macigno Unit | 44.368 | 10.064 | 1.861 | 0.860 | 8.70 | NA | 9.80 | Thomson et al. (2010) |
| SILL10 | AFT | Macigno Unit | 44.334 | 10.050 | 0.730 | 0.754 | 7.10 | NA | 10.33 | Thomson et al. (2010) |
| SILL2 | AFT | Macigno Unit | 44.361 | 10.074 | 1.790 | 0.863 | 9.60 | NA | 9.78 | Thomson et al. (2010) |
| SILL3 | AFT | Macigno Unit | 44.357 | 10.073 | 1.600 | 0.853 | 6.60 | NA | 9.84 | Thomson et al. (2010) |
| SILL4 | AFT | Macigno Unit | 44.455 | 10.076 | 1.530 | 0.951 | 5.80 | NA | 9.34 | Thomson et al. (2010) |
| SILL5 | AFT | Macigno Unit | 44.354 | 10.071 | 1.420 | 0.843 | 6.80 | NA | 9.89 | Thomson et al. (2010) |
| SILL6 | AFT | Macigno Unit | 44.353 | 10.057 | 1.260 | 0.815 | 6.80 | NA | 10.03 | Thomson et al. (2010) |
| SILL7 | AFT | Macigno Unit | 44.338 | 10.058 | 1.130 | 0.781 | 6.10 | NA | 10.19 | Thomson et al. (2010) |
| SILL9 | AFT | Macigno Unit | 44.334 | 10.050 | 0.780 | 0.755 | 5.30 | NA | 10.32 | Thomson et al. (2010) |
| VALD1 | AFT | Macigno Unit | 43.594 | 11.603 | 0.497 | 0.484 | 4.97 | NA | 11.68 | Thomson et al. (2010) |
| VALD10 | AFT | Macigno Unit | 43.653 | 11.640 | 1.400 | 0.836 | 7.33 | NA | 9.92 | Thomson et al. (2010) |
| VALD11 | AFT | Macigno Unit | 43.663 | 11.641 | 1.450 | 0.644 | 6.12 | NA | 10.88 | Thomson et al. (2010) |
| VALD12 | AFT | Macigno Unit | 43.696 | 11.673 | 0.960 | 0.717 | 6.63 | NA | 10.51 | Thomson et al. (2010) |
| VALD2 | AFT | Macigno Unit | 43.612 | 11.645 | 0.500 | 0.550 | 7.35 | NA | 11.35 | Thomson et al. (2010) |
| VALD3 | AFT | Macigno Unit | 43.614 | 11.656 | 0.580 | 0.559 | 6.73 | NA | 11.31 | Thomson et al. (2010) |
| VALD4 | AFT | Macigno Unit | 43.620 | 11.648 | 0.740 | 0.567 | 5.35 | NA | 11.27 | Thomson et al. (2010) |
| VALD5 | AFT | Macigno Unit | 43.621 | 11.656 | 0.850 | 0.572 | 4.43 | NA | 11.24 | Thomson et al. (2010) |
| VALD6 | AFT | Macigno Unit | 43.620 | 11.659 | 0.880 | 0.571 | 6.83 | NA | 11.25 | Thomson et al. (2010) |
| VALD7 | AFT | Macigno Unit | 43.604 | 11.651 | 1.100 | 0.536 | 6.88 | NA | 11.42 | Thomson et al. (2010) |
| VALD8 | AFT | Macigno Unit | 43.626 | 11.684 | 1.200 | 0.585 | 8.84 | NA | 11.18 | Thomson et al. (2010) |
| VALD9 | AFT | Macigno Unit | 43.646 | 11.652 | 1.200 | 0.619 | 8.58 | NA | 11.00 | Thomson et al. (2010) |
| C1 | AFT | Cervarola Unit | 44.113 | 11.002 | 0.500 | 0.789 | 6.70 | 0.08 | 10.15 | Ventura et al. (2001) |
| C10 | AFT | Cervarola Unit | 44.143 | 11.191 | 0.605 | 0.683 | 3.90 | 0.80 | 10.68 | Ventura et al. (2001) |
| C11 | AFT | Macigno Unit | 44.001 | 10.807 | 0.950 | 0.644 | 9.80 | 1.20 | 10.88 | Ventura et al. (2001) |
| C13 | AFT | Modino | 44.021 | 10.864 | 0.700 | 0.731 | 6.80 | 0.90 | 10.45 | Ventura et al. (2001) |
| C16 | AFT | Cervarola Unit | 44.060 | 10.913 | 0.630 | 0.807 | 2.70 | 0.80 | 10.07 | Ventura et al. (2001) |
| C17 | AFT | Cervarola Unit | 44.068 | 10.919 | 0.625 | 0.818 | 4.90 | 1.20 | 10.01 | Ventura et al. (2001) |
| C2 | AFT | Cervarola Unit | 44.004 | 11.012 | 0.830 | 0.548 | 6.50 | 0.80 | 11.36 | Ventura et al. (2001) |
| C22 | AFT | Cervarola Unit | 44.041 | 10.932 | 0.850 | 0.747 | 3.00 | 1.10 | 10.36 | Ventura et al. (2001) |
| C23 | AFT | Cervarola Unit | 44.021 | 10.929 | 0.884 | 0.689 | 5.20 | 1.00 | 10.65 | Ventura et al. (2001) |
| C29 | AFT | Cervarola Unit | 44.731 | 9.386 | 0.320 | 0.804 | 4.70 | NA | 10.08 | Ventura et al. (2001) |
| C3 | AFT | Cervarola Unit | 44.014 | 11.025 | 0.780 | 0.580 | 5.00 | 0.05 | 11.20 | Ventura et al. (2001) |
| C34 | AFT | Cervarola Unit | 44.417 | 9.949 | 0.510 | 0.820 | 8.60 | NA | 10.00 | Ventura et al. (2001) |
| C37 | AFT | Cervarola Unit | 44.246 | 10.683 | 0.641 | 1.016 | 2.60 | 0.50 | 9.02 | Ventura et al. (2001) |
| C38 | AFT | Cervarola Unit | 44.223 | 10.777 | 0.980 | 0.950 | 3.10 | 1.10 | 9.35 | Ventura et al. (2001) |
| C4 | AFT | Cervarola Unit | 44.028 | 11.038 | 0.680 | 0.620 | 3.30 | 0.50 | 11.00 | Ventura et al. (2001) |
| C40 | AFT | Cervarola Unit | 44.223 | 10.758 | 1.250 | 0.977 | 4.10 | 0.50 | 9.22 | Ventura et al. (2001) |
| C5 | AFT | Cervarola Unit | 44.049 | 11.044 | 0.610 | 0.679 | 5.70 | 0.60 | 10.71 | Ventura et al. (2001) |
| C52 | AFT | Cervarola Unit | 44.013 | 11.503 | 0.360 | 0.594 | 5.90 | 1.10 | 11.13 | Ventura et al. (2001) |
| C6 | AFT | Cervarola Unit | 44.095 | 11.044 | 0.650 | 0.735 | 5.00 | 0.60 | 10.42 | Ventura et al. (2001) |
| C7 | AFT | Cervarola Unit | 44.111 | 11.039 | 0.625 | 0.738 | 5.40 | 0.70 | 10.41 | Ventura et al. (2001) |
| C8 | AFT | Cervarola Unit | 44.115 | 11.207 | 0.700 | 0.680 | 6.20 | 0.60 | 10.70 | Ventura et al. (2001) |
| C9 | AFT | Cervarola Unit | 44.106 | 11.204 | 1.000 | 0.673 | 7.40 | 0.70 | 10.73 | Ventura et al. (2001) |
| 1927 | AFT | Marnoso Arenacea Unit | 44.064 | 11.597 | 0.350 | 0.667 | 8.50 | NA | 10.77 | Zattin et al. (2002) |





| 1929 | AFT | Marnoso Arenacea Unit | 44.037 | 11.504 | 0.700 | 0.619 | 6.40 | 0.70 | 11.00 | Zattin et al. (2002) |
| 1930 | AFT | Marnoso Arenacea Unit | 44.069 | 11.492 | 1.150 | 0.630 | 4.70 | NA | 10.95 | Zattin et al. (2002) |
| AP 10 | AFT | Marnoso Arenacea Unit | 44.121 | 11.396 | 0.500 | 0.685 | 3.90 | 0.70 | 10.67 | Zattin et al. (2002) |
| Ap 15 | AFT | Marnoso Arenacea Unit | 43.819 | 11.952 | 0.700 | 0.805 | 9.20 | 1.40 | 10.07 | Zattin et al. (2002) |
| AP 34 | AFT | Macigno Unit | 44.097 | 11.315 | 0.600 | 0.653 | 5.90 | 0.80 | 10.83 | Zattin et al. (2002) |
| AP 43 | AFT | Marnoso Arenacea | 43.818 | 11.733 | 0.515 | 0.805 | 5.30 | 0.80 | 10.08 | Zattin et al. (2002) |
| AP 45 | AFT | Marnoso Arenacea | 43.828 | 11.749 | 0.940 | 0.809 | 4.70 | 0.70 | 10.06 | Zattin et al. (2002) |
| AP 52 | AFT | Marnoso Arenacea Unit | 43.905 | 11.723 | 0.565 | 0.799 | 5.10 | 0.80 | 10.11 | Zattin et al. (2002) |
| AP 53 | AFT | Marnoso Arenacea Unit | 43.934 | 11.656 | 0.907 | 0.814 | 8.60 | 1.10 | 10.03 | Zattin et al. (2002) |
| AP 54 | AFT | Marnoso Arenacea | 43.961 | 11.670 | 0.690 | 0.747 | 5.60 | 0.70 | 10.36 | Zattin et al. (2002) |
| AP 55 | AFT | Marnoso Arenacea Unit | 44.013 | 11.687 | 1.070 | 0.672 | 7.90 | 0.80 | 10.74 | Zattin et al. (2002) |
| AP 56 | AFT | Marnoso Arenacea Unit | 43.983 | 11.686 | 0.500 | 0.723 | 4.10 | 0.70 | 10.49 | Zattin et al. (2002) |
| AP 57 | AFT | Marnoso Arenacea Unit | 43.995 | 11.719 | 0.450 | 0.688 | 3.60 | 0.50 | 10.66 | Zattin et al. (2002) |
| AP 9 | AFT | Marnoso Arenacea Unit | 44.115 | 11.431 | 0.400 | 0.673 | 3.90 | 0.70 | 10.73 | Zattin et al. (2002) |
| AP44 | AFT | Marnoso Arenacea Unit | 43.824 | 11.746 | 0.725 | 0.807 | 6.00 | 0.90 | 10.07 | Zattin et al. (2002) |
| AP47 | AFT | Marnoso Arenacea | 43.864 | 11.739 | 1.365 | 0.816 | 5.20 | 0.90 | 10.02 | Zattin et al. (2002) |

*Lithologies exposed in the Alpi Apuane metamorphic dome. These samples were excluded from the erosion rate inversions

**Table 3 Compilation of detrital AFT cooling ages and sample descriptions.**

| ID | Method | Lithology | Latitude | Longitude | Sample Elevation (km) | Age (Ma) | Error (2σ) | Error (1σ) | Reference |
|---|---|---|---|---|---|---|---|---|---|
| Enza | AFT | Ligurian/EpiLigurian/Macigno | 44.620 | 10.413 | 0.163 | 4.70 | 1.00 | 0.50 | Malusa and Balestrieri (2012) |
| Nure | AFT | Ligurian | 44.872 | 9.647 | 0.208 | 4.10 | 1.60 | 0.80 | Malusa and Balestrieri (2012) |
| Panaro | AFT | Ligurian/EpiLigurian/Macigno | 44.477 | 11.027 | 0.099 | 6.90 | 2.20 | 1.10 | Malusa and Balestrieri (2012) |
| Secchia | AFT | Ligurian/EpiLigurian/Macigno | 44.532 | 10.758 | 0.119 | 6.50 | 1.80 | 0.90 | Malusa and Balestrieri (2012) |
| Taro | AFT | Ligurian/ EpiLigurian | 44.713 | 10.120 | 0.117 | 4.60 | 1.60 | 0.80 | Malusa and Balestrieri (2012) |
| Trebbia | AFT | Ligurian | 44.901 | 9.584 | 0.140 | 4.00 | 1.40 | 0.70 | Malusa and Balestrieri (2012) |
| Bisenzio | AFT | Cervarola and Modino Units | 43.928 | 11.126 | 0.102 | 5.30 | 0.95 | 0.50 | *this study* |
| Lima1 | AFT | Cervarola/Modino/Macigno Units | 44.000 | 10.560 | 0.097 | 5.40 | 1.15 | 0.60 | *this study* |
| Lima2 | AFT | Cervarola/Modino/Macigno Units | 44.091 | 10.760 | 0.544 | 6.10 | 0.85 | 0.45 | *this study* |
| Magra1 | AFT | Ligurian and Macigno Units | 44.188 | 9.925 | 0.036 | 5.10 | 3.10 | 1.50 | *this study* |
| Magra2 | AFT | Macigno and Ligurian Units | 44.387 | 9.887 | 0.251 | 5.20 | 1.00 | 0.50 | *this study* |
| Pescia | AFT | Macigno Unit | 43.929 | 10.693 | 0.105 | 8.00 | 1.00 | 0.50 | *this study* |
| Serchio | AFT | Macigno Unit | 44.192 | 10.306 | 0.525 | 7.50 | 1.05 | 0.50 | *this study* |
| Vara | AFT | Ligurian and Macigno Units | 44.198 | 9.851 | 0.032 | 5.90 | 2.50 | 1.25 | *this study* |

**Table 4 Compilation of bedrock ZHe cooling ages and sample descriptions.**





| ID | Method | Lithology | Latitude | Longitude | Sample Elevation (km) | Age (Ma) | Error (2σ) | Mean Elevation (km) | Surface Temperature (°C) | Reference |
|---|---|---|---|---|---|---|---|---|---|---|
| CP3(4) | Zhe | Hercynian Basement Apuan autoch.* | 44.017 | 10.264 | 0.650 | 4.98 | 0.40 | 0.405 | 12.075 | Abbate et al. (1994) |
| G3(4) (G3A)a | Zhe | Hercynian Basement Apuan autoch.* | 44.067 | 10.199 | 0.170 | 5.7 | 0.46 | 0.451 | 11.843 | Abbate et al. (1994) |
| 020620-1 | ZHe | PseudoMacigno Unit/Apuan autoch.* | 44.096 | 10.325 | 0.958 | 3.61 | 0.29 | 0.708 | 10.560 | Fellin et al. (2007) |
| 020620-3 | ZHe | Macigno Unit | 44.122 | 10.068 | 0.756 | 9.35 | 0.75 | 0.395 | 12.125 | Fellin et al. (2007) |
| 020620-3 rep | ZHe | Macigno Unit | 44.122 | 10.068 | 0.756 | 9.27 | 0.74 | 0.395 | 12.125 | Fellin et al. (2007) |
| 03AP38 | ZHe | Met. Mesozoic succ. Massa Unit* | 44.069 | 10.139 | 0.925 | 5.94 | 0.47 | 0.392 | 12.140 | Fellin et al. (2007) |
| 03AP41 | ZHe | Hercynian Basement Massa Unit* | 44.050 | 10.161 | 0.080 | 6.44 | 0.52 | 0.387 | 12.164 | Fellin et al. (2007) |
| 03AP42 | ZHe | Hercynian Basement Apuan autoch.* | 44.069 | 10.175 | 0.125 | 7.19 | 0.58 | 0.430 | 11.949 | Fellin et al. (2007) |
| 03AP43 | ZHe | Hercynian Basement Massa Unit* | 44.048 | 10.179 | 0.505 | 5.11 | 0.41 | 0.399 | 12.104 | Fellin et al. (2007) |
| 03AP45 | ZHe | Hercynian Basement Massa Unit* | 44.032 | 10.194 | 0.810 | 5.93 | 0.47 | 0.386 | 12.172 | Fellin et al. (2007) |
| 03AP47 | ZHe | PseudoMacigno Unit/Apuan autoch.* | 44.128 | 10.259 | 0.890 | 4.62 | 0.37 | 0.693 | 10.634 | Fellin et al. (2007) |
| 03AP58 | ZHe | PseudoMacigno Unit/Apuan autoch.* | 44.003 | 10.308 | 0.305 | 4.33 | 0.35 | 0.410 | 12.050 | Fellin et al. (2007) |
| 03GB02 | ZHe | Hercynian Basement Apuan autoch.* | 43.995 | 10.248 | 0.080 | 5.41 | 0.43 | 0.356 | 12.318 | Fellin et al. (2007) |
| 03GB04 | ZHe | PseudoMacigno Unit/Apuan autoch.* | 43.974 | 10.277 | 0.600 | 6.93 | 0.55 | 0.333 | 12.434 | Fellin et al. (2007) |
| 03GB06 | ZHe | PseudoMacigno Unit/Apuan autoch.* | 43.966 | 10.330 | 0.440 | 5.98 | 0.48 | 0.348 | 12.358 | Fellin et al. (2007) |
| 03GB12 | ZHe | PseudoMacigno Unit/Apuan autoch.* | 44.013 | 10.303 | 0.670 | 5.09 | 0.41 | 0.428 | 11.958 | Fellin et al. (2007) |
| 03RE17 | ZHe | Hercynian Basement Apuan autoch.* | 44.036 | 10.253 | 0.799 | 5.29 | 0.42 | 0.437 | 11.917 | Fellin et al. (2007) |
| 03RE21 | ZHe | PseudoMacigno Unit/Apuan autoch.* | 44.075 | 10.327 | 0.810 | 6.40 | 0.51 | 0.645 | 10.875 | Fellin et al. (2007) |
| 03RE22 | ZHe | PseudoMacigno Unit/Apuan autoch.* | 44.066 | 10.324 | 0.510 | 4.77 | 0.38 | 0.611 | 11.043 | Fellin et al. (2007) |
| 03RE24 | ZHe | PseudoMacigno Unit/Apuan autoch.* | 44.159 | 10.200 | 0.915 | 5.58 | 0.45 | 0.666 | 10.771 | Fellin et al. (2007) |
| 03RE25A | ZHe | Hercynian Basement Apuan autoch.* | 44.133 | 10.186 | 1.500 | 5.42 | 0.43 | 0.582 | 11.190 | Fellin et al. (2007) |
| 03RE27 | ZHe | Hercyian Basment Massa Unit* | 44.071 | 10.155 | 0.500 | 5.54 | 0.44 | 0.412 | 12.040 | Fellin et al. (2007) |
| APUANE-1z1 | ZHe | Hercynian Basement Apuan autoch.* | 44.024 | 10.243 | 0.845 | 4.81 | 0.38 | 0.404 | 12.081 | Fellin et al. (2007) |
| APUANE-1z2 | ZHe | Hercynian Basement Apuan autoch.* | 44.024 | 10.243 | 0.845 | 4.58 | 0.37 | 0.404 | 12.081 | Fellin et al. (2007) |
| FIO4z2 | ZHe | PseudoMacigno Unit/Apuan autoch.* | 44.077 | 10.265 | 1.450 | 4.92 | 0.39 | 0.560 | 11.301 | Fellin et al. (2007) |
| FO4A | ZHe | PseudoMacigno Unit/Apuan autoch.* | 44.033 | 10.375 | 0.450 | 5.86 | 0.47 | 0.576 | 11.219 | Fellin et al. (2007) |

*Lithologies exposed in the Alpi Apuane metamorphic dome. These samples were excluded from the erosion rate inversions

We converted ages to erosion rates using a half-space cooling model and a closure temperature concept (Willett and Brandon, 2013). This model has the advantage of including an accurate representation of the transience associated with whole lithosphere geotherms. Reset ages were converted to erosion rates using the closure temperature concept (Dodson, 1979), with closure temperatures specific to each thermochronometer, although this is a simplification of diffusional daughter product loss that neglects effects associated with complex cooling histories. For monotonic cooling histories, the measured age of the sample is represented by the time needed for a rock to move from the closure depth to the surface (e.g. Reiners and Brandon, 2006).

The conversion to erosion rates was performed using the AGE2EDOT program, (Willett and Brandon, 2013), which estimates an erosion rate through solution of the 1-D thermal advection, diffusion problem for a lithospheric column subjected to a constant rate of erosion. Thermochronometric data required for the calculations include the measured ages and kinetic parameters from which a closure temperature is calculated. In addition, the thermal initial and boundary conditions, as well as thermal parameters, must be specified for each sample site.

For the kinetic parameters for AHe, we assumed grain sizes of 45 μm, given that sizes of dated grains are not reported by previous studies, and that a grain size of 60 μm is larger than the mean size of detrital apatites that are typically dated in the Northern Apennines. Thermal parameters (see definitions in Table 5) include an estimate for the age of onset of erosion ($t_1$);



the sample elevation, given as an elevation above a regional mean (h); surface temperature ($T_s$); and either an initial geothermal

gradient ($G_0$) or the final geothermal gradient ($G_f$). Only one estimate of the geothermal gradient is needed, but we took several

approaches, as will be discussed below. To calculate $T_s$, we adjusted a base temperature value for the elevation of each sample,

given a lapse rate of 5°C /km. For the base temperature, we used a modern surface temperature of 13.8°C, which represents

the calculated yearly average for an elevation of 53 m at Bologna from 1813–2004 (NOAA Global Temperature Summary of

170 the Year dataset).

**Table 5 Definitions of thermal parameters used in the erosion rate analysis.**

| Parameter | Description |
|---|---|
| $G_{0\_25}$ | Initial geothermal gradient of 25˚C/km (AGE2EDOT input) |
| $G_{f\_25}$ | Inferred final geothermal gradient (AGE2EDOT (output) |
| $G_{0\_heatflow}$ | Inferred initial geothermal gradient (AGE2EDOT output) |
| $G_{f\_heatflow}$ | Final geothermal gradient calculated from modern heat flow measurements (AGE2EDOT input) |
| h | Sample elevation above the regional mean elevation |
| tau | Thermochronometer cooling age |
| $t_{trans}$ | Transition time between AFT and AHe cooling intervals |
| $T_0$ | Temperature at transition time $t_{trans}$ |
| $T_s$ | Modern surface temperature |

The geothermal gradient is the most important parameter incorporated into the erosion rate analysis and is also the largest

source of uncertainty. It can be specified either as a final geothermal gradient ($G_f$), which is the present geothermal gradient at

the surface, or as an initial geothermal gradient ($G_0$) that is assumed to be constant with depth at the onset of exhumation

(Willett and Brandon, 2013). We calculated and compared erosion rates derived using two approaches. In the first method, we

imposed a spatially constant $G_0$ of 25°C/km ($G_{0\_25}$) (Balestrieri et al., 2003; Ventura et al., 2001; Zattin et al., 2002). In the

180 second method, we assumed that the present-day geothermal gradient ($G_f$) matches the geothermal gradient calculated from

geothermal heat flow measurements. We converted the heat flow measurements to a $G_f$ ($G_{f\_heatflow}$) using a spatially constant

thermal conductivity value for sandstone (2 W/mK). Heat flow values were extracted from contour maps that interpolate

geothermal well data (Pauselli et al., 2019; della Vedova et al., 2001). The della Vedova et al. (2001) flow map covers the

entire study area, whereas the Pauselli et al. (2019) map covers the area south of 44.5°N and includes only the Bisenzio River

(Fig. 3) within the study area. Because the heat flow map of della Vedova et al. (2001) is based on fewer geothermal well

measurements relative to the Pauselli et al. (2019) map, we consider the della Vedova et al. (2001) interpolation to have higher

uncertainties. Thus, where the Pauselli et al. (2019) map was available, a heat flow value was selected from this map.

Otherwise, a heat flow value was selected from the della Vedova et al. (2001) map.

The modelling procedure described above was applied to all ages, assuming that erosion initiated over the entire region at 10

Ma. The resulting erosion rate applies from the onset of exhumation at 10 Ma to the present and reflects the time-averaged



erosion rate constrained to pass through the closure temperature at the age and with a cooling rate commensurate with the average erosion rate. Thus, this method is limited to a single, average erosion rate. However, changes in exhumation rates through time in the Northern Apennines are supported by several lines of evidence, particularly by age-elevation transects

(AETs). In fact, AETs from the existing literature illustrate differences along the age-elevation slope for a single thermochronometer (as in Balestrieri et al., 1999) or among age-elevation slopes for multiple thermochronometers (as in Thomson et al., 2010).

It is possible to use AGE2EDOT in an incremental manner, allowing us to use paired thermochronometers analyzed from a

200 single sample. In this case, the temporal range of exhumation is bracketed by the AFT and AHe ages, with independent erosion rates determined from each age, thus resolving two time intervals (Willett et al., 2020). In principle, this violates the assumption of a constant rate of cooling implicit to use of the closure age concept, but provided that the transition between erosion rate intervals is not close to either age, the error will be small. We analyzed 30 available paired ages to detect temporal changes in erosion rate. For the paired ages analysis, the exhumation path is divided into two segments: the first segment extends from

205 the onset of exhumation to a specified transition time ($t_{trans}$) after cooling through the AFT system, and the second segment extends from this transition time to the present, thus passing through the AHe age in this second interval (Fig. 4). We derive an erosion rate for each of these time-segments by analyzing each segment with AGE2EDOT, linking the two solutions at $t_{trans}$. The solutions are matched by noting the depth and temperature of the sample at $t_{trans}$, based on the erosion rate in the second interval, and using this and the geothermal gradient at $t_{trans}$ as the boundary conditions for calculations of the first interval (Fig.

4).

The difference in age between some of our paired ages is less than 1 Ma, but larger than 0.5 Ma, so we set the transition at 0.5 Ma before the AHe closure for all samples (i.e. the AHe cooling age plus 0.5 Ma), in order to allow the onset of advection to precede the AHe closure. Calculation of the erosion rate over the second interval requires the modern surface temperature ($T_S$);

the sample elevation above the regional mean (h); the AHe cooling age (tau); the final geothermal gradient as derived from heat-flow measurements ($G_{f\_heatflow}$); and the length of the time interval ($t_{trans}$), calculated as the AHe age plus 0.5 Ma. The erosion rate is then solved from these data and the kinetic parameters.

To calculate the erosion rate for the first interval, we require in addition, the temperature at $t_{trans}$ ($T_0$) (Fig. 4); the sample

elevation above the regional mean (h); the sample AFT age (tau), and the time of onset of erosion, taken as 10 Ma. To match solutions at $t_{trans}$, we simply reduce the age by $t_{trans}$, and reduce the elevation by the amount of exhumation that occurred during the second interval. We take the initial geothermal gradient obtained from the model for the second interval as the final condition for the first interval.





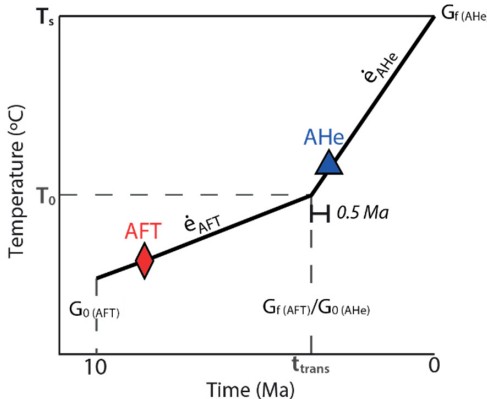

**Figure 4 Schematic of paired ages erosion rates analysis, illustrating the theoretical temperature path through time for a sample with paired AFT and AHe cooling ages, given the thermal parameters described in the text. è represents the erosion rate for each interval.**

### 2.3 Kinematic model

The kinematic model presented here approximates the Northern Apennines as a doubly tapering, asymmetric wedge, given the geometric parameters illustrated in Fig. 5. The Adriatic and Ligurian sides of the orogen are defined as the accreting prowedge and non-accreting retrowedge of the orogen, respectively (e.g. Willett et al., 2001). The geometry of the wedge is defined by surface and basal angles for the prowedge ($\alpha_P$ and $\beta_P$) and retrowedge ($\alpha_R$ and $\beta_R$). The lengths of the prowedge ($L_P$) and retrowedge ($L_R$) are 60 km and 40 km, respectively, based on average widths measured from an SRTM 90 Digital Elevation

Model (DEM). The maximum crustal thickness is 56 km (Spada et al., 2013), the maximum elevation is 2 km, and the thickness of the accreted crust is 20 km, partitioned between frontal accretion ($h_0 = 10$ km) and prowedge basal accretion ($h_1 = 10$ km). We assume no retrowedge accretion ($h_2 = 0$). Closure depths are set for ZHe (7.2 km), AFT (4.4 km), and AHe (2.8 km).

Material is accreted to the wedge through thrusts slices in the upper plate (frontal accretion) or is offscraped from the

240 subducting plate at depth (underplating). Material motion is constrained by balancing frontal and rearward fluxes, underplating, and erosion. We prescribe a compressional prowedge and an extensional retrowedge, where horizontal velocities decrease along the prowedge and increase along the retrowedge as a function of distance. The vertical rock velocity is also variable with depth, and is defined as the sum of the erosion rate and a component of crustal thickening driven by accretion.

The velocities in the model are defined as follows: plate subduction velocity ($V_P$), prowedge underplating velocity ($U_P$), prowedge erosional velocity ($e_P$), and retrowedge erosional velocity ($e_R$). The plate subduction velocity, or convergence rate, for the Northern Apennines is suggested to be driven entirely by slab rollback, so we used estimates of slab rollback to parameterize the convergence rate. Slab rollback rates are on the order of 6–10 km/My in this region of the Apennines





(Faccenna et al., 2014; Rosenbaum and Piana Agostinetti, 2015), so we run the model using these minimum and maximum

values as end-member scenarios. We also vary the spatial pattern of erosion rates in the model using two model assumptions:

1) a constant erosion rate across the orogenic wedge (SCR), and 2) a higher erosion rate on the prowedge relative to the

retrowedge (VER).

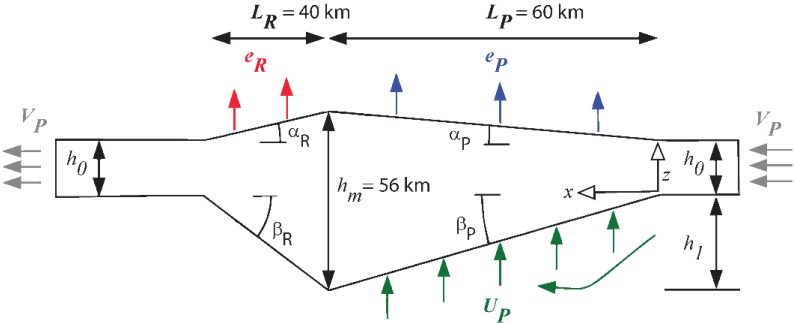

**Figure 5 Kinematic model of the Northern Apennines as an orogenic wedge with internal deformation driven by frontal and basal**
**accretion and surface erosion. Mass is balanced to maintain a steady size, and internal deformation is calculated to be consistent**
**with boundary conditions.**

## 3 Results

### 3.1 Detrital AFT cooling ages

New detrital AFT (8) sample ages are given in Fig. 3b and Tables 6–7. Central ages vary from $5.4 \pm 0.6$ Ma to $10.5 \pm 0.7$ Ma,

and single grain ages show a wide range of values from 5.1 to 145.3 Ma. All samples except Lima1 show at least two distinct

age populations, with minimum age peaks between 5.1 and 8 Ma. All minimum age peaks are younger than the stratigraphic

ages of units within the catchment (Cita Sironi et al., 2006; Delfrati et al., 2002; Pialli et al., 2000), with the exception of

Magra1. This site contains Plio-Pleistocene deposits exposed in its catchment that are younger than the minimum age peak,

but these are locally derived sediments, and the sedimentary bedrock has stratigraphic ages older than the young peak.

In the five southern samples (Serchio, Lima1, Lima2, Pescia and Bisenzio), the youngest peaks represent the largest age

populations and are similar to the sample central ages. The three northern samples (Vara, Magra1, Magra2) show two common

age populations at 5–6 Ma and at 12–13 Ma and have central ages older than the minimum peak ages, due to a large proportion

of older grains.

**Table 6 Central Ages and AFT dataset details.**



| Sample | Sampling Site | Lat | Long | Mount Num. | Num. of Grains | $\rho_s$ (x 10$^5$ cm$^{-2}$) | Ns | $\rho_i$ (x 10$^5$ cm$^{-2}$) | Ni | $\rho_D$ (x 10$^5$ cm$^{-2}$) | AFT Central age (Ma) | AFT Central age 1σ (Ma) | P(χ2) (%) | Age disp (%) |
|---|---|---|---|---|---|---|---|---|---|---|---|---|---|---|
| Vara | Piana Battolla | 44.1950° | 9.8569° | 1 | 74 | 1.13 | 283 | 30.7 | 7672 | 14.80 ± 0.00 | 10.53 | 0.74 | 0 | 49 |
| Vara | Piana Battolla | 44.1950° | 9.8569° | 2 | 76 | 1.83 | 432 | 39.0 | 9212 | 15.10 ± 0.01 | | | | |
| Magra2 | Pontremoli | 44.3873° | 9.8868° | 1 | 31 | 0.94 | 117 | 34.5 | 4285 | 14.30 ± 0.03 | 6.94 | 0.6 | 0 | 50 |
| Magra2 | Pontremoli | 44.3873° | 9.8868° | 2 | 69 | 1.17 | 220 | 41.8 | 7878 | 14.80 ± 0.03 | | | | |
| Magra1 | Isola | 44.1867° | 9.9258° | 1 | 127 | 0.95 | 642 | 33.9 | 22978 | 16.20 ± 0.03 | 7.81 | 0.45 | 0 | 27 |
| Magra1 | Isola | 44.1867° | 9.9258° | 2 | 23 | 0.67 | 72 | 30.9 | 3342 | 15.90 ± 0.03 | | | | |
| Serchio | Piazza al Serchio | 44.1920° | 10.3016° | 1 | 72 | 1.36 | 359 | 37.2 | 9834 | 12.80 ± 0.03 | 8.08 | 0.51 | 0 | 18 |
| Serchio | Piazza al Serchio | 44.1920° | 10.3016° | 2 | 28 | 0.96 | 78 | 34.6 | 2801 | 13.30 ± 0.03 | | | | |
| Lima2 | Cutigliano | 44.0907° | 10.7596° | 1 | 62 | 1.00 | 237 | 43.1 | 10202 | 15.00 ± 0.03 | 6.63 | 0.46 | 0 | 29 |
| Lima2 | Cutigliano | 44.0907° | 10.7596° | 2 | 38 | 1.15 | 152 | 41.8 | 5527 | 14.20 ± 0.04 | | | | |
| Lima1 | Borgo a Mozzano | 43.9993° | 10.5540° | 1 | 31 | 0.75 | 97 | 35.9 | 4616 | 14.50 ± 0.03 | 5.41 | 0.59 | 89 | 0 |
| Bisenzio | Vaiano | 43.9277° | 11.1258° | 1 | 87 | 1.02 | 215 | 41.2 | 8992 | 14.84 ± 0.03 | 7.09 | 0.71 | 0 | 57 |
| Pescia | Pietrabuona | 43.9294° | 10.6933° | 1 | 33 | 1.43 | 209 | 42.4 | 6192 | 13.80 ± 0.03 | 8.95 | 0.69 | 0 | 37 |
| Pescia | Pietrabuona | 43.9294° | 10.6933° | 2 | 44 | 0.29 | 242 | 3.5 | 6644 | 13.90 ± 0.03 | | | | |

$\rho_s$: spontaneous track density

$\rho_i$: Induced track density in external detector

$\rho_D$ : induced track density in external detector adjacent to dosimeter glass

Age disp: Age Dispersion

**Table 7 Peak Ages with standard error and size of major peaks (%).**

| Sample | Sampling Site | Peak Age (Ma) ± 1σ and size of major peaks | | | | | | | | | |
|---|---|---|---|---|---|---|---|---|---|---|---|
| Vara | Piana Battolla | 5.9 | $^{+1.4}/_{-1.1}$ | (35%) | 13 | $^{+1.0}/_{-0.9}$ | (65%) | 145.3 | $^{+57.3}/_{-41.2}$ | (1%) | |
| Magra2 | Pontremoli | 5.2 | $^{+0.5}/_{-0.5}$ | (79%) | 13.4 | $^{+2.0}/_{-1.8}$ | (21%) | | | | |
| Magra1 | Isola | 5.1 | $^{+1.7}/_{-1.3}$ | (28%) | 8.2 | $^{+0.8}/_{-0.7}$ | (60%) | 12.3 | $^{+13.5}/_{-6.4}$ | (12%) |
| Serchio | Piazza al Serchio | 7.5 | $^{+0.5}/_{-0.5}$ | (93%) | 18.3 | $^{+10.3}/_{-6.6}$ | (6%) | 99.2 | $^{+286.5}/_{-74.1}$ | (1%) |
| Lima2 | Cutigliano | 6.1 | $^{+0.5}/_{-0.4}$ | (93%) | 17.4 | $^{+7.0}/_{-5.0}$ | (7%) | | | | |
| Lima1 | Borgo a Mozzano | 5.4 | $^{+0.6}/_{-0.6}$ | (100%) | | | | | | | |
| Bisenzio | Vaiano | 5.3 | $^{+0.5}/_{-0.5}$ | (90%) | 29.4 | $^{+7.5}/_{-6.0}$ | (10%) | | | | |
| Pescia | Pietrabuona | 8.0 | $^{+0.5}/_{-0.5}$ | (92%) | 24.5 | $^{+5.8}/_{-4.7}$ | (7%) | 111.1 | $^{+78.8}/_{-46.3}$ | (1%) |

### 3.2 Geothermal gradients and erosion rates

We report initial geothermal gradients (G$_0$) and final geothermal gradients (G$_f$) using the two approaches described in the methods. Given a G$_0$ = 25 °C/km common to all samples, G$_{f\_25}$ ranges from 27.4 to 55.2 °C/km for AHe samples (Table 8)

and ranges from 31.2 to 49.7 °C/km for AFT samples (Table 9). Using the second method based on modern heat flow measurements, G$_{0\_heatflow}$ for AHe samples ranges from 9.3 to 42.2 °C/km (Table 8) and ranges from 12.4 to 38.0 °C/km for





AFT samples (Table 9). Relative to the $G_{f\_25}$, $G_{f\_heatflow}$ derived from Pauselli et al. (2019) are consistently lower, where all samples lie left of the 1:1 trendline for AHe samples (Fig. 6e) and all but one lie left of the 1:1 trendline for AFT samples (Fig. 6b). In contrast, $G_{f\_heatflow}$ derived from della Vedova et al. (2001) are highly variable, although the majority lie to the right of

the 1:1 line for both AHe (Fig. 6e) and AFT (Fig. 6b) samples, indicating that these values are higher relative to $G_{f\_25}$.

Erosion rates calculated using the two methods for estimating geothermal gradients also illustrate different trends for the della Vedova et al. (2001) and Pauselli et al. (2019) heat flow estimates. Erosion rates derived from $G_{0\_25}$ plotted against erosion rates derived from della Vedova et al. (2001) $G_{f\_heatflow}$ lie mostly on the 1:1 trendline for both AFT and AHe samples (Fig.

6c,f) and are thus similar. In contrast, erosion rates calculated with Pauselli et al. (2019) $G_{f\_heatflow}$ are lower relative to erosion rates derived from $G_{0\_25}$, but always by a factor of less than two (Fig. 6c,f).

**Table 8 Erosion rates and parameters for AHe bedrock samples.**



| ID | Method | Latitude | Longitude | Sample Elevation (km) | Mean Elevation (km) | Imposed G₀ = 25 (°C/km) | | | | | G_f calculated from heat flow measurements | | | | | Heat Flow Measurement Source |
|---|---|---|---|---|---|---|---|---|---|---|---|---|---|---|---|---|
| | | | | | | Inital Geothermal Gradient (°C/km) | Final Geothermal Gradient (°C/km) | Erosion Rate (km/My) | Closure Depth (km) | Closure Temperature (°C) | Inital Geothermal Gradient (°C/km) | Final Geothermal Gradient (°C/km) | Erosion Rate (km/My) | Closure Depth (km) | Closure Temperature | |
| 020620-3 | AHe | 44.122 | 10.068 | 0.756 | 0.383 | 25.0 | 35.1 | 0.558 | 2.2 | 66.2 | 29.6 | 40.0 | 0.500 | 1.84 | 66.7 | della Vedova et al. (2001) |
| 03AP08AB | AHe | 44.190 | 10.632 | 0.880 | 1.295 | 25.0 | 32.4 | 0.426 | 2.3 | 64.0 | 35.8 | 42.5 | 0.282 | 1.56 | 63.5 | della Vedova et al. (2001) |
| 03AP12A | AHe | 44.110 | 10.735 | 0.815 | 1.155 | 25.0 | 30.6 | 0.327 | 2.1 | 61.9 | 40.1 | 45.0 | 0.188 | 1.32 | 61.3 | della Vedova et al. (2001) |
| 03AP23A | AHe | 44.129 | 10.429 | 0.425 | 0.709 | 25.1 | 27.4 | 0.144 | 1.7 | 53.6 | 19.7 | 40.0 | 0.188 | 2.15 | 52.9 | della Vedova et al. (2001) |
| 03AP23B | AHe | 44.129 | 10.429 | 0.425 | 0.709 | 25.0 | 29 | 0.241 | 2.0 | 59.6 | 36.4 | 40.0 | 0.156 | 1.34 | 59.2 | della Vedova et al. (2001) |
| 03AP28A | AHe | 44.111 | 10.529 | 1.035 | 0.899 | 25.0 | 30.1 | 0.302 | 2.1 | 61.0 | 34.8 | 40.0 | 0.229 | 1.49 | 61.5 | della Vedova et al. (2001) |
| 03AP28C | AHe | 44.111 | 10.529 | 1.035 | 0.899 | 25.0 | 29.3 | 0.262 | 2.0 | 59.8 | 35.5 | 40.0 | 0.194 | 1.43 | 60.3 | della Vedova et al. (2001) |
| 03AP28D | AHe | 44.111 | 10.529 | 1.035 | 0.899 | 25.0 | 30.5 | 0.325 | 2.1 | 61.6 | 34.3 | 40.0 | 0.250 | 1.53 | 61.9 | della Vedova et al. (2001) |
| 03AP29A | AHe | 44.130 | 10.542 | 1.320 | 1.050 | 25.0 | 31.9 | 0.399 | 2.2 | 63.4 | 32.7 | 40.0 | 0.326 | 1.67 | 63.6 | della Vedova et al. (2001) |
| 03AP31A | AHe | 44.142 | 10.553 | 1.815 | 1.131 | 25.0 | 31 | 0.353 | 2.1 | 62.0 | 33.4 | 40.0 | 0.293 | 1.62 | 62.6 | della Vedova et al. (2001) |
| 03AP31B | AHe | 44.142 | 10.553 | 1.815 | 1.131 | 25.1 | 33.6 | 0.487 | 2.3 | 64.9 | 33.1 | 42.5 | 0.411 | 1.72 | 65.5 | della Vedova et al. (2001) |
| 03AP34 | AHe | 44.066 | 10.107 | 0.285 | 0.340 | 25.1 | 29.1 | 0.244 | 1.9 | 59.6 | 36.0 | 40.0 | 0.172 | 1.31 | 59.7 | della Vedova et al. (2001) |
| 03AP51 | AHe | 44.014 | 10.380 | 1.060 | 0.688 | 25.0 | 30.4 | 0.322 | 2.0 | 61.5 | 34.2 | 40.0 | 0.256 | 1.50 | 62.0 | della Vedova et al. (2001) |
| 03AP51C | AHe | 44.014 | 10.380 | 1.060 | 0.688 | 25.0 | 29.7 | 0.279 | 2.0 | 60.9 | 35.0 | 40.0 | 0.218 | 1.43 | 60.9 | della Vedova et al. (2001) |
| 03AP52A | AHe | 44.084 | 10.463 | 0.370 | 0.582 | 25.1 | 28.7 | 0.222 | 1.9 | 58.8 | 36.6 | 40.0 | 0.146 | 1.30 | 58.6 | della Vedova et al. (2001) |
| 03AP52B | AHe | 44.084 | 10.463 | 0.370 | 0.582 | 25.0 | 28.7 | 0.223 | 1.9 | 58.8 | 36.6 | 40.0 | 0.147 | 1.30 | 58.7 | della Vedova et al. (2001) |
| 03AP52C | AHe | 44.084 | 10.463 | 0.370 | 0.582 | 25.0 | 28.2 | 0.194 | 1.9 | 57.7 | 37.1 | 40.0 | 0.125 | 1.25 | 57.5 | della Vedova et al. (2001) |
| 03GB07 | AHe | 44.124 | 10.059 | 0.675 | 0.356 | 25.0 | 32 | 0.402 | 2.0 | 63.4 | 32.7 | 40.0 | 0.333 | 1.58 | 63.9 | della Vedova et al. (2001) |
| 03GB09 | AHe | 44.162 | 10.115 | 0.335 | 0.546 | 25.0 | 32.5 | 0.432 | 2.1 | 64.2 | 35.2 | 42.5 | 0.307 | 1.50 | 64.0 | della Vedova et al. (2001) |
| 03GB10 | AHe | 44.177 | 10.156 | 0.530 | 0.656 | 25.0 | 29.4 | 0.265 | 2.0 | 60.1 | 38.2 | 42.5 | 0.172 | 1.29 | 60.1 | della Vedova et al. (2001) |
| 03RE02 | AHe | 44.148 | 10.438 | 0.765 | 0.836 | 25.0 | 29.3 | 0.258 | 2.0 | 59.9 | 35.8 | 40.0 | 0.182 | 1.40 | 60.0 | della Vedova et al. (2001) |
| 03RE05A | AHe | 44.188 | 10.480 | 1.495 | 1.138 | 25.1 | 30.8 | 0.336 | 2.1 | 61.7 | 34.0 | 40.0 | 0.269 | 1.59 | 62.4 | della Vedova et al. (2001) |
| 03RE05B | AHe | 44.188 | 10.480 | 1.495 | 1.138 | 25.1 | 31.8 | 0.392 | 2.2 | 63.2 | 32.8 | 40.0 | 0.322 | 1.68 | 63.6 | della Vedova et al. (2001) |
| 03RE05C | AHe | 44.188 | 10.480 | 1.495 | 0.582 | 25.0 | 30.7 | 0.342 | 2.0 | 61.6 | 36.0 | 42.5 | 0.278 | 1.44 | 62.8 | della Vedova et al. (2001) |
| 03RE05CD | AHe | 44.188 | 10.480 | 1.495 | 1.138 | 25.0 | 29 | 0.243 | 2.0 | 58.0 | 35.7 | 40.0 | 0.185 | 1.42 | 59.1 | della Vedova et al. (2001) |
| 03RE05D | AHe | 44.188 | 10.480 | 1.495 | 0.582 | 25.0 | 32.4 | 0.427 | 2.1 | 63.6 | 34.2 | 42.5 | 0.360 | 1.56 | 64.7 | della Vedova et al. (2001) |
| 03RE06A | AHe | 44.201 | 10.488 | 1.640 | 1.194 | 25.0 | 31.9 | 0.395 | 2.2 | 63.0 | 35.1 | 42.5 | 0.313 | 1.59 | 63.8 | della Vedova et al. (2001) |
| 03RE06B | AHe | 44.201 | 10.488 | 1.640 | 1.194 | 25.0 | 31.3 | 0.367 | 2.2 | 62.4 | 35.7 | 42.5 | 0.286 | 1.54 | 63.3 | della Vedova et al. (2001) |
| 03RE12A | AHe | 44.059 | 10.767 | 0.460 | 0.942 | 25.0 | 28.8 | 0.229 | 2.0 | 59.2 | 42.2 | 45.0 | 0.104 | 1.14 | 57.5 | della Vedova et al. (2001) |
| 03RE12B | AHe | 44.059 | 10.767 | 0.460 | 0.942 | 25.0 | 29 | 0.243 | 2.0 | 59.6 | 42.0 | 45.0 | 0.112 | 1.16 | 57.9 | della Vedova et al. (2001) |
| 03RE14A | AHe | 44.005 | 10.665 | 0.840 | 0.635 | 25.0 | 31.5 | 0.377 | 2.1 | 62.9 | 33.2 | 40.0 | 0.303 | 1.58 | 63.2 | della Vedova et al. (2001) |
| 03RE14B | AHe | 44.005 | 10.665 | 0.840 | 0.635 | 25.0 | 27.9 | 0.184 | 1.8 | 56.7 | 36.7 | 40.0 | 0.139 | 1.18 | 54.3 | della Vedova et al. (2001) |
| 03RE20 | AHe | 44.098 | 10.326 | 1.055 | 0.808 | 25.0 | 32.5 | 0.432 | 2.2 | 64.1 | 32.1 | 40.0 | 0.358 | 1.69 | 64.3 | della Vedova et al. (2001) |
| 03RE7 | AHe | 44.200 | 10.676 | 1.600 | 1.214 | 25.0 | 38.6 | 0.727 | 2.4 | 68.7 | 30.7 | 45.0 | 0.639 | 1.99 | 69.1 | della Vedova et al. (2001) |
| 03RE7R1 | AHe | 44.200 | 10.676 | 1.600 | 1.214 | 25.0 | 37.5 | 0.676 | 2.4 | 68.0 | 31.7 | 45.0 | 0.581 | 1.90 | 68.5 | della Vedova et al. (2001) |
| 03TH02 | AHe | 44.086 | 10.568 | 0.979 | 0.881 | 25.0 | 29.9 | 0.291 | 2.0 | 60.7 | 37.5 | 42.5 | 0.206 | 1.37 | 61.2 | della Vedova et al. (2001) |
| 03TH02B | AHe | 44.086 | 10.568 | 0.979 | 0.881 | 25.0 | 29.3 | 0.261 | 2.0 | 59.8 | 38.1 | 42.5 | 0.181 | 1.33 | 60.3 | della Vedova et al. (2001) |
| 03TH12B | AHe | 44.080 | 10.600 | 0.678 | 0.904 | 25.1 | 31.4 | 0.371 | 2.1 | 63.0 | 33.8 | 40.0 | 0.273 | 1.57 | 62.7 | della Vedova et al. (2001) |
| 03TH13A | AHe | 44.013 | 10.593 | 0.153 | 0.578 | 25.0 | 27.6 | 0.160 | 1.8 | 56.4 | 16.0 | 40.0 | 0.273 | 2.82 | 56.5 | della Vedova et al. (2001) |
| 03TH13C | AHe | 44.013 | 10.593 | 0.153 | 0.578 | 25.0 | 28 | 0.182 | 1.8 | 57.4 | 37.6 | 40.0 | 0.102 | 1.20 | 56.4 | della Vedova et al. (2001) |
| 03TH18A | AHe | 43.980 | 10.552 | 0.047 | 0.449 | 25.1 | 28.3 | 0.197 | 1.8 | 58.0 | 37.3 | 40.0 | 0.114 | 1.22 | 57.2 | della Vedova et al. (2001) |
| 03TH23A | AHe | 44.124 | 10.628 | 1.645 | 1.205 | 25.0 | 33.6 | 0.485 | 2.3 | 64.9 | 33.3 | 42.5 | 0.400 | 1.72 | 65.4 | della Vedova et al. (2001) |
| 03TH23BD | AHe | 44.124 | 10.628 | 1.645 | 1.205 | 25.1 | 32.6 | 0.431 | 2.2 | 63.8 | 34.3 | 42.5 | 0.347 | 1.64 | 64.4 | della Vedova et al. (2001) |
| 03TH23C | AHe | 44.124 | 10.628 | 1.645 | 1.205 | 25.0 | 33.9 | 0.502 | 2.3 | 65.1 | 33.0 | 42.5 | 0.418 | 1.75 | 65.8 | della Vedova et al. (2001) |
| 050320-1C | AHe | 44.263 | 10.664 | 1.112 | 1.024 | 25.0 | 51.7 | 1.255 | 2.6 | 74.7 | 17.8 | 42.5 | 1.533 | 3.68 | 74.4 | della Vedova et al. (2001) |
| 050320-1D | AHe | 44.263 | 10.664 | 1.112 | 1.024 | 25.0 | 42.5 | 0.896 | 2.5 | 70.8 | 25.0 | 42.5 | 0.896 | 2.48 | 70.8 | della Vedova et al. (2001) |
| 050320-2B | AHe | 44.276 | 10.674 | 1.239 | 0.965 | 25.0 | 31.9 | 0.401 | 2.2 | 63.3 | 35.2 | 42.5 | 0.311 | 1.55 | 63.9 | della Vedova et al. (2001) |
| 050320-2C | AHe | 44.276 | 10.674 | 1.239 | 0.965 | 25.0 | 31.7 | 0.392 | 2.2 | 63.1 | 35.4 | 42.5 | 0.302 | 1.54 | 63.8 | della Vedova et al. (2001) |
| 050320-3A | AHe | 44.280 | 10.668 | 1.272 | 0.957 | 25.0 | 28.9 | 0.237 | 2.0 | 58.2 | 38.3 | 42.5 | 0.171 | 1.30 | 59.3 | della Vedova et al. (2001) |
| 050320-3B | AHe | 44.280 | 10.668 | 1.272 | 0.957 | 25.0 | 31.2 | 0.360 | 2.1 | 63.0 | 35.9 | 42.5 | 0.276 | 1.50 | 63.0 | della Vedova et al. (2001) |
| 050320-3C | AHe | 44.280 | 10.668 | 1.272 | 0.957 | 25.0 | 30.6 | 0.331 | 2.1 | 61.6 | 36.5 | 42.5 | 0.249 | 1.45 | 62.4 | della Vedova et al. (2001) |
| 1926 | AHe | 44.107 | 11.729 | 0.250 | 0.471 | 25.1 | 29.2 | 0.249 | 1.9 | 59.8 | 21.3 | 25.5 | 0.295 | 2.26 | 59.9 | Pauselli et al. (2019) |
| 1926B | AHe | 44.107 | 11.729 | 0.250 | 0.471 | 25.0 | 33.2 | 0.458 | 2.1 | 64.8 | 17.2 | 25.5 | 0.650 | 3.08 | 64.9 | Pauselli et al. (2019) |
| 1926C | AHe | 44.107 | 11.729 | 0.250 | 0.471 | 25.0 | 29.4 | 0.260 | 1.9 | 60.1 | 21.1 | 25.5 | 0.311 | 2.30 | 60.1 | Pauselli et al. (2019) |
| 1926D | AHe | 44.107 | 11.729 | 0.250 | 0.471 | 25.0 | 34 | 0.504 | 2.1 | 65.6 | 16.5 | 25.5 | 0.723 | 3.25 | 65.6 | Pauselli et al. (2019) |
| 1929 | AHe | 44.037 | 11.504 | 0.700 | 0.702 | 25.0 | 42.9 | 0.910 | 2.4 | 71.1 | 12.6 | 28.5 | 1.431 | 4.78 | 70.7 | Pauselli et al. (2019) |
| AP1 | AHe | 43.790 | 12.146 | 0.700 | 0.776 | 25.0 | 45.7 | 1.022 | 2.5 | 72.3 | 15.1 | 23.5 | 1.431 | 4.12 | 72.3 | Pauselli et al. (2019) |
| AP17 | AHe | 43.876 | 12.110 | 0.600 | 0.605 | 25.0 | 40.2 | 0.800 | 2.5 | 69.8 | 9.4 | 22.5 | 1.539 | 6.17 | 69.1 | Pauselli et al. (2019) |
| AP2 | AHe | 43.789 | 12.151 | 0.600 | 0.771 | 25.0 | 36.4 | 0.622 | 2.3 | 67.4 | 13.1 | 24.0 | 1.033 | 4.36 | 67.4 | Pauselli et al. (2019) |
| AP3 | AHe | 43.815 | 12.149 | 0.900 | 0.727 | 25.0 | 35.4 | 0.575 | 2.2 | 66.5 | 14.1 | 23.5 | 0.867 | 3.95 | 65.9 | Pauselli et al. (2019) |
| AP30 | AHe | 43.895 | 11.779 | 0.750 | 0.879 | 25.0 | 42 | 0.873 | 2.4 | 70.7 | 12.0 | 27.5 | 1.454 | 5.07 | 70.6 | Pauselli et al. (2019) |
| AP33 | AHe | 43.919 | 11.792 | 0.650 | 0.801 | 25.0 | 45.3 | 1.011 | 2.5 | 72.3 | 9.3 | 27.0 | 1.921 | 6.64 | 72.1 | Pauselli et al. (2019) |
| AP36E | AHe | 44.097 | 11.955 | 0.370 | 0.271 | 25.0 | 28.1 | 0.189 | 1.7 | 56.3 | 19.4 | 22.5 | 0.236 | 2.20 | 55.6 | Pauselli et al. (2019) |
| AP37 | AHe | 44.015 | 11.951 | 0.150 | 0.423 | 25.0 | 33.4 | 0.478 | 2.1 | 65.1 | 12.5 | 21.0 | 0.870 | 4.24 | 65.2 | Pauselli et al. (2019) |
| AP38 | AHe | 43.797 | 11.914 | 1.200 | 0.858 | 25.0 | 43.6 | 0.947 | 2.5 | 71.5 | 10.9 | 26.0 | 1.533 | 5.55 | 70.2 | Pauselli et al. (2019) |
| AP43R1 | AHe | 43.818 | 11.733 | 0.515 | 0.860 | 25.1 | 33.1 | 0.458 | 2.2 | 64.8 | 21.3 | 29.5 | 0.537 | 2.58 | 64.8 | Pauselli et al. (2019) |
| AP43R2 | AHe | 43.818 | 11.733 | 0.515 | 0.860 | 25.1 | 32.5 | 0.425 | 2.2 | 64.1 | 22.0 | 29.5 | 0.485 | 2.48 | 64.3 | Pauselli et al. (2019) |
| AP44R1 | AHe | 43.824 | 11.746 | 0.725 | 0.868 | 25.0 | 38.7 | 0.735 | 2.4 | 69.1 | 16.2 | 29.5 | 1.017 | 3.64 | 68.9 | Pauselli et al. (2019) |
| AP45R1 | AHe | 43.844 | 11.749 | 0.940 | 0.897 | 25.0 | 35.3 | 0.568 | 2.3 | 66.5 | 19.5 | 29.5 | 0.687 | 2.90 | 66.3 | Pauselli et al. (2019) |
| AP45R2 | AHe | 43.844 | 11.749 | 0.940 | 0.897 | 25.0 | 53.1 | 1.304 | 2.6 | 75.1 | NA | 29.0 | 0.687 | 0.99 | 75.7 | Pauselli et al. (2019) |



| | | | | | | | | | | | | | | | |
|---|---|---|---|---|---|---|---|---|---|---|---|---|---|---|---|
| AP47R1 | AHe | 43.864 | 11.739 | 1.365 | 0.906 | 25.0 | 42 | 0.873 | 2.4 | 70.6 | 14.5 | 29.0 | 1.200 | 4.15 | 69.7 | Pauselli et al. (2019) |
| AP48R1 | AHe | 43.879 | 11.711 | 1.655 | 0.907 | 25.0 | 38.8 | 0.736 | 2.4 | 68.9 | 17.7 | 30.0 | 0.893 | 3.30 | 67.9 | Pauselli et al. (2019) |
| AP48R2 | AHe | 43.879 | 11.711 | 1.655 | 0.907 | 25.0 | 38.8 | 0.736 | 2.4 | 68.9 | 17.7 | 30.0 | 0.893 | 3.30 | 67.9 | Pauselli et al. (2019) |
| AP5 | AHe | 44.189 | 11.501 | 0.200 | 0.521 | 25.0 | 29 | 0.241 | 1.9 | 59.6 | 24.5 | 28.5 | 0.247 | 1.96 | 59.6 | della Vedova et al. (2001) |
| AP52 | AHe | 43.905 | 11.791 | 0.565 | 0.836 | 25.1 | 38.2 | 0.706 | 2.3 | 68.7 | 13.7 | 26.5 | 1.141 | 4.32 | 68.9 | Pauselli et al. (2019) |
| AP53 | AHe | 43.934 | 11.656 | 0.907 | 0.830 | 25.0 | 31.1 | 0.356 | 2.1 | 62.5 | 23.0 | 29.0 | 0.383 | 2.28 | 62.4 | Pauselli et al. (2019) |
| AP54 | AHe | 43.961 | 11.670 | 0.690 | 0.811 | 25.0 | 39.5 | 0.765 | 2.4 | 69.4 | 13.9 | 27.5 | 1.176 | 4.26 | 69.1 | Pauselli et al. (2019) |
| AP55 | AHe | 44.013 | 11.687 | 1.070 | 0.722 | 25.0 | 42.7 | 0.901 | 2.4 | 70.9 | 12.7 | 27.5 | 1.347 | 4.69 | 69.9 | Pauselli et al. (2019) |
| AP57 | AHe | 43.995 | 11.719 | 0.450 | 0.746 | 25.0 | 42.9 | 0.910 | 2.4 | 71.2 | 10.8 | 27.5 | 1.669 | 5.68 | 71.5 | Pauselli et al. (2019) |
| AP5B | AHe | 44.189 | 11.501 | 0.200 | 0.521 | 25.0 | 36.1 | 0.610 | 2.2 | 67.3 | 17.3 | 28.5 | 0.841 | 3.24 | 67.6 | della Vedova et al. (2001) |
| AP5C | AHe | 44.189 | 11.501 | 0.200 | 0.521 | 25.0 | 46.5 | 1.057 | 2.4 | 72.8 | 32.9 | 28.5 | 0.841 | 1.86 | 72.6 | della Vedova et al. (2001) |
| AP5D | AHe | 44.189 | 11.501 | 0.200 | 0.521 | 25.0 | 33.9 | 0.501 | 2.2 | 65.5 | 19.5 | 28.5 | 0.634 | 2.78 | 65.7 | della Vedova et al. (2001) |
| AP8 | AHe | 44.147 | 11.449 | 0.300 | 0.629 | 25.1 | 42.7 | 0.903 | 2.4 | 71.2 | 12.9 | 30.0 | 1.483 | 4.69 | 71.4 | della Vedova et al. (2001) |
| AP9 | AHe | 44.115 | 11.431 | 0.400 | 0.674 | 25.0 | 42.5 | 0.893 | 2.4 | 71.0 | 11.1 | 27.5 | 1.601 | 5.43 | 71.3 | Pauselli et al. (2019) |
| C1 | AHe | 44.113 | 11.002 | 0.500 | 0.765 | 25.0 | 32.2 | 0.415 | 2.1 | 63.9 | 33.0 | 40.0 | 0.312 | 1.62 | 63.6 | della Vedova et al. (2001) |
| C10 | AHe | 44.143 | 11.191 | 0.605 | 0.766 | 25.0 | 55.2 | 1.386 | 2.6 | 76.0 | NA | 35.0 | 0.312 | 0.49 | 73.3 | della Vedova et al. (2001) |
| C11 | AHe | 44.001 | 10.807 | 0.950 | 0.667 | 25.1 | 30.9 | 0.342 | 2.0 | 62.0 | 38.7 | 45.0 | 0.247 | 1.34 | 62.8 | della Vedova et al. (2001) |
| C13 | AHe | 44.021 | 10.864 | 0.700 | 0.747 | 25.0 | 32.8 | 0.443 | 2.2 | 64.3 | 39.7 | 47.5 | 0.295 | 1.37 | 64.6 | della Vedova et al. (2001) |
| C16 | AHe | 44.060 | 10.913 | 0.630 | 0.909 | 25.0 | 34.1 | 0.513 | 2.2 | 65.6 | 36.2 | 45.0 | 0.353 | 1.54 | 65.3 | della Vedova et al. (2001) |
| C17 | AHe | 44.068 | 10.919 | 0.625 | 0.921 | 25.0 | 34.2 | 0.517 | 2.3 | 65.8 | 36.2 | 45.0 | 0.355 | 1.54 | 65.3 | della Vedova et al. (2001) |
| C2 | AHe | 44.004 | 11.012 | 0.830 | 0.660 | 25.0 | 34.4 | 0.522 | 2.2 | 65.7 | 35.1 | 45.0 | 0.406 | 1.57 | 66.1 | della Vedova et al. (2001) |
| C22 | AHe | 44.041 | 10.932 | 0.850 | 0.821 | 25.1 | 33 | 0.454 | 2.2 | 64.6 | 36.8 | 45.0 | 0.330 | 1.49 | 64.8 | della Vedova et al. (2001) |
| C23 | AHe | 44.021 | 10.929 | 0.884 | 0.719 | 25.0 | 32.9 | 0.453 | 2.2 | 64.6 | 36.7 | 45.0 | 0.338 | 1.49 | 65.0 | della Vedova et al. (2001) |
| C29 | AHe | 44.731 | 9.386 | 0.320 | 0.775 | 25.0 | 32.9 | 0.454 | 2.2 | 64.7 | 27.2 | 35.0 | 0.412 | 2.00 | 64.5 | della Vedova et al. (2001) |
| C3 | AHe | 44.014 | 11.025 | 0.780 | 0.706 | 25.0 | 33 | 0.452 | 2.2 | 64.4 | 36.7 | 45.0 | 0.331 | 1.48 | 64.8 | della Vedova et al. (2001) |
| C34 | AHe | 44.417 | 9.949 | 0.510 | 0.855 | 25.1 | 34.1 | 0.508 | 2.2 | 65.7 | 31.2 | 40.0 | 0.405 | 1.78 | 65.3 | della Vedova et al. (2001) |
| C37 | AHe | 44.246 | 10.683 | 0.641 | 1.064 | 25.0 | 39.3 | 0.755 | 2.4 | 69.3 | 31.2 | 45.0 | 0.610 | 1.93 | 69.0 | della Vedova et al. (2001) |
| C4 | AHe | 44.028 | 11.038 | 0.680 | 0.753 | 25.1 | 39.7 | 0.772 | 2.4 | 69.5 | 28.7 | 43.5 | 0.696 | 2.06 | 69.6 | della Vedova et al. (2001) |
| C40 | AHe | 44.223 | 10.758 | 1.250 | 0.980 | 25.1 | 45.5 | 1.017 | 2.5 | 72.3 | 25.4 | 46.0 | 1.007 | 2.48 | 72.2 | della Vedova et al. (2001) |
| C5 | AHe | 44.049 | 11.044 | 0.610 | 0.798 | 25.0 | 32.6 | 0.434 | 2.2 | 64.2 | 35.0 | 42.5 | 0.313 | 1.54 | 64.0 | della Vedova et al. (2001) |
| C52A | AHe | 44.013 | 11.503 | 0.360 | 0.661 | 25.1 | 32.4 | 0.419 | 2.1 | 63.9 | 21.1 | 28.5 | 0.497 | 2.53 | 64.1 | Pauselli et al. (2019) |
| C6 | AHe | 44.095 | 11.044 | 0.650 | 0.779 | 25.0 | 39.5 | 0.762 | 2.4 | 69.3 | 25.5 | 40.0 | 0.751 | 2.32 | 69.3 | della Vedova et al. (2001) |
| C7 | AHe | 44.111 | 11.039 | 0.625 | 0.755 | 25.0 | 38.5 | 0.725 | 2.3 | 68.9 | 26.4 | 40.0 | 0.693 | 2.22 | 68.8 | della Vedova et al. (2001) |
| C8 | AHe | 44.115 | 11.204 | 0.700 | 0.757 | 25.0 | 40.3 | 0.802 | 2.4 | 69.8 | 22.4 | 37.5 | 0.870 | 2.66 | 69.8 | della Vedova et al. (2001) |
| C9 | AHe | 44.106 | 11.204 | 1.000 | 0.738 | 25.0 | 46.3 | 1.051 | 2.5 | 72.7 | 18.0 | 37.5 | 1.275 | 3.44 | 72.4 | della Vedova et al. (2001) |
| CIM1 | AHe | 44.194 | 10.699 | 2.165 | 1.174 | 25.0 | 40.2 | 0.800 | 2.5 | 69.5 | 28.9 | 45.0 | 0.745 | 2.14 | 70.0 | della Vedova et al. (2001) |
| CIM1R1 | AHe | 44.194 | 10.699 | 2.165 | 1.174 | 25.0 | 40.2 | 0.802 | 2.5 | 69.5 | 28.9 | 45.0 | 0.748 | 2.14 | 70.1 | della Vedova et al. (2001) |
| CIM2 | AHe | 44.194 | 10.704 | 2.045 | 1.165 | 25.1 | 42.8 | 0.908 | 2.5 | 71.0 | 26.8 | 45.0 | 0.879 | 2.35 | 71.2 | della Vedova et al. (2001) |
| CIM3 | AHe | 44.196 | 10.692 | 1.950 | 1.184 | 25.1 | 42.7 | 0.903 | 2.5 | 70.9 | 26.9 | 45.0 | 0.871 | 2.34 | 71.1 | della Vedova et al. (2001) |
| CIM3R1 | AHe | 44.196 | 10.692 | 1.950 | 1.184 | 25.0 | 38.1 | 0.705 | 2.4 | 68.3 | 30.8 | 45.0 | 0.629 | 1.97 | 68.7 | della Vedova et al. (2001) |
| CIM4 | AHe | 44.200 | 10.684 | 1.830 | 1.196 | 25.0 | 40.4 | 0.810 | 2.5 | 69.8 | 28.8 | 45.0 | 0.749 | 2.15 | 70.0 | della Vedova et al. (2001) |
| CIM4R1 | AHe | 44.200 | 10.684 | 1.830 | 1.196 | 25.0 | 39.7 | 0.779 | 2.5 | 69.3 | 29.5 | 45.0 | 0.710 | 2.09 | 69.8 | della Vedova et al. (2001) |
| CIM5 | AHe | 44.202 | 10.677 | 1.750 | 1.208 | 25.0 | 39.2 | 0.751 | 2.4 | 69.0 | 30.0 | 45.0 | 0.675 | 2.04 | 69.2 | della Vedova et al. (2001) |
| CIM5A | AHe | 44.202 | 10.677 | 1.750 | 1.208 | 25.0 | 40.6 | 0.818 | 2.5 | 70.0 | 28.7 | 45.0 | 0.756 | 2.16 | 70.1 | della Vedova et al. (2001) |
| CIM5R1 | AHe | 44.202 | 10.677 | 1.750 | 1.208 | 25.1 | 41.7 | 0.857 | 2.5 | 70.4 | 27.9 | 45.0 | 0.808 | 2.25 | 70.7 | della Vedova et al. (2001) |
| CIM6 | AHe | 44.201 | 10.666 | 1.660 | 1.230 | 25.0 | 40.2 | 0.800 | 2.5 | 69.7 | 29.2 | 45.0 | 0.730 | 2.13 | 70.0 | della Vedova et al. (2001) |
| CIM6R1 | AHe | 44.201 | 10.666 | 1.660 | 1.230 | 25.0 | 39.9 | 0.785 | 2.5 | 69.4 | 29.4 | 45.0 | 0.712 | 2.10 | 69.8 | della Vedova et al. (2001) |
| VALD10a | AHe | 43.653 | 11.640 | 1.400 | 0.836 | 25.0 | 36.9 | 0.651 | 2.3 | 67.7 | 17.7 | 28.5 | 0.804 | 3.22 | 67.1 | Pauselli et al. (2019) |
| VALD1a | AHe | 43.594 | 11.603 | 0.497 | 0.471 | 25.0 | 29.3 | 0.259 | 1.9 | 59.9 | 24.7 | 29.0 | 0.262 | 1.95 | 59.9 | Pauselli et al. (2019) |
| VALD2a | AHe | 43.612 | 11.645 | 0.500 | 0.685 | 25.0 | 32 | 0.405 | 2.1 | 63.7 | 23.0 | 30.0 | 0.438 | 2.31 | 63.7 | Pauselli et al. (2019) |
| VALD2R1 | AHe | 43.612 | 11.645 | 0.500 | 0.685 | 25.0 | 32 | 0.408 | 2.1 | 63.7 | 23.0 | 30.0 | 0.441 | 2.31 | 63.8 | Pauselli et al. (2019) |
| VALD4a1 | AHe | 43.620 | 11.648 | 0.740 | 0.730 | 25.0 | 33.8 | 0.497 | 2.2 | 65.4 | 20.8 | 29.5 | 0.575 | 2.63 | 65.2 | Pauselli et al. (2019) |
| VALD4a2 | AHe | 43.620 | 11.648 | 0.740 | 0.730 | 25.0 | 31.2 | 0.359 | 2.1 | 62.5 | 23.3 | 29.5 | 0.381 | 2.23 | 62.4 | Pauselli et al. (2019) |
| VALD4R1 | AHe | 43.620 | 11.648 | 0.740 | 0.730 | 25.1 | 32.8 | 0.442 | 2.1 | 64.3 | 21.8 | 29.5 | 0.495 | 2.46 | 64.2 | Pauselli et al. (2019) |
| VALD5a | AHe | 43.621 | 11.656 | 0.850 | 0.747 | 25.0 | 33.3 | 0.470 | 2.2 | 64.8 | 21.9 | 30.0 | 0.523 | 2.48 | 64.7 | Pauselli et al. (2019) |
| VALD5R1 | AHe | 43.621 | 11.656 | 0.850 | 0.747 | 25.0 | 33.4 | 0.474 | 2.2 | 64.8 | 21.8 | 30.0 | 0.529 | 2.50 | 64.7 | Pauselli et al. (2019) |
| VALD6a | AHe | 43.620 | 11.659 | 0.880 | 0.746 | 25.0 | 33.1 | 0.465 | 2.2 | 64.7 | 22.0 | 30.0 | 0.513 | 2.46 | 64.5 | Pauselli et al. (2019) |
| VALD6R1 | AHe | 43.620 | 11.659 | 0.880 | 0.746 | 25.0 | 33.6 | 0.490 | 2.2 | 65.2 | 21.5 | 30.0 | 0.548 | 2.54 | 64.9 | Pauselli et al. (2019) |
| VALD7a | AHe | 43.604 | 11.651 | 1.100 | 0.658 | 25.0 | 35.3 | 0.575 | 2.2 | 66.5 | 21.0 | 31.0 | 0.642 | 2.63 | 66.1 | Pauselli et al. (2019) |
| VALD8R1 | AHe | 43.626 | 11.684 | 1.200 | 0.782 | 25.0 | 34.5 | 0.528 | 2.2 | 65.6 | 21.4 | 30.5 | 0.587 | 2.58 | 65.4 | Pauselli et al. (2019) |
| VALD8R2 | AHe | 43.626 | 11.684 | 1.200 | 0.782 | 25.0 | 36.2 | 0.615 | 2.3 | 67.1 | 19.9 | 30.5 | 0.712 | 2.84 | 66.8 | Pauselli et al. (2019) |

**Kinetic Parameters for (U-Th)/He apatite** *(from Farley, 2000 and Reiners and Brandon, 2006)*

$E_a$ = 138 kJ mol$^{-1}$ (activation energy)

$a_s$ = 45 μm (effective spherical radius for the diffusion domain)

$\Omega$ = 1.36 x 10$^6$ (frequency factor calculated as 55$D_0a^{-2}$)

$t_{c,10}$ = 62.7°C (effective closure temperature for 10 Myr$^{-1}$ cooling rates and specified $a_s$ value)

**Table 9 Erosion rates and parameters for AFT bedrock samples.**





| ID | Method | Latitude | Longitude | Sample Elevation n (km) | Mean Elevation (km) | Imposed $G_0$ = 25 (°C/km) | | | | | $G_f$ calculated from heat flow measurements | | | | | Heat Flow Measurement Source |
|---|---|---|---|---|---|---|---|---|---|---|---|---|---|---|---|---|
| | | | | | | Inital Geothermal Gradient (°C/km) | Final Geothermal Gradient (°C/km) | Erosion Rate (km/My) | Closure Depth (km) | Closure Temperature (°C) | Inital Geothermal Gradient (°C/km) | Final Geothermal Gradient (°C/km) | Erosion Rate (km/My) | Closure Depth (km) | Closure Temperature (°C) | |
| 1927 | AFT | 44.064 | 11.597 | 0.350 | 0.667 | 25.1 | 32.2 | 0.410 | 4.1 | 113.9 | 20.8 | 28.0 | 0.494 | 4.9 | 113.4 | Pauselli et al. (2019) |
| 1929 | AFT | 44.037 | 11.504 | 0.700 | 0.619 | 25.0 | 35.3 | 0.574 | 4.3 | 118.8 | 18.3 | 28.5 | 0.743 | 5.8 | 117.9 | Pauselli et al. (2019) |
| 1930 | AFT | 44.069 | 11.492 | 1.150 | 0.630 | 25.0 | 40.4 | 0.805 | 4.5 | 123.2 | 14.8 | 29.0 | 1.155 | 7.5 | 121.8 | Pauselli et al. (2019) |
| 03GB07 | AFT | 44.124 | 10.059 | 0.675 | 0.356 | 25.0 | 34.1 | 0.508 | 4.2 | 116.3 | 30.8 | 40.0 | 0.429 | 3.4 | 117.2 | della Vedova et al. (2001) |
| 03RE20 | AFT | 44.098 | 10.326 | 1.055 | 0.760 | 25.0 | 34.6 | 0.536 | 4.3 | 117.1 | 30.2 | 40.0 | 0.459 | 3.6 | 117.7 | della Vedova et al. (2001) |
| 050320-1a | AFT | 44.263 | 10.664 | 1.112 | 0.976 | 25.0 | 34.5 | 0.532 | 4.3 | 117.4 | 32.9 | 42.5 | 0.422 | 3.3 | 118.1 | della Vedova et al. (2001) |
| 050320-1b | AFT | 44.263 | 10.664 | 1.112 | 0.976 | 25.0 | 38.3 | 0.712 | 4.5 | 121.7 | 29.1 | 42.5 | 0.634 | 3.9 | 122.0 | della Vedova et al. (2001) |
| AP 10 | AFT | 44.121 | 11.396 | 0.500 | 0.685 | 25.0 | 39.8 | 0.780 | 4.5 | 123.2 | 13.2 | 27.5 | 1.277 | 8.5 | 121.9 | Pauselli et al. (2019) |
| Ap 15 | AFT | 43.819 | 11.952 | 0.700 | 0.805 | 25.0 | 32.1 | 0.411 | 4.1 | 112.4 | 22.9 | 30.0 | 0.446 | 4.4 | 111.8 | Pauselli et al. (2019) |
| AP 34 | AFT | 44.097 | 11.315 | 0.600 | 0.653 | 25.0 | 35.7 | 0.592 | 4.3 | 119.3 | 17.4 | 28.0 | 0.802 | 6.2 | 118.3 | Pauselli et al. (2019) |
| AP 43 | AFT | 43.818 | 11.733 | 0.515 | 0.805 | 25.0 | 36 | 0.608 | 4.4 | 119.9 | 18.4 | 29.5 | 0.793 | 5.9 | 119.2 | Pauselli et al. (2019) |
| AP 44 | AFT | 43.824 | 11.746 | 0.725 | 0.807 | 25.0 | 38.6 | 0.725 | 4.5 | 122.0 | 16.7 | 30.0 | 0.996 | 6.7 | 121.3 | Pauselli et al. (2019) |
| AP 45 | AFT | 43.828 | 11.749 | 0.940 | 0.809 | 25.0 | 38.9 | 0.741 | 4.5 | 122.3 | 16.1 | 29.5 | 1.031 | 6.9 | 121.0 | Pauselli et al. (2019) |
| AP 47 | AFT | 43.864 | 11.739 | 1.365 | 0.816 | 25.0 | 40.6 | 0.816 | 4.5 | 123.1 | 14.6 | 29.0 | 1.176 | 7.6 | 121.4 | Pauselli et al. (2019) |
| AP 52 | AFT | 43.905 | 11.723 | 0.565 | 0.799 | 25.1 | 36.7 | 0.633 | 4.4 | 120.1 | 18.4 | 30.0 | 0.824 | 6.0 | 119.7 | Pauselli et al. (2019) |
| AP 53 | AFT | 43.934 | 11.656 | 0.907 | 0.814 | 25.0 | 33.1 | 0.458 | 4.2 | 114.5 | 21.0 | 29.0 | 0.536 | 4.9 | 113.9 | Pauselli et al. (2019) |
| AP 54 | AFT | 43.961 | 11.670 | 0.690 | 0.747 | 25.0 | 36.3 | 0.617 | 4.4 | 119.8 | 17.4 | 28.5 | 0.835 | 6.3 | 119.1 | Pauselli et al. (2019) |
| AP 55 | AFT | 44.013 | 11.687 | 1.070 | 0.672 | 25.0 | 34.4 | 0.525 | 4.2 | 116.6 | 19.3 | 28.5 | 0.648 | 5.4 | 115.5 | Pauselli et al. (2019) |
| AP 56 | AFT | 43.983 | 11.686 | 0.500 | 0.723 | 25.0 | 39 | 0.747 | 4.5 | 122.4 | 14.7 | 28.5 | 1.134 | 7.6 | 121.8 | Pauselli et al. (2019) |
| AP 57 | AFT | 43.995 | 11.719 | 0.450 | 0.688 | 25.0 | 40.6 | 0.814 | 4.5 | 123.5 | 12.4 | 27.5 | 1.384 | 9.0 | 122.5 | Pauselli et al. (2019) |
| AP 9 | AFT | 44.115 | 11.431 | 0.400 | 0.673 | 25.0 | 39.4 | 0.761 | 4.5 | 122.8 | 13.9 | 28.0 | 1.211 | 8.0 | 122.0 | Pauselli et al. (2019) |
| BOR2 | AFT | 44.435 | 9.425 | 0.452 | 0.779 | 25.0 | 33 | 0.457 | 4.2 | 115.9 | 32.2 | 40.0 | 0.355 | 3.3 | 116.2 | della Vedova et al. (2001) |
| BORl | AFT | 44.435 | 9.425 | 0.450 | 0.779 | 25.0 | 34.2 | 0.519 | 4.3 | 117.8 | 30.9 | 40.0 | 0.422 | 3.5 | 117.8 | della Vedova et al. (2001) |
| BT1 | AFT | 44.085 | 9.787 | 0.475 | 0.104 | 25.0 | 39.8 | 0.780 | 4.4 | 123.0 | 25.2 | 40.0 | 0.775 | 4.3 | 122.6 | della Vedova et al. (2001) |
| BT2 | AFT | 44.085 | 9.787 | 0.525 | 0.104 | 25.0 | 39.6 | 0.770 | 4.4 | 122.6 | 25.4 | 40.0 | 0.762 | 4.3 | 122.5 | della Vedova et al. (2001) |
| C1 | AFT | 44.113 | 11.002 | 0.500 | 0.789 | 25.0 | 34 | 0.505 | 4.3 | 117.1 | 33.7 | 42.5 | 0.379 | 3.2 | 117.5 | della Vedova et al. (2001) |
| C10 | AFT | 44.143 | 11.191 | 0.605 | 0.683 | 25.0 | 40.3 | 0.802 | 4.5 | 123.3 | 19.9 | 35.0 | 0.954 | 5.6 | 122.9 | della Vedova et al. (2001) |
| C11 | AFT | 44.001 | 10.807 | 0.950 | 0.644 | 25.1 | 32.2 | 0.414 | 3.8 | 106.7 | 37.4 | 45.0 | 0.299 | 2.7 | 110.7 | della Vedova et al. (2001) |
| C13 | AFT | 44.021 | 10.864 | 0.700 | 0.731 | 25.0 | 34.6 | 0.533 | 4.3 | 117.6 | 38.0 | 47.5 | 0.369 | 2.8 | 118.6 | della Vedova et al. (2001) |
| C16 | AFT | 44.060 | 10.913 | 0.630 | 0.807 | 25.0 | 45.3 | 1.014 | 4.7 | 126.6 | 24.7 | 45.0 | 1.022 | 4.7 | 126.5 | della Vedova et al. (2001) |
| C17 | AFT | 44.068 | 10.919 | 0.625 | 0.818 | 25.0 | 37.2 | 0.661 | 4.4 | 121.0 | 32.9 | 45.0 | 0.522 | 3.4 | 121.2 | della Vedova et al. (2001) |
| C2 | AFT | 44.004 | 11.012 | 0.830 | 0.548 | 25.0 | 35.8 | 0.593 | 4.3 | 118.9 | 33.9 | 45.0 | 0.466 | 3.2 | 119.5 | della Vedova et al. (2001) |
| C22 | AFT | 44.041 | 10.932 | 0.850 | 0.747 | 25.0 | 45.3 | 1.012 | 4.6 | 126.7 | 24.7 | 45.0 | 1.020 | 4.7 | 126.4 | della Vedova et al. (2001) |
| C23 | AFT | 44.021 | 10.929 | 0.884 | 0.689 | 25.1 | 38 | 0.692 | 4.4 | 121.0 | 32.7 | 46.0 | 0.563 | 3.4 | 121.6 | della Vedova et al. (2001) |
| C29 | AFT | 44.731 | 9.386 | 0.320 | 0.804 | 25.0 | 36.5 | 0.630 | 4.4 | 120.4 | 23.4 | 35.0 | 0.668 | 4.7 | 120.2 | della Vedova et al. (2001) |
| C3 | AFT | 44.014 | 11.025 | 0.780 | 0.580 | 25.1 | 38.4 | 0.711 | 4.4 | 121.4 | 30.0 | 43.5 | 0.621 | 3.7 | 122.2 | della Vedova et al. (2001) |
| C34 | AFT | 44.417 | 9.949 | 0.510 | 0.820 | 25.1 | 32.2 | 0.410 | 4.1 | 113.8 | 33.1 | 40.0 | 0.309 | 3.2 | 114.4 | della Vedova et al. (2001) |
| C37 | AFT | 44.246 | 10.683 | 0.641 | 1.016 | 25.0 | 44.8 | 0.991 | 4.7 | 126.6 | 23.7 | 43.5 | 1.032 | 4.9 | 126.3 | della Vedova et al. (2001) |
| C38 | AFT | 44.223 | 10.777 | 0.980 | 0.950 | 25.0 | 44.5 | 0.977 | 4.7 | 125.9 | 26.0 | 45.5 | 0.952 | 4.5 | 126.2 | della Vedova et al. (2001) |
| C4 | AFT | 44.028 | 11.038 | 0.680 | 0.620 | 25.0 | 43.4 | 0.933 | 4.6 | 125.5 | 25.1 | 43.5 | 0.930 | 4.6 | 125.5 | della Vedova et al. (2001) |
| C40 | AFT | 44.223 | 10.758 | 1.250 | 0.977 | 25.0 | 41.5 | 0.851 | 4.6 | 123.8 | 29.2 | 46.0 | 0.763 | 3.9 | 124.4 | della Vedova et al. (2001) |
| C5 | AFT | 44.049 | 11.044 | 0.610 | 0.679 | 25.0 | 36 | 0.606 | 4.4 | 119.6 | 32.4 | 43.5 | 0.484 | 3.4 | 120.0 | della Vedova et al. (2001) |
| C52 | AFT | 44.013 | 11.503 | 0.360 | 0.594 | 25.0 | 35.1 | 0.562 | 4.3 | 118.8 | 18.4 | 28.5 | 0.739 | 5.8 | 118.3 | Pauselli et al. (2019) |
| C6 | AFT | 44.095 | 11.044 | 0.650 | 0.735 | 25.0 | 37.3 | 0.669 | 4.4 | 120.9 | 27.6 | 40.0 | 0.614 | 4.0 | 121.0 | della Vedova et al. (2001) |
| C7 | AFT | 44.111 | 11.039 | 0.625 | 0.738 | 25.0 | 36.4 | 0.626 | 4.4 | 120.1 | 28.5 | 40.0 | 0.558 | 3.8 | 120.2 | della Vedova et al. (2001) |
| C8 | AFT | 44.115 | 11.207 | 0.700 | 0.680 | 25.0 | 35.5 | 0.580 | 4.3 | 118.8 | 27.0 | 37.5 | 0.544 | 4.0 | 119.1 | della Vedova et al. (2001) |
| C9 | AFT | 44.106 | 11.204 | 1.000 | 0.673 | 25.0 | 36.8 | 0.512 | 4.0 | 117.7 | 27.7 | 37.5 | 0.502 | 3.9 | 117.9 | della Vedova et al. (2001) |
| CAS1 | AFT | 44.206 | 10.446 | 1.300 | 1.061 | 25.0 | 33.2 | 0.466 | 4.2 | 113.9 | 31.7 | 40.0 | 0.380 | 3.4 | 115.0 | della Vedova et al. (2001) |
| CAS2 | AFT | 44.174 | 10.424 | 0.965 | 0.980 | 25.1 | 32.4 | 0.424 | 4.1 | 112.6 | 32.7 | 40.0 | 0.331 | 3.2 | 113.7 | della Vedova et al. (2001) |
| CAST2 | AFT | 44.105 | 10.415 | 0.270 | 0.811 | 25.0 | 31.4 | 0.372 | 4.1 | 112.2 | 34.0 | 40.0 | 0.264 | 3.0 | 113.0 | della Vedova et al. (2001) |
| CAST3 | AFT | 44.105 | 10.415 | 0.240 | 0.811 | 25.1 | 31.9 | 0.395 | 4.1 | 113.9 | 33.5 | 40.0 | 0.285 | 3.1 | 114.0 | della Vedova et al. (2001) |
| CB3 | AFT | 44.051 | 9.830 | 0.000 | 0.079 | 25.0 | 32.8 | 0.443 | 4.0 | 115.1 | 32.3 | 40.0 | 0.350 | 3.2 | 115.7 | della Vedova et al. (2001) |
| CB4 | AFT | 44.051 | 9.830 | 0.000 | 0.079 | 25.1 | 32.5 | 0.428 | 4.0 | 114.5 | 32.6 | 40.0 | 0.334 | 3.1 | 115.0 | della Vedova et al. (2001) |
| CB5 | AFT | 44.051 | 9.830 | 0.000 | 0.079 | 24.9 | 33.7 | 0.493 | 4.1 | 116.7 | 31.3 | 40.0 | 0.402 | 3.3 | 117.3 | della Vedova et al. (2001) |
| CH1 | AFT | 43.601 | 11.411 | 0.303 | 0.337 | 25.0 | 36.1 | 0.611 | 4.3 | 119.7 | 23.9 | 35.0 | 0.634 | 4.5 | 119.8 | Pauselli et al. (2019) |
| CH2 | AFT | 43.541 | 11.430 | 0.504 | 0.379 | 25.0 | 35.8 | 0.596 | 4.3 | 119.2 | 29.1 | 40.0 | 0.526 | 3.7 | 119.7 | Pauselli et al. (2019) |
| CH3 | AFT | 43.565 | 11.382 | 0.722 | 0.373 | 25.0 | 35.3 | 0.571 | 4.2 | 118.2 | 28.1 | 38.5 | 0.521 | 3.8 | 118.7 | Pauselli et al. (2019) |
| CH4 | AFT | 43.562 | 11.380 | 0.857 | 0.376 | 25.1 | 35.1 | 0.556 | 4.2 | 117.7 | 28.4 | 38.5 | 0.506 | 3.7 | 118.3 | Pauselli et al. (2019) |
| CIM1 | AFT | 44.194 | 10.699 | 2.165 | 1.121 | 25.0 | 36.7 | 0.637 | 4.4 | 118.9 | 32.5 | 45.0 | 0.533 | 3.4 | 119.8 | della Vedova et al. (2001) |
| CIM2 | AFT | 44.194 | 10.704 | 2.045 | 1.116 | 25.0 | 36 | 0.603 | 4.4 | 118.0 | 33.4 | 45.0 | 0.494 | 3.3 | 119.3 | della Vedova et al. (2001) |
| CIM3 | AFT | 44.196 | 10.692 | 1.950 | 1.124 | 25.0 | 36.2 | 0.616 | 4.4 | 118.5 | 33.2 | 45.0 | 0.504 | 3.4 | 119.8 | della Vedova et al. (2001) |
| CIM4 | AFT | 44.200 | 10.684 | 1.830 | 1.128 | 25.0 | 37 | 0.652 | 4.4 | 119.7 | 30.1 | 42.5 | 0.571 | 3.7 | 120.2 | della Vedova et al. (2001) |
| CIM5 | AFT | 44.202 | 10.677 | 1.750 | 1.134 | 25.1 | 36 | 0.603 | 4.4 | 118.8 | 33.0 | 44.5 | 0.490 | 3.4 | 119.4 | della Vedova et al. (2001) |
| CIM6 | AFT | 44.201 | 10.666 | 1.660 | 1.149 | 25.1 | 36.6 | 0.632 | 4.4 | 119.6 | 32.0 | 44.0 | 0.526 | 3.5 | 120.2 | della Vedova et al. (2001) |
| GOM2 | AFT | 44.125 | 10.642 | 1.850 | 1.061 | 25.0 | 33.9 | 0.498 | 4.1 | 112.0 | 34.6 | 44.0 | 0.392 | 3.1 | 114.6 | della Vedova et al. (2001) |
| GOM3 | AFT | 44.134 | 10.656 | 1.300 | 1.096 | 25.0 | 36.3 | 0.618 | 4.4 | 119.4 | 32.5 | 44.0 | 0.501 | 3.4 | 120.0 | della Vedova et al. (2001) |
| MG3 | AFT | 44.235 | 9.472 | 0.000 | 0.193 | 25.0 | 31.2 | 0.362 | 3.8 | 108.0 | 33.9 | 40.0 | 0.269 | 2.9 | 110.1 | della Vedova et al. (2001) |
| MS1 | AFT | 44.134 | 9.638 | 0.000 | 0.136 | 25.0 | 31.3 | 0.367 | 3.8 | 108.1 | 33.8 | 40.0 | 0.276 | 2.9 | 110.3 | della Vedova et al. (2001) |
| MS2 | AFT | 44.134 | 9.638 | 0.000 | 0.136 | 25.0 | 33.1 | 0.462 | 4.1 | 115.9 | 32.0 | 40.0 | 0.367 | 3.2 | 116.4 | della Vedova et al. (2001) |
| MS4 | AFT | 44.134 | 9.638 | 0.000 | 0.136 | 25.1 | 32.8 | 0.442 | 4.1 | 115.2 | 32.3 | 40.0 | 0.347 | 3.2 | 115.6 | della Vedova et al. (2001) |
| MS5 | AFT | 44.134 | 9.638 | 0.000 | 0.136 | 25.1 | 32.2 | 0.413 | 4.0 | 113.7 | 33.0 | 40.0 | 0.317 | 3.1 | 114.5 | della Vedova et al. (2001) |





| | | | | | | | | | | | | | | | | |
|---|---|---|---|---|---|---|---|---|---|---|---|---|---|---|---|---|
| PR 11 | AFT | 44.463 | 9.930 | 0.880 | 0.808 | 25.0 | 32.9 | 0.452 | 4.2 | 114.4 | 32.0 | 40.0 | 0.361 | 3.3 | 115.1 | della Vedova et al. (2001) |
| PR 12 | AFT | 44.353 | 9.776 | 0.668 | 0.700 | 25.0 | 32.6 | 0.438 | 4.1 | 114.0 | 32.4 | 40.0 | 0.344 | 3.2 | 115.0 | della Vedova et al. (2001) |
| PR 15 | AFT | 44.472 | 9.966 | 1.085 | 0.837 | 25.1 | 34.8 | 0.543 | 4.3 | 117.7 | 30.1 | 40.0 | 0.466 | 3.6 | 118.0 | della Vedova et al. (2001) |
| PR 17 | AFT | 44.379 | 10.196 | 0.860 | 1.001 | 25.0 | 39.6 | 0.770 | 4.5 | 122.9 | 32.8 | 47.5 | 0.617 | 3.5 | 123.1 | della Vedova et al. (2001) |
| PR 18 | AFT | 44.380 | 10.194 | 0.780 | 1.001 | 25.0 | 37.9 | 0.693 | 4.5 | 121.5 | 34.6 | 47.5 | 0.522 | 3.3 | 121.5 | della Vedova et al. (2001) |
| PR 20 | AFT | 44.338 | 10.528 | 0.500 | 0.881 | 25.0 | 46.6 | 1.068 | 4.7 | 127.6 | 16.4 | 37.5 | 1.451 | 7.2 | 127.5 | della Vedova et al. (2001) |
| PR 22 | AFT | 44.331 | 10.564 | 1.082 | 0.851 | 25.0 | 49.7 | 1.186 | 4.8 | 129.0 | 14.9 | 37.5 | 1.634 | 7.9 | 127.9 | della Vedova et al. (2001) |
| PR 23.1 | AFT | 44.329 | 10.562 | 1.111 | 0.861 | 25.0 | 39.5 | 0.762 | 4.5 | 122.3 | 23.2 | 37.5 | 0.807 | 4.8 | 122.2 | della Vedova et al. (2001) |
| PR 25.1 | AFT | 44.320 | 9.995 | 0.248 | 0.639 | 25.0 | 33.3 | 0.471 | 4.2 | 116.5 | 34.5 | 42.5 | 0.339 | 3.1 | 116.7 | della Vedova et al. (2001) |
| PR 26 | AFT | 44.463 | 9.602 | 1.135 | 0.883 | 25.0 | 37.8 | 0.688 | 4.4 | 120.9 | 27.1 | 40.0 | 0.647 | 4.1 | 121.3 | della Vedova et al. (2001) |
| PR 27 | AFT | 44.525 | 9.824 | 0.618 | 0.726 | 25.0 | 37.9 | 0.696 | 4.4 | 121.6 | 24.6 | 37.5 | 0.705 | 4.5 | 121.3 | della Vedova et al. (2001) |
| PR 28.1 | AFT | 44.522 | 9.931 | 0.710 | 0.776 | 25.0 | 43.2 | 0.930 | 4.6 | 125.4 | 17.4 | 35.0 | 1.208 | 6.6 | 125.1 | della Vedova et al. (2001) |
| PR 28.2 | AFT | 44.522 | 9.931 | 0.702 | 0.776 | 25.0 | 39.8 | 0.776 | 4.5 | 122.7 | 20.5 | 35.0 | 0.908 | 5.5 | 122.8 | della Vedova et al. (2001) |
| PR 3 | AFT | 44.446 | 9.943 | 0.600 | 0.817 | 25.0 | 34.9 | 0.548 | 4.3 | 118.1 | 30.2 | 40.0 | 0.461 | 3.6 | 118.6 | della Vedova et al. (2001) |
| PR 5 | AFT | 44.456 | 9.804 | 0.718 | 0.777 | 25.0 | 33.4 | 0.476 | 4.2 | 115.9 | 29.2 | 37.5 | 0.414 | 3.6 | 116.5 | della Vedova et al. (2001) |
| PR 6.1 | AFT | 44.456 | 9.783 | 0.600 | 0.789 | 25.1 | 38.7 | 0.728 | 4.5 | 122.2 | 23.8 | 37.5 | 0.757 | 4.7 | 121.9 | della Vedova et al. (2001) |
| PR 7 | AFT | 44.550 | 9.940 | 0.301 | 0.748 | 25.0 | 36.2 | 0.615 | 4.4 | 120.2 | 23.7 | 35.0 | 0.645 | 4.6 | 120.0 | della Vedova et al. (2001) |
| RAM1 | AFT | 44.434 | 9.311 | 1.318 | 0.632 | 25.0 | 34.3 | 0.524 | 4.2 | 115.7 | 30.4 | 40.0 | 0.451 | 3.5 | 116.8 | della Vedova et al. (2001) |
| RAM3 | AFT | 44.427 | 9.312 | 1.075 | 0.606 | 25.1 | 36.4 | 0.620 | 4.3 | 119.4 | 28.5 | 40.0 | 0.563 | 3.8 | 119.8 | della Vedova et al. (2001) |
| RAM4 | AFT | 44.427 | 9.312 | 1.075 | 0.606 | 25.0 | 35 | 0.555 | 4.3 | 117.6 | 29.9 | 40.0 | 0.484 | 3.6 | 118.4 | della Vedova et al. (2001) |
| RAM5 | AFT | 44.422 | 9.311 | 0.950 | 0.587 | 25.0 | 34.9 | 0.554 | 4.3 | 117.8 | 29.9 | 40.0 | 0.480 | 3.6 | 118.4 | della Vedova et al. (2001) |
| RAM6 | AFT | 44.422 | 9.311 | 0.948 | 0.587 | 25.0 | 36 | 0.606 | 4.3 | 119.2 | 28.9 | 40.0 | 0.542 | 3.8 | 119.7 | della Vedova et al. (2001) |
| ROM1 | AFT | 44.002 | 11.472 | 0.890 | 0.562 | 25.0 | 37.7 | 0.682 | 4.4 | 120.9 | 16.8 | 29.0 | 0.918 | 6.4 | 119.7 | Pauselli et al. (2019) |
| ROM2 | AFT | 44.001 | 11.472 | 0.760 | 0.560 | 25.0 | 38.3 | 0.712 | 4.4 | 121.7 | 16.2 | 29.0 | 0.986 | 6.7 | 120.4 | Pauselli et al. (2019) |
| ROM3 | AFT | 44.000 | 11.475 | 0.675 | 0.560 | 25.0 | 41.1 | 0.839 | 4.5 | 123.8 | 13.8 | 29.0 | 1.280 | 8.0 | 122.5 | Pauselli et al. (2019) |
| ROM4 | AFT | 43.998 | 11.477 | 0.575 | 0.563 | 25.0 | 40.3 | 0.802 | 4.5 | 123.3 | 14.4 | 29.0 | 1.208 | 7.7 | 122.4 | Pauselli et al. (2019) |
| ROM5 | AFT | 43.994 | 11.477 | 0.480 | 0.562 | 25.0 | 34.6 | 0.535 | 4.3 | 117.8 | 19.4 | 29.0 | 0.667 | 5.5 | 117.4 | Pauselli et al. (2019) |
| S 1 | AFT | 44.178 | 10.160 | 0.546 | 0.662 | 25.1 | 34.7 | 0.539 | 4.3 | 118.0 | 32.9 | 42.5 | 0.421 | 3.3 | 118.5 | della Vedova et al. (2001) |
| S 3 | AFT | 44.139 | 10.073 | 0.494 | 0.374 | 25.1 | 35.1 | 0.559 | 4.2 | 118.4 | 32.3 | 42.5 | 0.451 | 3.3 | 118.9 | della Vedova et al. (2001) |
| S 4 | AFT | 44.128 | 10.059 | 0.636 | 0.330 | 25.0 | 38.3 | 0.714 | 4.4 | 121.6 | 28.9 | 42.5 | 0.640 | 3.8 | 121.8 | della Vedova et al. (2001) |
| SC 2 | AFT | 44.081 | 10.083 | 0.204 | 0.342 | 25.1 | 34.6 | 0.534 | 4.2 | 118.1 | 30.5 | 40.0 | 0.448 | 3.5 | 118.4 | della Vedova et al. (2001) |
| SILL1 | AFT | 44.368 | 10.064 | 1.861 | 0.860 | 25.0 | 35 | 0.560 | 4.3 | 116.2 | 33.9 | 44.5 | 0.453 | 3.2 | 118.0 | della Vedova et al. (2001) |
| SILL10 | AFT | 44.334 | 10.050 | 0.730 | 0.754 | 25.1 | 34.3 | 0.516 | 4.3 | 117.1 | 35.3 | 44.5 | 0.381 | 3.0 | 118.0 | della Vedova et al. (2001) |
| SILL2 | AFT | 44.361 | 10.074 | 1.790 | 0.863 | 25.1 | 34 | 0.503 | 4.1 | 111.6 | 35.4 | 45.0 | 0.394 | 3.0 | 114.4 | della Vedova et al. (2001) |
| SILL3 | AFT | 44.357 | 10.073 | 1.600 | 0.853 | 25.0 | 37.1 | 0.659 | 4.4 | 120.0 | 32.3 | 45.0 | 0.551 | 3.4 | 121.0 | della Vedova et al. (2001) |
| SILL4 | AFT | 44.455 | 10.076 | 1.530 | 0.951 | 25.0 | 38.1 | 0.704 | 4.5 | 121.2 | 26.8 | 40.0 | 0.671 | 4.2 | 121.4 | della Vedova et al. (2001) |
| SILL5 | AFT | 44.354 | 10.071 | 1.420 | 0.843 | 25.0 | 36.3 | 0.620 | 4.4 | 119.2 | 33.2 | 45.0 | 0.503 | 3.3 | 120.2 | della Vedova et al. (2001) |
| SILL6 | AFT | 44.353 | 10.057 | 1.260 | 0.815 | 25.0 | 35.9 | 0.602 | 4.4 | 118.9 | 33.1 | 44.5 | 0.484 | 3.3 | 119.5 | della Vedova et al. (2001) |
| SILL7 | AFT | 44.338 | 10.058 | 1.130 | 0.781 | 25.0 | 36.7 | 0.639 | 4.4 | 120.0 | 32.4 | 44.5 | 0.524 | 3.4 | 120.4 | della Vedova et al. (2001) |
| SILL9 | AFT | 44.334 | 10.050 | 0.780 | 0.755 | 25.0 | 37.1 | 0.658 | 4.4 | 120.5 | 32.2 | 44.5 | 0.534 | 3.4 | 121.0 | della Vedova et al. (2001) |
| SM 3 | AFT | 44.164 | 10.129 | 0.250 | 0.558 | 25.0 | 31.9 | 0.398 | 4.1 | 113.1 | 35.9 | 42.5 | 0.274 | 2.9 | 114.0 | della Vedova et al. (2001) |
| SU 1 | AFT | 44.170 | 10.188 | 0.773 | 0.724 | 25.0 | 35.1 | 0.566 | 4.3 | 118.6 | 32.3 | 42.5 | 0.454 | 3.4 | 119.1 | della Vedova et al. (2001) |
| TCGA | AFT | 43.993 | 11.476 | 0.360 | 0.560 | 25.0 | 36.8 | 0.643 | 4.4 | 120.6 | 17.2 | 29.0 | 0.877 | 6.3 | 119.7 | Pauselli et al. (2019) |
| VALD1 | AFT | 43.594 | 11.603 | 0.497 | 0.484 | 25.0 | 37.6 | 0.681 | 4.4 | 121.3 | 16.7 | 29.0 | 0.937 | 6.5 | 120.1 | Pauselli et al. (2019) |
| VALD10 | AFT | 43.653 | 11.640 | 1.400 | 0.836 | 25.0 | 35.6 | 0.582 | 4.3 | 118.1 | 18.3 | 28.5 | 0.740 | 5.8 | 116.9 | Pauselli et al. (2019) |
| VALD11 | AFT | 43.663 | 11.641 | 1.450 | 0.644 | 25.0 | 38.1 | 0.701 | 4.4 | 120.8 | 15.9 | 28.0 | 0.962 | 6.8 | 119.3 | Pauselli et al. (2019) |
| VALD12 | AFT | 43.696 | 11.673 | 0.960 | 0.717 | 25.0 | 35.6 | 0.582 | 4.3 | 118.5 | 17.7 | 28.0 | 0.769 | 6.1 | 117.5 | Pauselli et al. (2019) |
| VALD2 | AFT | 43.612 | 11.645 | 0.500 | 0.550 | 25.0 | 33.8 | 0.495 | 4.2 | 116.6 | 21.2 | 30.0 | 0.573 | 4.9 | 116.1 | Pauselli et al. (2019) |
| VALD3 | AFT | 43.614 | 11.656 | 0.580 | 0.559 | 25.0 | 34.7 | 0.541 | 4.3 | 117.8 | 20.4 | 30.0 | 0.646 | 5.2 | 117.5 | Pauselli et al. (2019) |
| VALD4 | AFT | 43.620 | 11.648 | 0.740 | 0.567 | 25.0 | 37.4 | 0.673 | 4.4 | 120.9 | 17.4 | 29.5 | 0.888 | 6.2 | 119.9 | Pauselli et al. (2019) |
| VALD5 | AFT | 43.621 | 11.656 | 0.850 | 0.572 | 25.0 | 40.1 | 0.796 | 4.5 | 123.1 | 15.7 | 30.0 | 1.109 | 7.0 | 121.8 | Pauselli et al. (2019) |
| VALD6 | AFT | 43.620 | 11.659 | 0.880 | 0.571 | 25.0 | 35.4 | 0.575 | 4.3 | 118.3 | 19.8 | 30.0 | 0.692 | 5.4 | 117.6 | Pauselli et al. (2019) |
| VALD7 | AFT | 43.604 | 11.651 | 1.100 | 0.536 | 25.0 | 36 | 0.606 | 4.3 | 119.0 | 20.2 | 31.0 | 0.709 | 5.3 | 118.0 | Pauselli et al. (2019) |
| VALD8 | AFT | 43.626 | 11.684 | 1.200 | 0.585 | 25.0 | 33.9 | 0.503 | 4.2 | 115.2 | 21.7 | 30.5 | 0.562 | 4.7 | 114.2 | Pauselli et al. (2019) |
| VALD9 | AFT | 43.646 | 11.652 | 1.200 | 0.619 | 25.0 | 34.1 | 0.512 | 4.2 | 115.7 | 20.1 | 29.0 | 0.608 | 5.1 | 114.4 | Pauselli et al. (2019) |
| ZAT2 | AFT | 44.391 | 9.442 | 1.349 | 0.637 | 25.0 | 33.5 | 0.483 | 4.0 | 112.2 | 31.2 | 40.0 | 0.408 | 3.3 | 113.7 | della Vedova et al. (2001) |

**Kinetic Parameters for FT apatite** *(from Ketcham et al., 1999)*

$E_a$ = 147 kJ mol$^{-1}$ (activation energy)

$\Omega$ = 2.05 x 10$^6$ (measured directly from annealing experiments)

$t_{c,10}$ = 116°C (effective closure temperature for 10 Myr$^{-1}$ cooling rates)

**Table 10 Erosion rates and parameters for AFT detrital samples.**





| ID | Method | Latitude | Longitude | Sample Elevation (km) | Imposed G₀ = 25 (°C/km) Geothermal Gradient (°C/km) | Geothermal Gradient (°C/km) | Erosion Rate (km/My) | Closure Depth (km) | Closure Temperature (°C) | Gf calculated from heat flow measurements Initial Geothermal Gradient (°C/km) | Final Geothermal Gradient (°C/km) | Erosion Rate (km/My) | Closure Depth (km) | Closure Temperature (°C) | Heat Flow Measurement Source |
|---|---|---|---|---|---|---|---|---|---|---|---|---|---|---|---|
| Enza | AFT Detrital | 44.620 | 10.413 | 0.163 | 25.0 | 38.0 | 0.699 | 4.3 | 121.5 | 12.5 | 25.0 | 1.189 | 8.5 | 120.1 | della Vedova et al. (2001) |
| Nure | AFT Detrital | 44.872 | 9.647 | 0.208 | 25.0 | 39.7 | 0.775 | 4.4 | 123.0 | 12.1 | 26.0 | 1.327 | 8.9 | 121.2 | della Vedova et al. (2001) |
| Panaro | AFT Detrital | 44.477 | 11.027 | 0.099 | 25.0 | 34.2 | 0.517 | 4.2 | 117.3 | 13.3 | 22.5 | 0.882 | 7.6 | 115.2 | della Vedova et al. (2001) |
| Secchia | AFT Detrital | 44.532 | 10.758 | 0.119 | 25.0 | 34.7 | 0.543 | 4.2 | 118.1 | 9.9 | 19.5 | 1.173 | 10.3 | 114.9 | della Vedova et al. (2001) |
| Taro | AFT Detrital | 44.713 | 10.120 | 0.117 | 25.0 | 38.2 | 0.710 | 4.3 | 121.7 | 14.7 | 27.5 | 1.067 | 7.3 | 120.7 | della Vedova et al. (2001) |
| Trebbia | AFT Detrital | 44.901 | 9.584 | 0.140 | 25.0 | 39.9 | 0.789 | 4.4 | 123.2 | 11.0 | 25.0 | 1.439 | 9.8 | 121.3 | della Vedova et al. (2001) |
| Bisenzio | AFT Detrital | 43.928 | 11.126 | 0.102 | 25.0 | 36.6 | 0.637 | 4.3 | 120.4 | 19.5 | 31.0 | 0.779 | 5.5 | 119.8 | Pauselli et al. (2019) |
| Lima1 | AFT Detrital | 44.000 | 10.560 | 0.097 | 25.0 | 36.4 | 0.628 | 4.3 | 120.2 | 28.4 | 40.0 | 0.563 | 3.7 | 120.2 | della Vedova et al. (2001) |
| Lima2 | AFT Detrital | 44.091 | 10.760 | 0.544 | 25.0 | 35.5 | 0.582 | 4.3 | 118.8 | 34.4 | 45.0 | 0.442 | 3.1 | 119.6 | della Vedova et al. (2001) |
| Magra1 | AFT Detrital | 44.188 | 9.925 | 0.036 | 25.0 | 37.0 | 0.654 | 4.3 | 120.6 | 27.9 | 40.0 | 0.597 | 3.8 | 120.6 | della Vedova et al. (2001) |
| Magra2 | AFT Detrital | 44.387 | 9.887 | 0.251 | 25.0 | 36.9 | 0.651 | 4.3 | 120.5 | 27.9 | 40.0 | 0.592 | 3.9 | 120.6 | della Vedova et al. (2001) |
| Pescia | AFT Detrital | 43.929 | 10.693 | 0.105 | 25.1 | 33.1 | 0.459 | 4.1 | 115.6 | 34.4 | 42.5 | 0.343 | 3.0 | 116.2 | della Vedova et al. (2001) |
| Serchio | AFT Detrital | 44.192 | 10.306 | 0.525 | 25.0 | 33.7 | 0.494 | 4.2 | 116.6 | 33.8 | 42.5 | 0.377 | 3.1 | 117.2 | della Vedova et al. (2001) |
| Vara | AFT Detrital | 44.198 | 9.851 | 0.032 | 25.0 | 35.5 | 0.585 | 4.2 | 119.3 | 29.4 | 40.0 | 0.509 | 3.6 | 119.5 | della Vedova et al. (2001) |

**Kinetic Parameters for FT apatite** *(from Ketcham et al., 1999)*

$E_a$ = 147 kJ mol$^{-1}$ (activation energy)

$\Omega$ = 2.05 x 10$^6$ (measured directly from annealing experiments)

$t_{c,10}$ = 116°C (effective closure temperature for 10 Myr$^{-1}$ cooling rates)

**Table 11 Erosion rates and parameters for ZHe samples.**

| ID | Method | Latitude | Longitude | Sample Elevation (km) | Mean Elevation (km) | Imposed G₀ = 25 (°C/km) Initial Geothermal Gradient (°C/km) | Final Geothermal Gradient (°C/km) | Erosion Rate (km/My) | Closure Depth (km) | Closure Temperature (°C) | Gf calculated from heat flow measurements Initial Geothermal Gradient (°C/km) | Final Geothermal Gradient (°C/km) | Erosion Rate (km/My) | Closure Depth (km) | Closure Temperature (°C) | Heat Flow Measurement Source |
|---|---|---|---|---|---|---|---|---|---|---|---|---|---|---|---|---|
| 020620-3 | ZHe | 44.122 | 10.068 | 0.756 | 0.395 | 25 | 38.2 | 0.71 | 6.6 | 178.1 | 27 | 40 | 0.667 | 6.23 | 179.4 | della Vedova et al. (2001) |
| 020620-3 rep | ZHe | 44.122 | 10.068 | 0.756 | 0.395 | 25 | 38.3 | 0.71 | 6.7 | 179.0 | 27 | 40 | 0.675 | 6.29 | 180.1 | della Vedova et al. (2001) |

**Kinetic Parameters for (U-Th)/He zircon** *(from Reiners et al., 2004)*

$E_a$ = 169 kJ mol$^{-1}$ (activation energy)

$a_s$ = 60 µm (effective spherical radius for the diffusion domain)

$\Omega$ = 7.03 x 10$^5$ (frequency factor calculated as $55D_0a^{-2}$)

$t_{c,10}$ = 183°C (effective closure temperature for 10 Myr$^{-1}$ cooling rates and specified $a_s$ value)



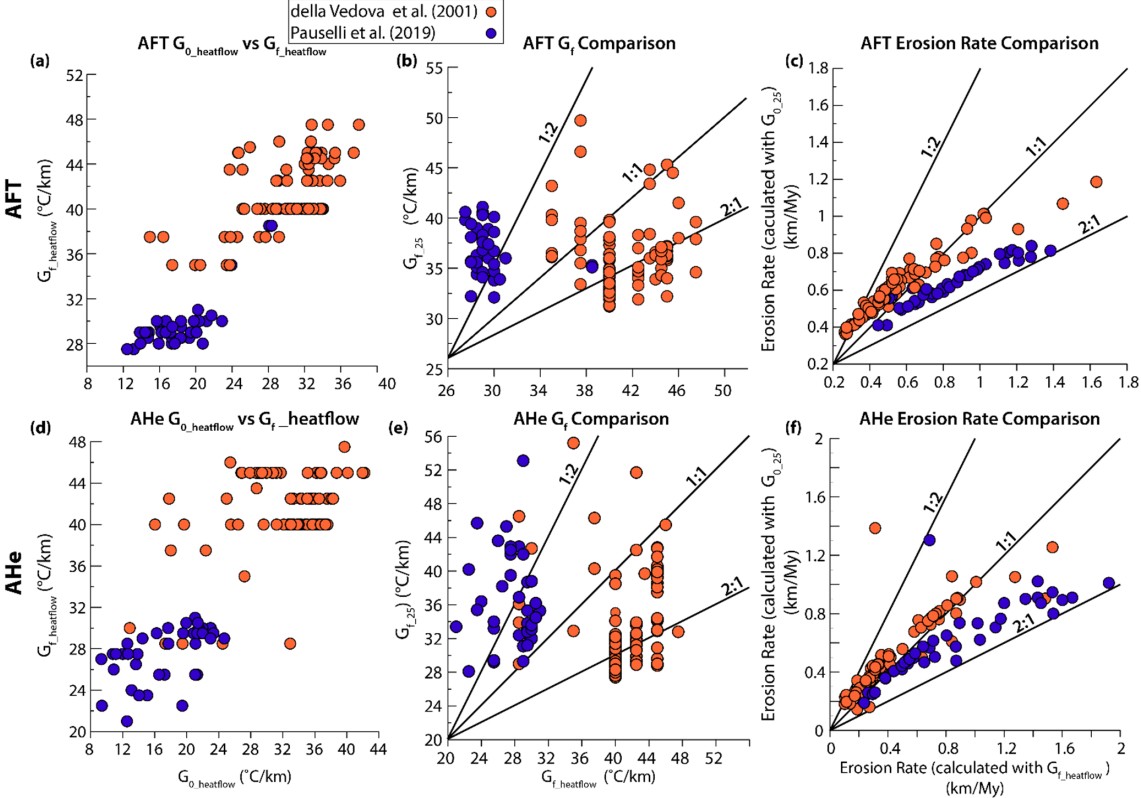

**Figure 6 Comparison of initial geothermal gradients (G₀) and final geothermal gradients (Gf) for AFT samples (a–c) and AHe samples (d–f). (a) and (d) Comparison of G$_{0\_heatflow}$ and G$_{f\_heatflow}$. (b) and (e) Comparison of G$_{f\_heatflow}$ with G$_{f\_25}$. (c) and (f) Comparison of erosion rates derived from G$_{f\_heatflow}$ measurements versus erosion rates derived from imposed G$_{0\_25}$.**

Here, we present the erosion rate results for the Adriatic and Ligurian sides of the orogen. Calculated with G$_{0\_25}$, erosion rates inverted from AFT bedrock ages (Table 9; top row, Fig. 8) vary between 0.41 km/My and 1.19 km/My on the Adriatic side (Fig. 7d) and between 0.36 km/My and 0.84 km/My on the Ligurian side (Fig. 7c). The highest erosion rates on the Ligurian side are located in the Macigno Unit near the Alpi Apuane, whereas the highest rates on the Adriatic side are located near the

315 drainage divide (Cervarola Unit; top row, Fig. 8). AFT bedrock erosion rates across the divide are similar or slightly higher on the Adriatic side (Fig. 7c–d, top row, Fig. 8). AFT detrital erosion rates are similar to bedrock AFT rates, which also exhibit erosion rates that are higher on the Adriatic side (top row, Fig. 8). Erosion rates derived from AHe ages (Table 8) range between 0.14 and 1.39 km/My on the Adriatic side and between 0.14 and 0.74 km/My on the Ligurian side. AHe erosion rates on the Adriatic side are more variable relative to the Ligurian side, particularly in the southeast region of the field area (Fig. 7c–d).



Similar to the bedrock AFT erosion rates, the highest AHe erosion rates are found on the Adriatic side near the drainage divide and are lowest near the Ligurian coastline (top row, Fig. 8).

Calculated with $G_{f\_heatflow}$, the pattern of erosion across the drainage divide is similar to the pattern for erosion rates calculated with $G_{0\_25}$ (middle row, Fig. 8). On the Adriatic side, erosion rates inverted from AFT bedrock and detrital ages vary between

325 0.34 and 1.63 km/My and between 0.88 and 1.44 km/My, respectively. On the Ligurian side, AFT bedrock erosion rates vary between 0.26 and 1.28 km/My, and detrital AFT erosion rates vary between 0.34 and 0.78 km/My. Erosion rates derived from AHe ages range from 0.17 to 1.92 km/My on the Adriatic side and from 0.10 to 1.02 km/My on the Ligurian side. Detrital AFT erosion rates on the Adriatic side are higher relative to the Ligurian side, regardless of the method used for constraining the geothermal gradient. However, calculated from $G_{f\_heatflow}$, detrital AFT erosion rates on the Adriatic side are up to a factor

of two higher than erosion rates calculated with $G_{0\_25}$ (Fig. 8).


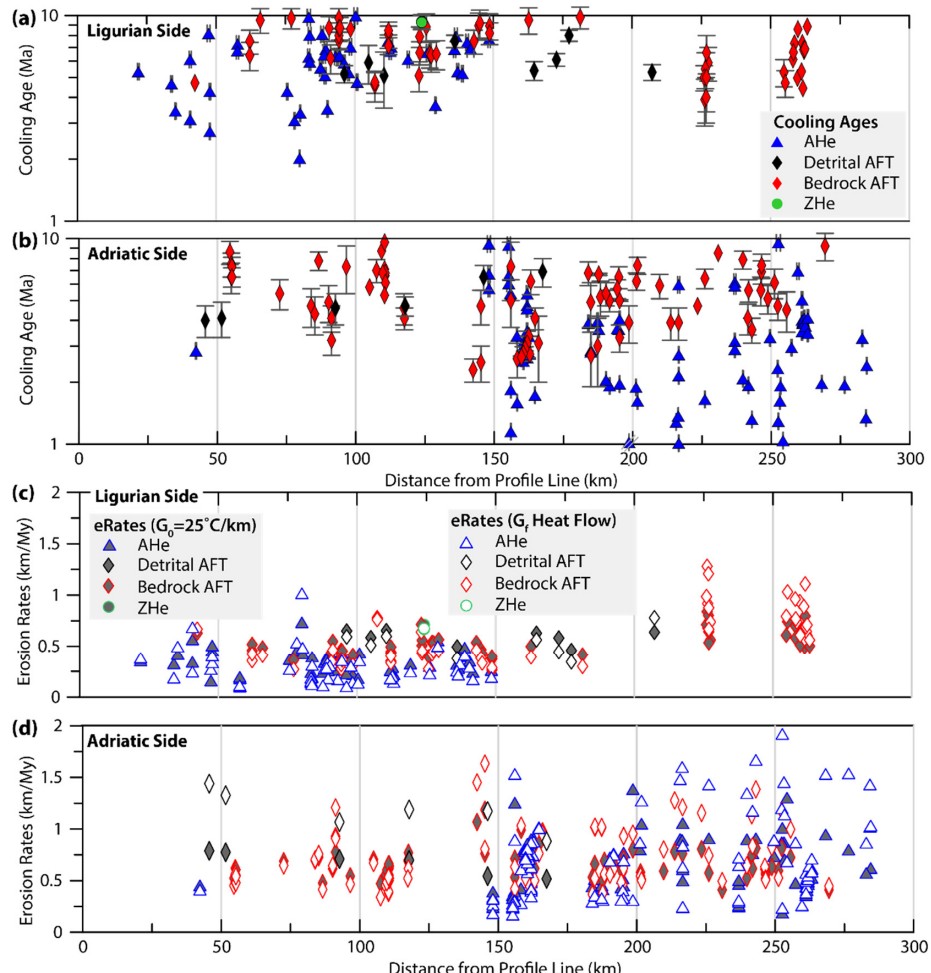

**Figure 7 Cooling ages for all thermochronometers plotted along the Adriatic side (a) and the Ligurian side (b). Erosion rates for the (c) Adriatic side and (d) Ligurian side. Profile location for all plots is illustrated in Fig. 3a. In (b), individual AHe symbol dated to <1 Ma is illustrated with a double gray slash.**

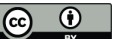



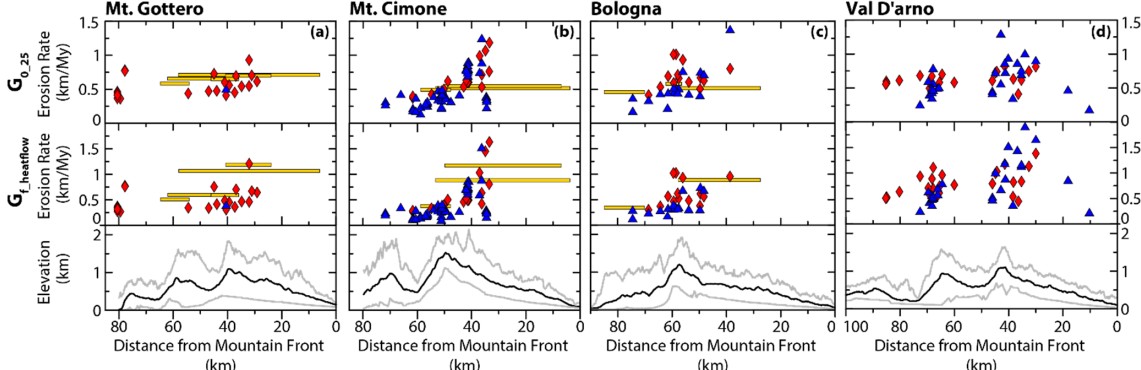

**Figure 8 Erosion rates for AHe, bedrock AFT, and detrital AFT thermochronometers calculated from $G_{0\_25}$ (left column, a-d) or calculated with $G_{f\_heatflow}$ (right column, a-d). Length of the detrital AFT boxes reflects the distance from sample location to the catchment headwaters where the erosion rate is valid. Swath profile locations are shown in Fig. 3a. In the Bologna swath profile (right column), one AHe sample could not be resolved for an erosion rate with $G_{f\_heatflow}$.**

### 3.3 Paired ages

Of the 30 paired samples analyzed here, erosion rates for two samples could not be resolved (Table 12), due to the similarity in ages between the AFT and AHe thermochronometers (sample C16) or due to an AHe age that is older than the AFT age (sample C22). Six paired samples are located on the Ligurian side of the orogen, and the remaining 22 samples are located on the Adriatic side of the orogen (Fig. 9).

Erosion rates from samples on the Adriatic side vary from ~0.3 to 5.2 km/My (Table 12). Twelve samples illustrate an increase in erosion through time (Fig. 9a), with the shift generally occurring between ~1 and 2 Ma. Ten samples from the Adriatic side illustrate a decrease in erosion rate through time (Fig. 9b), with the shift generally occurring between ~2 and 4 Ma. With the exception of three samples (AP53, AP57, and C29), decelerating sites are located in the headwaters of the Reno River (inset map, Fig. 9c), which drains the Cervarola and Macigno Units east of Mt. Cimone (Fig. 1). For the six paired samples from the Ligurian side, the range of erosion rates is similar and varies from ~0.3 to 5.1 km/My (Table 12). However, only one sample on the Ligurian side illustrates an increase in erosion rate, occurring at ~3 Ma (Fig. 9c). All other samples illustrate a decrease in erosion through time, with the shift occurring between ~6 and 2 Ma (Fig. 9d).





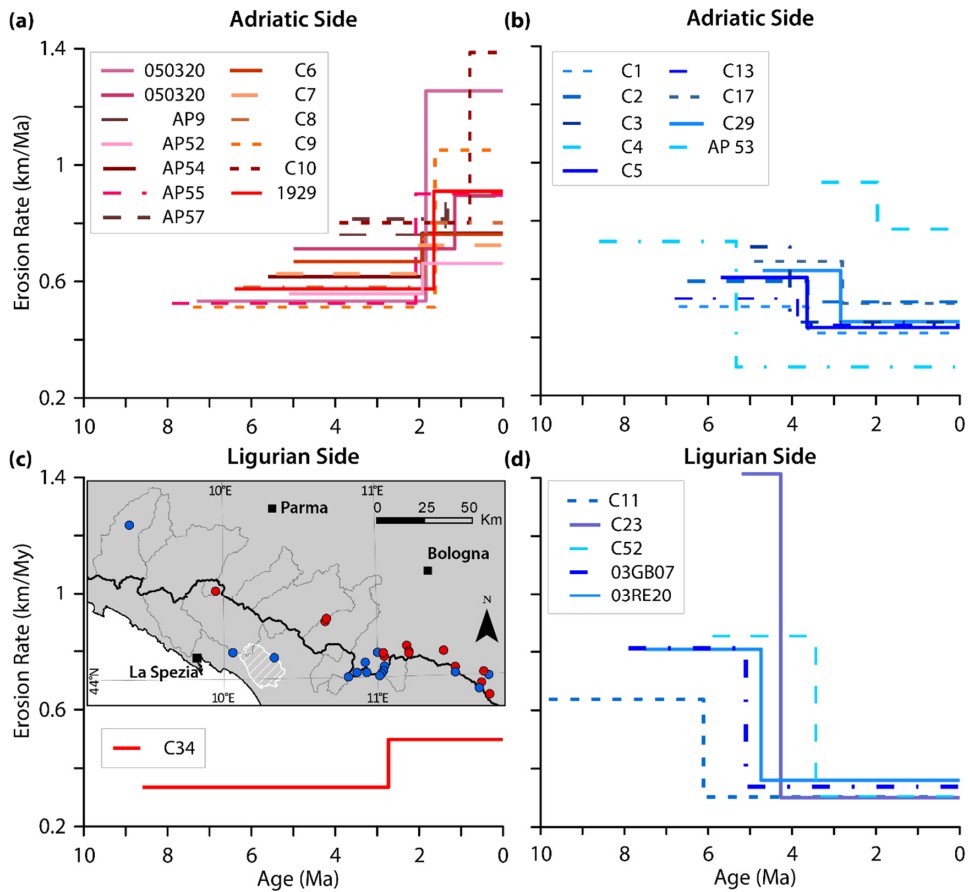

**Figure 9** Erosion rates through time from paired AFT and AHe age samples on the Adriatic side for (a) increasing erosion rates and (b) decreasing erosion rates, and on the Ligurian side for (c) increasing erosion rates and for (d) decreasing erosion rates. Inset map in (c) shows the locations of paired thermochronometer age samples (circles). Red circles indicate locations where the erosion rate through time is increasing, and blue circles indicates locations where the erosion rate through time is decreasing.

**Table 12** Erosion rates and parameters for paired AFT-AHe thermochronometer samples.





| | ID | Latitude | Longitude | Sample Elevation (km) | Method | Corrected Age (Ma) | Mean Elevation (km) | $t_1$ (Ma) | $G_0$ (°C/km) | $G_f$ (°C/km) | Erosion Rate (km/My) | $D_c$ (km) | $T_c$ (°C) | Method | Mean Elevation (km) | $t_1$ (Ma) | $G_0$ (°C/km) | $G_f$ (°C/km) | Erosion Rate (km/My) | $D_c$ (km) | $T_c$ (°C) |
|---|---|---|---|---|---|---|---|---|---|---|---|---|---|---|---|---|---|---|---|---|---|
| Ligurian Side | C11 | 44.00 | 10.81 | 0.95 | AFT | 3.19 | 0.64 | 3.39 | 33.78 | 35.15 | 0.46 | 1.47 | 117.40 | AHe | 0.67 | 6.61 | 39.85 | 41.07 | 0.25 | 1.53 | 61.96 |
| | C23 | 44.02 | 10.93 | 0.88 | AFT | 0.43 | 0.69 | 5.23 | 6.34 | 36.79 | 5.09 | 2.19 | 138.26 | AHe | 0.72 | 4.77 | 39.05 | 40.72 | 0.35 | 1.48 | 64.01 |
| | C34 | 44.42 | 9.95 | 0.51 | AFT | 5.37 | 0.82 | 6.77 | 30.14 | 32.02 | 0.27 | 1.47 | 113.63 | AHe | 0.85 | 3.23 | 34.54 | 36.57 | 0.42 | 1.15 | 64.11 |
| | C52 | 44.01 | 11.50 | 0.36 | AFT | 1.97 | 0.59 | 6.07 | 11.42 | 20.78 | 1.59 | 3.13 | 123.26 | AHe | 0.66 | 3.93 | 23.41 | 25.08 | 0.52 | 1.78 | 62.35 |
| | 03GB07 | 44.12 | 10.06 | 0.68 | AFT | 2.30 | 0.36 | 4.40 | 25.35 | 31.29 | 0.77 | 1.76 | 122.35 | AHe | 0.36 | 5.60 | 34.35 | 35.84 | 0.34 | 1.73 | 62.86 |
| | 03RE20 | 44.10 | 10.33 | 1.06 | AFT | 2.26 | 0.76 | 4.76 | 24.27 | 31.01 | 0.82 | 1.84 | 122.76 | AHe | 0.81 | 5.24 | 33.98 | 35.72 | 0.37 | 1.74 | 63.04 |
| Adriatic Side | 1929 | 44.04 | 11.50 | 0.70 | AFT | 4.25 | 0.62 | 7.85 | 13.16 | 16.43 | 0.61 | 2.61 | 114.70 | AHe | 0.70 | 2.15 | 18.23 | 22.72 | 1.61 | 2.66 | 67.83 |
| | 050320-1a | 44.26 | 10.66 | 1.11 | AFT | 5.65 | 0.98 | 8.35 | 25.82 | 27.17 | 0.16 | 0.89 | 108.66 | AHe | 1.02 | 1.65 | 28.22 | 35.41 | 1.70 | 1.96 | 72.78 |
| | 050320-1b | 44.26 | 10.66 | 1.11 | AFT | 2.66 | 0.98 | 7.66 | 24.38 | 30.40 | 0.53 | 1.40 | 119.45 | AHe | 1.02 | 2.34 | 32.04 | 36.54 | 0.96 | 1.77 | 69.28 |
| | AP52 | 43.91 | 11.72 | 0.57 | AFT | 2.68 | 0.80 | 7.58 | 9.81 | 16.26 | 1.21 | 3.26 | 119.27 | AHe | 0.84 | 2.42 | 18.23 | 21.66 | 1.27 | 2.44 | 66.08 |
| | AP53 | 43.93 | 11.66 | 0.91 | AFT | 2.77 | 0.81 | 4.17 | 16.30 | 20.55 | 1.02 | 2.84 | 119.53 | AHe | 0.81 | 5.83 | 24.20 | 25.47 | 0.39 | 2.09 | 60.74 |
| | AP 54 | 43.96 | 11.67 | 0.69 | AFT | 3.17 | 0.75 | 7.57 | 11.58 | 16.74 | 0.93 | 2.95 | 117.93 | AHe | 0.81 | 2.43 | 18.68 | 22.33 | 1.31 | 2.52 | 66.51 |
| | AP 55 | 44.01 | 11.69 | 1.07 | AFT | 5.32 | 0.67 | 7.42 | 13.60 | 15.55 | 0.49 | 2.60 | 112.30 | AHe | 0.72 | 2.58 | 17.52 | 21.41 | 1.50 | 3.13 | 67.07 |
| | AP 57 | 44.00 | 11.72 | 0.45 | AFT | 1.78 | 0.69 | 8.18 | 6.12 | 15.23 | 2.04 | 3.62 | 122.67 | AHe | 0.75 | 1.82 | 16.93 | 21.92 | 1.93 | 2.55 | 68.58 |
| | AP9 | 44.12 | 11.43 | 0.40 | AFT | 2.03 | 0.67 | 8.13 | 7.33 | 15.42 | 1.66 | 3.38 | 121.43 | AHe | 0.67 | 1.87 | 17.12 | 22.00 | 1.84 | 2.52 | 68.13 |
| | C1 | 44.11 | 11.00 | 0.50 | AFT | 2.58 | 0.79 | 5.88 | 25.88 | 32.66 | 0.67 | 1.72 | 121.30 | AHe | 0.76 | 4.12 | 35.25 | 36.82 | 0.32 | 1.16 | 62.44 |
| | C10 | 44.14 | 11.19 | 0.61 | AFT | 2.61 | 0.68 | 8.71 | 12.44 | 17.76 | 0.75 | 1.97 | 117.30 | AHe | 0.77 | 1.29 | 18.98 | 28.02 | 2.92 | 2.31 | 73.22 |
| | C13 | 44.02 | 10.86 | 0.70 | AFT | 2.43 | 0.73 | 5.63 | 32.26 | 39.36 | 0.58 | 1.41 | 122.14 | AHe | 0.75 | 4.37 | 41.97 | 43.76 | 0.30 | 1.17 | 63.40 |
| | C16 | 44.06 | 10.91 | 0.63 | AFT | | 0.81 | NA | NA | NA | NA | NA | NA | AHe | 0.91 | NA | NA | NA | NA | NA | NA |
| | C17 | 44.07 | 10.92 | 0.63 | AFT | 1.60 | 0.82 | 6.70 | 23.57 | 37.11 | 1.08 | 1.72 | 126.83 | AHe | 0.92 | 3.30 | 39.53 | 41.55 | 0.37 | 1.03 | 64.26 |
| | C2 | 44.00 | 11.01 | 0.83 | AFT | 2.38 | 0.55 | 5.88 | 28.32 | 35.73 | 0.66 | 1.57 | 122.59 | AHe | 0.66 | 4.12 | 38.26 | 40.39 | 0.42 | 1.52 | 65.06 |
| | C22 | 44.04 | 10.93 | 0.85 | AFT | | 0.75 | NA | NA | NA | NA | NA | NA | AHe | 0.82 | NA | NA | NA | NA | NA | NA |
| | C29 | 44.73 | 9.39 | 0.32 | AFT | 1.36 | 0.80 | 6.66 | 13.37 | 27.72 | 1.75 | 2.38 | 127.55 | AHe | 0.77 | 3.34 | 30.05 | 31.87 | 0.43 | 1.22 | 63.09 |
| | C3 | 44.01 | 11.03 | 0.78 | AFT | 0.45 | 0.58 | 5.45 | 6.08 | 36.98 | 5.15 | 2.32 | 138.07 | AHe | 0.71 | 4.55 | 39.29 | 40.93 | 0.34 | 1.38 | 63.95 |
| | C4 | 44.03 | 11.04 | 0.68 | AFT | 0.84 | 0.62 | 7.54 | 11.77 | 33.02 | 2.29 | 1.93 | 132.26 | AHe | 0.75 | 2.46 | 34.89 | 38.48 | 0.74 | 1.45 | 68.27 |
| | C5 | 44.05 | 11.04 | 0.61 | AFT | 1.56 | 0.68 | 5.86 | 21.50 | 34.80 | 1.23 | 1.92 | 126.81 | AHe | 0.80 | 4.14 | 37.54 | 39.10 | 0.32 | 1.17 | 63.09 |
| | C6 | 44.10 | 11.04 | 0.65 | AFT | 2.58 | 0.74 | 7.58 | 23.04 | 29.78 | 0.61 | 1.57 | 120.27 | AHe | 0.78 | 2.42 | 31.58 | 35.10 | 0.80 | 1.54 | 68.05 |
| | C7 | 44.11 | 11.04 | 0.63 | AFT | 2.85 | 0.74 | 7.45 | 24.02 | 30.05 | 0.54 | 1.53 | 119.02 | AHe | 0.75 | 2.55 | 31.88 | 35.26 | 0.74 | 1.51 | 67.35 |
| | C8 | 44.12 | 11.21 | 0.70 | AFT | 3.81 | 0.68 | 7.61 | 22.86 | 26.72 | 0.42 | 1.60 | 116.20 | AHe | 0.76 | 2.39 | 28.51 | 32.28 | 0.94 | 1.77 | 68.29 |
| | C9 | 44.11 | 11.20 | 1.00 | AFT | 5.28 | 0.67 | 7.88 | 22.23 | 24.01 | 0.26 | 1.39 | 111.75 | AHe | 0.74 | 2.12 | 25.51 | 30.82 | 1.40 | 2.27 | 70.19 |

### 3.4 Kinematic model

The orogenic wedge model shows how the spatial pattern of exhumation rates relates to the polarity of accretion and the pattern

of horizontal and vertical motion. Figure 10 illustrates the predicted horizontal velocities, uplift rates, and material paths through the wedge for the spatially constant erosion rate setup (SCR) and the spatially variable erosion rate setup (VER). Horizontal velocities at the toes of the wedge are equal to the rate of slab rollback and decrease to a minimum at the drainage divide between the prowedge and retrowedge (Fig. 10a). Extension in the retrowedge results in higher horizontal erosion rates towards the Ligurian coast.

For the SCR setup, we ran the model with a single, orogen-wide erosion rate varying between 0.4 and 1.0 km/My, which are values consistent with average AHe-derived erosion rates for the Adriatic side of the orogen. Here, we illustrate the best-fit model results for an orogen-wide erosion rate of 0.8 km/My and a slab rollback rate ($V_P$) = 10 km/My. Horizontal velocities decrease to a minimum of 2.0 km/My at the drainage divide (Fig. 10a). Uplift rates on the prowedge vary from 0.89 to 1.1

375 km/My, and from 0.3 to 0.66 km/My on the retrowedge (Fig. 10b). Predicted reset cooling ages across the orogen for AHe samples (triangles) vary from 2.8 to 4.4 Ma, from 4.2 to 7.6 Ma for AFT (diamonds), and from 6.5 to 11.5 Ma for ZHe samples





(circles) (Fig. 11a, c). Maximum burial depth increases almost linearly across the orogenic wedge, ranging from 2.1 to 6.1 km on the prowedge and from 6.9 to 13.1 km on the retrowedge (Fig. 11b).

For the VER setup, we illustrate the best-fit model results for a prowedge erosion rate of 0.8 km/My and a retrowedge erosion rate of 0.4 km/My. Horizontal velocities decrease to a minimum of 2.0 km/My at the drainage divide (Fig. 10D). Uplift rates on the prowedge are the same as in the SCR (0.87–1.1 km/My), but uplift rates on the retrowedge are lower (-0.1–0.26 km/My) (Fig. 10e). Predicted reset cooling ages range from 3.8 to 5.8 Ma for AHe samples, from 6.2 to 8.3 Ma for AFT samples, and from 12.1 to 12.8 Ma for ZHe samples. For samples that reach the surface on the prowedge side of the

range, predicted reset cooling ages decrease towards the primary divide. On the retrowedge, ages initially increase away from the divide, but young slightly at the model boundary (Fig. 11d). Maximum burial depth increases almost linearly along the prowedge (2.9–6.1 km) up to a kink at the drainage divide, which reflects the shift towards a shallower slope and smaller range of maximum burial depths along the retrowedge (6.9–9.4 km) (Fig. 11e).

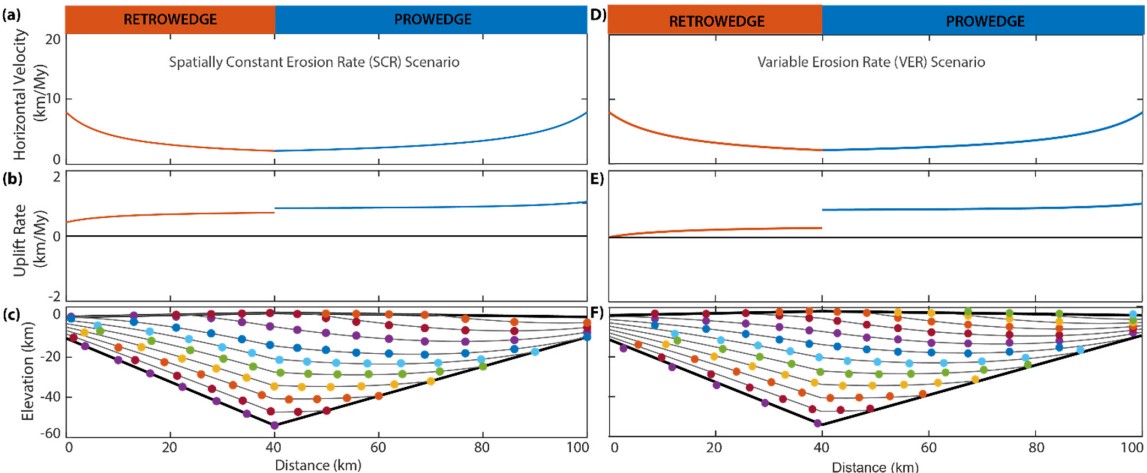

**Figure 10 Kinematic model results. (a) Predicted material horizontal velocities across the orogenic wedge and (b) Predicted uplift rates across the wedge. (c) Material motion paths (lines within wedge) and particle positions and paths, equally spaced in time (solid, colored circles).**


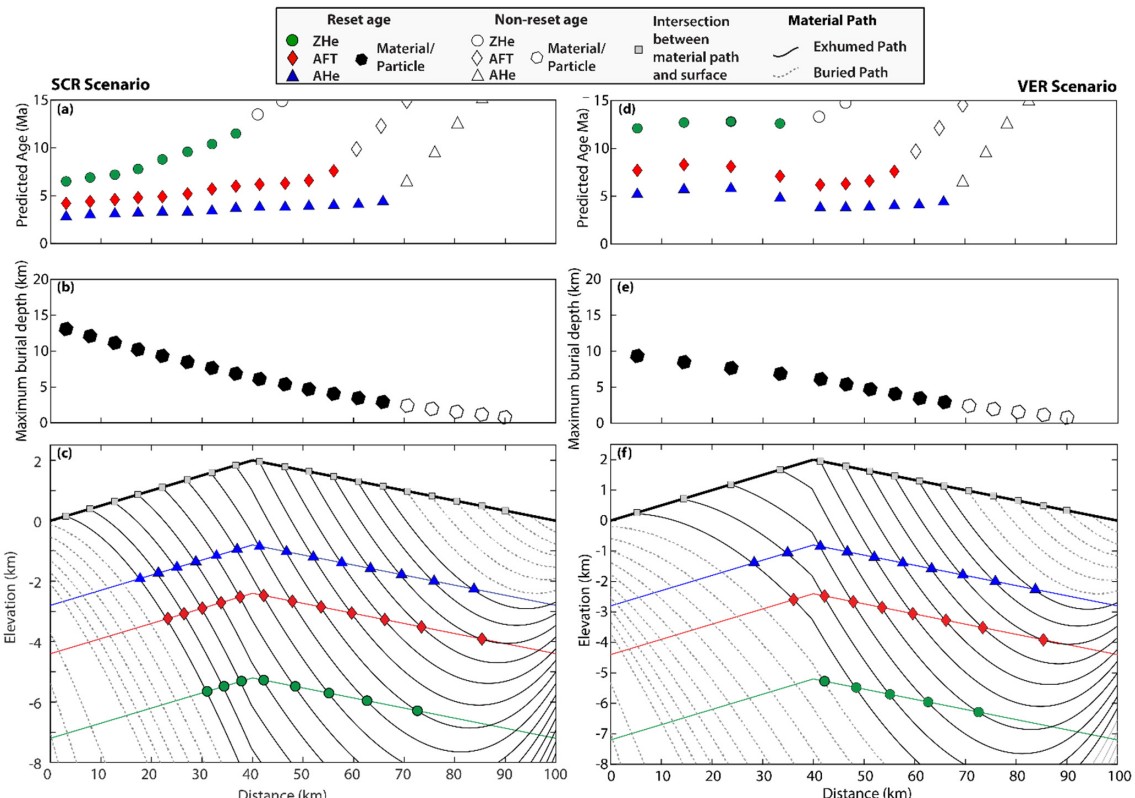

**Figure 11** Closeup of kinematic model shown in Fig. 9. Left and right panels represent spatially constant erosion rate (SCR) and variable erosion rate (VER) model setups, respectively. (a) and (d) Predicted age of thermochronometers with distance along the wedge. (b) and (e) Predicted maximum burial depths for each particle path. (c) and (f) Material paths in upper 8 km of kinematic model. Colored lines illustrate closure depths for AHe, AFT, and ZHe thermochronometers.

## 4 Discussion

### 4.1 Detrital versus bedrock ages

Previous studies place the onset of exhumation between 8 and 14 Ma (Balestrieri et al., 1996; Ventura et al., 2001), so it is not clear whether the 12–13 Ma old population in the Vara and Magra samples represent partially or completely reset cooling ages. However, these ages are consistent with high-elevation samples west of the Vara catchment that record slow cooling prior to 8 Ma. The 8.2 Ma age peak is present in the Magra1 sample, which drains the extensional intermontane basin within the catchment, but is absent in the Magra2 sample, which drains only small tributaries upstream of the basin. Thus, the peak at 8.2

405   Ma in Magra1 likely reflects exhumation ages of the nearby Macigno Unit, which would have been eroded and redeposited



into the Pliocene basins (Balestrieri, 2000; Fellin et al., 2007). The pulse of exhumation in the Northern Apennines between 6 Ma and 4 Ma, when most of the Northern Apennines became sub-aerially exposed (Zattin et al., 2002; Balestrieri et al., 2003; Fellin et al., 2007), is consistent with the youngest peaks shown in the Vara, Magra, Lima, and Bisenzio Rivers.

The pattern of detrital AFT ages on the Ligurian and Adriatic sides (Malusà and Balestrieri, 2012) is consistent with the pattern of bedrock AFT ages, which illustrate younging exhumation ages towards the northeast, regardless of whether cooling ages are corrected or uncorrected for topography (Fig. 2). Overall, we find consistent results between the detrital AFT and bedrock AFT ages, reinforcing that the reset detrital ages illustrate a true exhumation signal, rather than an artefact of the technique. Fertility analysis of sediment from sampled Adriatic catchments also confirm that the detrital samples are representative of the

eroded bedrock (Malusà et al., 2016), in the absence of hydraulic sorting effects. Finally, we note that using an initial exhumation age of 14 Ma would not change the pattern of exhumation, but would proportionally decrease all erosion rates in the Northern Apennines if the geothermal gradient increases in response to a longer period of erosion. As we incorporate only minimum reset ages from each sample, inclusion of the 13 Ma-year age population would have no effect on the erosion rate calculations.

**4.2 Inferred erosion rate and relationship to the geothermal gradient**

The geothermal gradient is the most important external parameter for converting cooling ages into erosion rates (Willett and Brandon, 2013). The two major studies of regional heat flow (della Vedova et al., 2001 and Pauselli et al., 2019) show large variations in heat flow across the region, but are also internally inconsistent by up to 50 mW/m$^2$ in the regions where they overlap. It is thus unclear how much of the spatial variability is real and how much is due to local effects or local errors in heat

flow measurements, a large source of uncertainty for the geothermal gradients and ultimately, the erosion rates. To address this uncertainty, we took two approaches. First, we assumed that the initial geothermal gradient in the region was uniform and all variations in the modern geothermal gradient are due to advection in response to erosion. Second, we constrained the thermal model to be consistent with the modern heat flow measurements and inferred an initial geothermal gradient that was spatially variable.

The uncertainties in the modern heat flow measurements are evident in the erosion rate analysis, particularly when comparing the range of $G_{0\_heatflow}$ inferred from $G_{f\_heatflow}$ inputs that are calculated with heat flow measurements (Fig. 6a, d). However, it is unclear whether the large range of $G_{0\_heatflow}$ values represents how the geothermal gradient may have varied in either space or time at the onset of erosion. As the Northern Apennines evolved, sediments were accreted to the accretionary wedge shortly

after being deposited in a subsiding foreland basin, whose modern equivalents are the Po Plain and the Adriatic Sea. There, modern heat flow values are generally low (≤ 50 mW m$^{-2}$) (della Vedova et al. 2001), although with significant spatial variations (Pauselli et al., 2019), indicating that the present geothermal gradients in the foreland should be not higher than





about 30 ºC/km. Given the uncertainties on the modern heat flow measurements, the erosion analysis approach based on a common, $G_{0\_25}$ can be considered a viable alternative to the approach based on an input $G_{f\_heatflow}$.

The erosion rates resulting from these two analyses differ significantly: erosion rates from one analysis are a factor of two different from the alternate analysis (Fig. 6). However, the two sets of results projected along swath profiles from SW to NE show little difference in their spatial patterns across the main Apenninic divide (Fig. 8). The main differences are that the erosion rates derived from $G_{f\_heatflow}$ vary over a larger and higher range than those derived from $G_{0\_25}$, and the maximum rates

are higher from $G_{f\_heatflow}$. In particular, the youngest detrital age populations give much larger rates on the Adriatic side than on the Ligurian side with the analysis based on $G_{f\_heatflow}$. These observations suggest that the erosion rates derived from $G_{0\_25}$ may be more conservative estimates overall. However, the most important observation for the scope of this contribution remains that the large-scale spatial pattern of erosion rates along the swath profile does not change with the employed analysis method.

**4.3 Erosion rate patterns**

Bedrock cooling ages on the Ligurian side of the Northern Apennines generally vary between 4 and 10 Ma, with only a few ages younger than 4 Ma. On the Adriatic side, bedrock cooling ages younger than 4 Ma are a large component of the age distributions, especially among the AHe ages (Fig. 7b). Similarly, the youngest populations for detrital AFT samples on the Ligurian side are nearly all older than the youngest detrital AFT populations on the Adriatic side (Fig. 7a-b).

Erosion rates derived from both bedrock and detrital thermochronometric ages suggest a difference between the Ligurian and Adriatic sides that is valid at the regional scale, regardless of the method used for constraining the geothermal gradients. An exception to this general pattern may be the Alpi Apuane massif, which represents a structural culmination exposing a deep section. It is likely that this exhumation reflects structural control that is unique to the Northern Apennines. On the Ligurian

side, erosion rates derived from bedrock AFT ages tend to be higher than erosion rates obtained from bedrock AHe ages (Fig. 7C) reflecting a regional decrease in erosion rate. This is particularly evident in the region east of the Alpi Apuane, at the main drainage divide north of Florence and in the Val d'Arno (Fig. 1 and 2). In contrast, on the Adriatic side, erosion rates derived from AHe ages tend to be higher than erosion rates obtained from AFT ages (Fig. 7D) and suggest an increase in erosion rate over the last 5 Ma.

Paired thermochronometers on the same sample (as for instance, AFT and AHe) or age-elevation transects (AETs) also indicate changes in erosion rate. The majority of the paired age samples (12) from the Adriatic side illustrate an increase in erosion rate through time (Fig. 9a), although 10 of these samples illustrate a decrease in erosion through time (Fig. 9b). Of these 10 samples, seven are from one region of the upper Reno River Valley (inset map, Fig. 9c). The headwaters of the Reno River Valley

extend farther south than the adjacent basins of the Serchio River and Bisenzio River, which flow to the Ligurian Sea (Fig.



3a). Interestingly, the exhumation rates from the upper Reno River are similar to rates from the Serchio River, suggesting that the upper Reno River presents an erosion rate signal akin to Ligurian Rivers, rather than to Adriatic Rivers, and are thus resolving a consistent pattern of erosion rate in space, but not restricted to catchment boundaries. We also note that modern erosion rates from cosmogenic nuclide concentrations in the upper Reno tributaries are at least a factor of three lower than

rates for the entire basin (Cyr et al., 2010), suggesting that this trend of lower erosion rates in this area has continued to the present.

Paired ages from the Ligurian side show the opposite trend. With the exception of one sample (C34) (Fig. 9c) located within the Magra2 catchment area (Fig. 3a), all other samples consistently illustrate a decrease in erosion rate through time (Fig. 9d).

Thus, the results from the paired thermochronometer ages on the Adriatic and Ligurian sides of the orogen confirm the regional trends observed from the simple erosion rate analysis method.

The results from our paired ages analysis can also be discussed in the context of the AETs from Mt. Falterona, Mt. Cimone, and Val d'Arno (see Fig. 1 for locations). Between 4 and 2 Ma, these AETs have previously been interpreted to reflect an

orogen-wide increase in exhumation and erosion rates, although there are notable differences between the results from the profiles on the Adriatic side (Mt Falterona and Mt. Cimone) and on the Ligurian side (Val d'Arno) (Thomson et al., 2010). The Mt. Falterona and Cimone AETs illustrate a two-fold increase in erosion rates between 4 and 5 Ma, from $0.29 \pm 0.1$ km/My to $0.58 \pm 0.23$ km/My and from $0.22 \pm 0.09$ km/My to $0.58 \pm 0.16$ km/My, respectively (Thomson et al., 2010). Excluding the samples from the upper Reno River Valley that illustrate a decrease in erosion rate through time, our paired ages analysis

reflects erosion rates that increase from $0.68 \pm 0.42$ km/My to $1.31 \pm 0.70$ km/My, given 1s uncertainties. Our average erosion rates are higher than the average erosion rates calculated for the Mt. Falterona and Cimone AETs, although, given the high uncertainties in our values, they are within the range of the AETs.

The results from the Val d'Arno AET are less straightforward, due to the fact that some samples illustrate a decrease in erosion

rate through time, while others illustrate an increase in erosion rate through time. When corrected for topographic and advection effects, this AET shows a negative slope that was previously interpreted to reflect post-cooling tilting of the footwall block of an extensional fault (Thomson et al., 2010). On the Ligurian side, cooling ages and erosion rates vary locally as a function of elevation and of fault activity, and extensional faults can control differences in the exhumation pattern. However, in light of our results from the simple analysis of erosion rates and the paired ages, this negative slope could also be interpreted as a

decrease in erosion rates, and thus would reflect a regional signal, rather than local tectonics. We infer that such regional scale-differences must be controlled by first-order features of the Northern Apennines. In order to address the question of what could control such differences, we compare two different kinematic models for the Northern Apennines orogenic wedge.





**4.4 Kinematic model**

The orogenic wedge kinematic models illustrate differences in cooling ages, maximum burial temperatures, and material
paths across the Northern Apennines, assuming simple continuum accretion and mass balance (Fig. 11). Using a spatially
constant erosion rate across the orogenic wedge (SCR) predicts that reset ages decrease from northeast to southwest and are
youngest on the retrowedge model boundary. In contrast, the variable erosion rate setup (VER) predicts minimum reset ages
near the drainage divide and maximum reset ages on the retrowedge, close to the model boundary. The VER model is
consistent with observed cooling ages, which are youngest near the drainage divide in the core of the Northern Apennines
(Fig. 2).

Vitrinite reflectance (VR) data provide an additional estimate for maximum paleotemperature and burial depth and thus, an
additional calibration of the kinematic model. In the Northern Apennines, Ro values reach 5.1% at the Ligurian coastline
along the Mt. Gottero swath profile (Fig. 2a). With the exception of this profile, maximum VR values are generally within
the range of 1.5–2.5 % for the Mt. Cimone, Bologna, and Val d'Arno profiles (Fig. 2b-d). Maximum paleotemperatures from
VR are estimated at 200-250°C in the core of the range and along the Ligurian coastline in the northwest (Fellin et al., 2007),
whereas paleotemperatures are 150–190°C in the Cervarola Unit (VR = 1.0–1.7%), and are 100-110°C in the Ligurian Unit
(VR = 0.5–0.6%) (Ventura et al., 2001; Botti et al., 2004). Maximum paleotemperatures should correspond to maximum
burial depths; thus, we expect to find the maximum burial depths along the Ligurian coastline and near the drainage divide in
the Cervarola Unit. Both the SCR and VER models predict maximum burial depths near the Ligurian coastline, consistent
with the trends observed in the Mt. Gottero and Val d'Arno profile. Near the drainage divide on the prowedge, predicted
maximum burial depths are 6.1 km for both models.

We can estimate maximum burial depths at the drainage divide, given the generalized relationship between Ro and burial
depth (Suggate, 1998), VR values, and a modern geothermal gradient. To estimate the modern geothermal gradient at the
drainage divide, we use $G_{f\_heatflow}$ rates from the simple erosion analysis for AHe samples in the Cervarola Unit (Tables 1 and
7). Given these parameters. we estimate maximum burial depths in the range of 4.0–5.5 km near the drainage divide; the
upper bound of this range are similar to the predicted maximum burial depths (6.1 km) for both the SCR and VER models.
Collectively, our erosion rate analysis and kinematic model illustrate that the east to west particle trajectories, combined with
lower erosion rates on the retrowedge, are consistent with the spatial pattern of cooling ages and maximum
paleotemperatures estimated from both vitrinite reflectance and thermochronometric data.

The particle paths in the VER kinematic model, combined with the lower erosion rates in the retrowedge, suggest an
explanation for the apparent decrease in erosion rates with time on the Ligurian side of the Apennines. As rocks are advected
from prowedge to retrowedge, the vertical component of their motion decreases (Fig. 10f). Particle paths where the AFT age



is set in the prowedge, but where the AHe is set in the retrowedge, will record this change as a temporal deceleration of cooling rate. However, rather than representing a change in surface erosion rate, the change in cooling rate reflects the motion of the rock from the fast erosion rate prowedge into the low erosion rate retrowedge. The fact that the decelerating sites are all found to the southwest, regardless of drainage basin (inset, Fig. 8c) supports the idea that this is a tectonically
controlled spatial pattern.

The acceleration of exhumation observed in the prowedge is not explained by the kinematic model. Although the apparent increase in exhumation rate might be explained by spatial changes in tectonic uplift and an associated increase in erosion rate, there is no strong evidence for this in the spatial pattern of ages or in the geomorphology, which should show higher
uplift rates in the range interior. In contrast, the highest uplift rates are more often observed at the mountain front (Picotti and Pazzaglia, 2008). It is more likely that this is a true temporal increase in erosion rate, which could be associated with an increase in accretionary flux as the mountain front advanced into the Alpine sediments of the Adriatic foreland. The foreland basin fill thickened as Miocene alpine sediments filled the foredeep and again in the Quaternary as glacial sediments filled the Po plain and parts of the Adriatic Sea. The increase in accretionary flux would lead to an increase in wedge size and in
erosion rate, processes that our kinematic model does not include. The increase in erosion rate could also be associated with a direct, externally-driven increase in surface erosion rate associated with Quaternary climate change. Although the Apennines were not significantly affected by alpine glaciation, the cooling and strong cyclicity of the Quaternary climate may have led to an increase in erosion rate through the efficiency of periglacial processes and hillslope processes such as landsliding (Amorosi et al., 1996; Borgatti and Soldati, 2010; Simoni et al., 2013; Wegmann and Pazzaglia, 2009).

**5 Conclusion**

We present evidence from multiple thermochronometers that the spatial and temporal pattern of erosion rates in the Northern Apennines orogen differs at the regional scale. Time-averaged erosion rates from individual thermochronometers predict faster erosion rates derived from AHe ages on the Adriatic side relative to the Ligurian side. These results are consistent with erosion rates derived from paired AFT-AHe thermochronometer samples, which illustrate an increase in erosion rates through time on
the Adriatic side, but a decrease in erosion rates through time on the Ligurian side. The pattern of erosion rates across the orogen is also consistent with a kinematic model for an asymmetric orogen that includes both frontal accretion and underplating modes of crustal accretion, a slab rollback rate of 10 km/My, and prowedge erosion rates that are a factor of two higher than retrowedge erosion rates. This model suggests that that observed decelerations on the retrowedge are the result of the spatial advection of rock to the SW, although the observed acceleration of erosion rates on the prowedge requires external forcing,
either through an increase in accretionary flux or through more erosive conditions linked to climate change.

*Code/Data Availability*





All data used in this study are included in the text. Related codes for the erosion rate analysis and kinematic model are available upon request to the corresponding author.

*Author Contribution*

EDE, MGF and SDW collected the detrital AFT samples in Northern Apennine catchments. EDE and MGF processed and analyzed the detrital AFT samples. SDW wrote the kinematic model. EDE, MGF, and SDW all contributed to the interpretation of the data. EDE wrote the manuscript and created the figures, with input from MGF and SDW.

*Competing Interests*

The authors declare that they have no conflict of interest.

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
