# Peer review of "Exhumation and erosion of the Northern Apennines, Italy: new insights from low-temperature thermochronometers"

_Solid Earth, 2021_

## Author Response (AR1)

**Reviewer 1:**

**The manuscript aims to determine the orogeny-scale (northern Apennine) erosion pattern derived from multiple thermochronometers, and to reconstruct erosion rate variation with space and time. The authors process a large data set of already published apatite fission track and apatite (U/Th)/He data that are accompanied by new detrital AFT data from 7 catchments (modern river sand). Erosion rates have been calculated for each samples using AGE2EDOT code and by applying different values of geothermal gradients.**

**The method to manage the geothermal data is particularly interesting and provide a very rigorous approach, mainly in light to the relevance of geothermal gradient to calculate erosion rate from thermochronological ages.**

**A numerical kinetic model of an asymmetric orogenic wedge evolution is set to explain the observed erosion rates, thermochronological age and maximum burial data.**

**The paper is well written and in general clear to read. The new data are of high quality. The obtained erosion rates data set is particularly interesting and they worth alone to be published. The application of a kinematic model and an interesting discussion made this paper perfectly suitable to be published in Solid Earth, with a only minor corrections.**

The reviewer provides an accurate summary of our study and we appreciate the positive feedback.

**My main criticism is focused on the mechanism invoked to explain decreasing in erosion rate along the Ligurian side (the retrowedge of Apennine orogenic wedge). The change in trajectory in the retrowedge seems the first order raison to explain a decrease in erosion rate with time. I feel that the depth of this variation can have a strong impact in the change in erosion rate with time. This variation in trajectory should occur between AFT closure depth and the AHe closure depth. Therefore the closure depth for AFT and AHe systems should deeply control the erosion rate pattern in the retrowedge. In the text it not very clear how closure depth is calculated line 237). Moreover, I am wondering to see the impact of different closure depths in modeling results.**

This is an important point, and we agree that the methodology for calculating the closure depths should be included. We address this question in the "Kinematic Model" Section of the reviewer's comments.

**Regional pattern of several data set (i.e. Ro, fg. 2, thermochronological ages in inset map of fig. 9) shows a clear variation along strike. In the manuscript this along strike variation is never discussed, although has been interpreted in literature as a first-order tectonic control on erosion and exhumation. I would like to know the raison and conditions to apply the same kinematic model of the entire of Apennine wedge.**

We briefly explain the pattern of vitrinite reflectance across the orogen and have added additional text to explain the pattern along strike of the orogen.

*"Ro values also decrease along strike of the orogen from NW to SE (Fig. 2), illustrating that maximum burial depths also decrease towards the SE. This pattern was in turn interpreted to reflect the shape of the Ligurian Unit as a wedge that thinned towards the east (Zattin et al., 2002), and thus resulted in shallower burial depths for the underlying Cenozoic Foredeep deposits. "*

We agree that the pattern of vitrinite and cooling ages reflects a first-order tectonic control related to rollback of the Adriatic slab and retreat of the hinge (Thomson et al., 2010) The timing and rate of rollback and hinge rate vary across the orogen, but the fundamental mechanism is the same. In our model, it is not possible to input a spatially variable slab rollback rate, although we do allow for a range of slab rollback rates that is consistent with estimates in our study area. Providing a single rate of rollback may be a model limitation, so we provide some additional text in the discussion to clarify these points.

*"The acceleration of exhumation may be related to a change in the timing or rate of slab rollback, which has varied along strike and across the orogen (Faccenna et al., 2014; Rosenbaum and Piana Agostinetti, 2015) and is a first-order tectonic control on exhumation and erosion (Thomson et al., 2010). We allow for a range of rollback rates that are consistent with rates for the field area, although the kinematic model is not able to resolve variability in rollback rates in either space or time."*

**Line 100 to 102. Variation of Ro is clear to follow also a NW-SE gradient.**

The reviewer brings up a good point. We do not discuss the along-strike variability in Ro from the Gottero to the Val D'Arno swaths, although it is clear that Ro values decrease from NW to SE, but that there is also less variability in Ro values on the Adriatic side from NW to SE. We have added extra text to bring up these points.

*"Ro values also decrease along strike of the orogen from NW to SE (Fig. 2), illustrating that maximum burial depths also decrease towards the SE. This pattern was in turn interpreted to reflect the shape of the Ligurian Unit as a wedge that thinned towards the east (Zattin et al., 2002), and thus resulted in shallower burial depths for the underlying Cenozoic Foredeep deposits."*

**In the erosion rate result section, I found some difficulties to read the text following figures 7 and 8. Figure 8 is described before figure 7. To be fair, I do not understand the meaning of figure 7 and what information the authors want to explain. It could be useful to add the geographic orientation, i.e NW to SE or NE to SW**

The purpose of Figure 7 is to illustrate the along-strike differences in ages (related to the reviewer's comment above) and erosion rates for the Adriatic (Figure 7b,d) and Ligurian (Figure 7a,c) sides. This figure is the only example that illustrates the patterns of cooling ages and erosion rates from this orientation, whereas all other figures illustrate data along transects perpendicular to the strike of the orogen. However, we think that this figure is perhaps difficult to read because we have combined all thermochronometers in each panel. To make the figure easier to read, we have created two panels per row, one with the AFT data and one with the AHe data.

We are also more explicit on the difference between Figures 7 and 8 in the text when we introduce the erosion rate results:

*"Here, we present the erosion rate results for the Adriatic and Ligurian sides, given the two different methods used for constraining the final geothermal gradient, and by illustrating the data with two perspectives: (1) along a profile oriented parallel to the orogen strike (Fig. 7, location shown in Fig. 3) and (2) along swath profiles oriented perpendicular to the orogen strike (Fig. 8)."*

**Kinematic model.**

**In this section I suggest to add some lines to describe the code and the environment of modeling. Line 237: It is not clear how the closure depth are chosen.**

To calculate the closure depths shown on line 237, we used closure temperatures from the literature: AHe = 70°C (Farley, 2000), AFT = 110°C , (Wagner and Van den Haute, 1992), and ZHe = 180°C (Farley, 2000). These temperatures were converted to a closure depth by assuming a geothermal gradient of 25°C/km. We think that this approach is likely too simplistic, given the constraints we have on final geothermal gradients derived from heat flow maps. Instead, we now use an average final geothermal gradient of 36.4 °C /km for all sample locations in our field area, calculated from the $G_{F\_heatflow}$ estimates. Using a higher geothermal gradient will produce shallower isotherms. Given the temperatures listed above for each thermochronometer, this produces closure depths of 1.9 km (AHe), 3.0 km (AFT), and 4.9 km (ZHe). We added the following text to explain our calculations and procedure:

 *"Closure depths were calculated using the closure temperature for each thermochronometer, divided by a spatially and temporally constant geothermal gradient. Closure temperatures are given as:  AHe = 70°C (Farley, 2000), AFT = 110°C , (Wagner and Van den Haute, 1992), and ZHe = 180°C (Farley, 2000). Excluding the Alpi Apuane samples, we used the full set of unique sample locations in our field area (Tables 1–4) to calculate an average $G_{f\_heatflow}$ = 36.4  °C/km and closure depths for the ZHe (4.9 km), AFT (3.0 km), and AHe (1.9 km) thermochronometers."*

**At line 372: the best –fit between what data? For large audience could be useful a short description how this model works. Moreover it is not very clear why the authors show this run.**

We agree that the term "best-fit" could be clearer for the reader. We add text to both the Methods section and Results section to better describe the objectives of the model and how we found the "best-fit" model results.

Lines 230-234
*"We used a range of kinematic and thermal parameters applicable to the Northern Apennines to characterize a kinematic model that aims to: (1) model the path of rock particles from accretion into the wedge to their erosion at the surface, (2) calculate uplift and horizontal rock velocities across the wedge, (3) predict reset cooling ages for AHe, AFT, and ZHe thermochronometers, and (4) calculate maximum*

*burial depths across the model. Here, we describe the model geometry and the kinematic and thermal parameters used to constrain the model."*

Lines 373-380
*"To construct the best-fit model, our goal is to reproduce the pattern of reset and non-reset thermochronometer ages and uplift rates from geodetic releveling (D'Anastasio et al., 2006) for the prowedge (0.5–1 km/My) and retrowedge (-0.15–0.12 km/My). To this end, we adjust the slab rollback rate within the acceptable range for our field area (6–11 km/Ma), and the AHe erosion rates within the range of values calculated from GF$_{heat\ flow}$ (0.17–1.9 km/My) (Table 8). Since ZHe samples are only reset near the Alpi Apuane, an acceptable model should have ZHe cooling ages that are >10 Ma across the orogenic wedge, whereas AFT and AHe samples should be reset across the wedge. Increasing the erosion rates within the kinematic model produces younger cooling ages. Increasing the rate of slab rollback increases the horizontal component of motion for rock particles and produces particle paths with shallower maximum burial depths."*

We are not entirely clear about which run the reviewer is referring to. We include the full outputs for both the SCR and VER scenarios. We hope that the above explanation has clarified the reasons for which we included each scenario in our results.

**Erosion rate pattern: 459, it could be interesting to specify what kind of tectonic control could be responsible for local high exhumation rate for the Apuane Alps, and to add a reference.**

We have added the following text to specify in more detail the tectonic control on the Alpi Apuane, and add the reference from Molli et al. (2018):

*"An exception to this general pattern may be the Alpi Apuane massif, which represents a structural culmination exposing a deep section and where high exhumation rates from the latest Miocene to the Present likely reflect post-orogenic processes of crustal thinning (Fellin et al., 2007; Molli et al., 2018)."*

**Fig 11. If the figure represent an enlarged portion of figure 10 (and not figure 9), so I do not understand the 100 km of horizontal scale.**

Thank you for pointing this out. This reference is an error and should in fact be Figure 10. We have fixed this mistake. In reference to the scale, we have kept the horizontal scale the same, but have enlarged the vertical scale only so that we can more clearly see the pattern of material motion in the wedge for the depths relevant to the low-temperature thermochronometers (ZHe, AFT, AHe) that we included in the study.

**Fig 9. To make the reading easier, it could be better to move the inset map within the panel 9b.**

We agree that the position of the inset makes the figure more difficult to read, so we have moved the inset out of the figure panels and place it above as panel (a). The other panels have been relabeled accordingly and adjusted in the text.

**Reviewer 2:**

This manuscript presents a compilation of published low-temperature thermochronology data from the Ligurian and Adriatic side of the Northern Apennines that are analyzed to infer the evolution of erosion rates on the pro side and retro side of the wedge. The authors also present new detrital AFT data from modern-sand samples from the Ligurian side. However, most of their conclusions derive from published bedrock data.

I think that the manuscript requires a major re-organization before further consideration and should be shortened considerably for the sake of clarity. My first impression is that what is presented here should be divided into two separate manuscripts: a first manuscript that analyzes the compiled bedrock ages and illustrates the main conclusions shown in this work, and a second manuscript that presents the new data on detrital samples from the Ligurian side. However, it is not clear to me what are the main implications revealed by this new detrital data set.

As far as this submission is concerned, improvements are required in the description of the tectonic setting and associated references, and in the description of the methodological approach. The authors should better describe the strategy they have adopted to check that their results are not compromised by cooling histories that are not monotonical. They should also better discuss the assumptions they have made concerning their modelling approach. For example, they have adopted the same initial geothermal gradient for samples located in the frontal and in the rear part of the orogen, which is difficult to understand given the different tectonic settings, contractional in the frontal part vs extensional in the rear part. Their model assumes that erosion initiated over the entire region at 10 Ma, despite compiled ages derive from samples belonging to tectonic units that were accreted at different stages of the orogeny. Also, the evolution of the Northern Apennines in the past 10 Ma includes a major strike-slip component that was not considered by the authors, and slab rollback rates were probably higher. I think all these points should be addressed before further consideration of the manuscript. Below are more specific comments that I hope will help improving this work.

One of the main concerns of Reviewer 2 is that our conclusions are derived from published bedrock data and that the implications of our new detrital data are unclear. Our work is based on a compilation of the existing cooling data, which include bedrock and detrital data, and we base our conclusions on both datasets. The detrital data are complementary to the bedrock data. In the Northern Apennines, bedrock data are unevenly distributed, even in areas with no limitations related to rock types. Because of the uneven and potentially biased spatial data distribution, the cooling record at the scale of the orogen could be only partially sampled by bedrock data. The detrital data are therefore useful to check against these potential biases. In fact, detrital data by a previous study on the Adriatic side of the Apennines revealed the presence of a much younger thermochronologic signal than the one recorded by the bedrock samples alone (Malusa' and Balestrieri, 2012). Thus, although detrital data can also be biased as a result of spatial variability in erosion rate combined with variability in apatite fertility, they ensure sampling at the scale of a drainage basin. In this context, they can also be useful to spot a missing record if compared with bedrock data. Finally, our new detrital data indicate that exhumation in some areas of the Northern Apennines could have started earlier than suggested thus far, as they show a large population centered at 13 Ma. This is discussed in section 4.1. In the light of the reviewer's concerns,

we partly modified the introduction to clarify our approach and to highlight the importance of the combination of detrital and bedrock data.

With regard to the reviewer's suggestion to split the manuscript into two manuscripts, we think that splitting the manuscript would be detrimental to our work. We collected our new data purposefully to complement the existing ones, to ensure the largest possible spatial sampling of the thermochronologic record in the Northern Apennines, and to propose a new orogenic model that builds on existing models. Instead of splitting the manuscript, we revised our introduction in order to explain better our goals and approaches, and to justify the need to integrate data and orogenic models.

Another concern of reviewer 2 is that our manuscript is too long, but it is unclear which parts should be shortened, since much of our text is devoted to discussing our assumptions and to constraining the thermal boundary conditions for our erosion rate analysis. We nevertheless hope that the improvements we have made to the manuscript address the comments that reviewer 2 has provided.

Finally, reviewer 2 elaborates on most of the points and improvements to the manuscript in the specific comments below, where we address them in detail. To summarize, we have added references where requested in the figure captions, added text to the methodology clarifying our model assumptions, justify our treatment of the initial geothermal gradient, and address the concern regarding the strike-slip component and slab rollback rates.

**Specific comments**
**Lines 44-45: "Development of the Apenninic wedge began at ~30 Ma, due to convergence and southwest-directed subduction of the Adriatic microplate beneath Eurasia." This statement is incorrect if referred to the Apennines generally, as the development of the Apenninic wedge in the south started in the Eocene at the latest (see Lustrino et al. 2009 – Tectonics).**

Since we refer to the entire Apenninic wedge here, we amend our statement to say that development of the wedge began during the Eocene and cite the Lustrino et al. 2009 study.

**Lines 45-46: "From the late Oligocene, sediments supplied largely by the Alps were deposited as turbidite sequences into a series of northward-migrating foredeep basins (Macigno, Cervarola, and Marnoso-Arenacea Basins)".**

**Here the authors should be more precise: sediments were supplied largely by the Central Alps (see, e.g., Garzanti and Malusa 2008 – EPSL; Malusa et al 2015 – Geology)**

We have clarified in Lines (47-49) that the Central Alps provided the sediment and use the suggested references.

**Line 49: "Tertiary foredeep deposits". Change to Cenozoic, Tertiary is an obsolete term**

We have changed all the references for "Tertiary" in the text to "Cenozoic".

**Figure 1: There is a typo in the keys on the top-right. In the caption, it should be made clear that thermochronological ages are from the literature, and the original papers should be also quoted.**

We found two typos in the spelling of "turbidite" and have fixed these in the legend. We have also added the sources for the cooling ages shown in Figure 1 to the caption and clarify that the ages shown in Figure 1 are only those that are <10 Ma.

**Figure 2: Vitrinite reflectance and cooling age data are from the literature. References should be explicitly indicated in the caption.**

We have added references for the vitrinite data and cooling ages in the figure caption.

**Lines 77-80: "The first evidence for emergent topography in the Northern Apennines is documented in the Early Pliocene, both by lacustrine deposits in an intermontane extensional basin located within the Magra River catchment (Fig. 3) (Bertoldi, 1988; Balestrieri et al., 2003), and by the exhumation of the Alpi Apuane metamorphic dome (white, hatched area in Fig. 3a) to the surface (Fellin et al., 2007)."**

**The second part of this sentence is conceptually wrong, because topography and exhumation are not directly related.**

We have changed the text to say:

*"The first evidence for emergent topography in the Northern Apennines is documented in the early Pliocene by lacustrine deposits in an intermontane extensional basin located within the Magra River catchment (Fig. 3) (Bertoldi, 1988; Balestrieri et al., 2003). These deposits are overlain by late Pliocene alluvial conglomerates that contain metamorphic pebbles sourced from the Alpi Apuane metamorphic dome (white, hatched area in Fig. 3a), indicating that the Alpi Apuane was emergent at this time (Fellin et al., 2007)."*

**Line 82: "recorded by Pleistocene surface uplift of rocks at the drainage divide".**
**Surface uplift or rock uplift? This sentence should be amended.**

We agree that this sentence is unclear. We have changed it to say: *"The onset of topographic relief then migrated eastward (Abbate et al., 1999; Thomson et al., 2010; Carlini et al., 2013), recorded by increased rock uplift rates at the drainage divide during the Pleistocene (Balestrieri et al., 2003)…"*

**Figure 3: Reference for published detrital AFT data should be explicitly indicated in the caption. Measurement units (Ma?) are missing in the diagrams. What is the meaning of n.?**

We have included references for the published detrital AFT data in the caption. Due to space limitations within the figure panels, we state in the caption that all ages are given in Ma and that "n." refers to the number of dated grains.

**Lines 110-112: "Bulk samples were sieved, and heavy minerals were separated using standard techniques, involving the use of the Wilfley table, heavy liquids, and the Frantz magnetic separator."**

**References are needed here, for example: Kohn, B., Chung, L., & Gleadow, A. (2019). Fission-track analysis: field collection, sample preparation and data acquisition. In Fissiontrack thermochronology and its application to geology (pp. 25-48). Springer, Cham.**

We respectfully disagree with the reviewer that a reference is needed, since these are standard techniques that have been established for more than 50 years.

**Lines 123-124: "We determined age populations for detrital samples based on dominant age peaks identified with the Binomfit program (Brandon, 2002), which is well suited for AFT data with low spontaneous track density." Which strategy was employed not to miss zero-track grains? Please, provide details on this point**

For each sample, we counted all countable grains, including zero track grains. Countable grains specifically refer to any grains that expose a section parallel to the c axis, independently of whether it has zero or more spontaneous tracks. We added text to this effect on lines 128-129.

**Lines 124-126: "In order to estimate the degree of resetting of the detrital age populations relative to the Apenninic orogenic event, we compared the detrital cooling ages with minimum depositional ages of the Tertiary foredeep units exposed in the drainage areas (Fig. 1)."**

**The range of stratigraphic ages in the drainage should be indicated also in the plots of Fig. 3b (e.g., by horizontal bars)**

We have increased the size of the panels in Figure 3b for readability, and have added the range of stratigraphic ages for the rocks and sediments that each basin drains. The colors in the legend correspond to the colors shown on the geologic map in Figure 1.

**Lines 128-130: "We compiled ages from new and existing detrital AFT samples (23), bedrock AFT samples (139), AHe samples (135), and ZHe samples (26) (Tables 1–4) (Abbate et al., 1994; Balestrieri et al., 1996; Ventura et al., 2001; Zattin et al., 2002; Balestrieri et al., 2003; Fellin et al., 2007; Thomson et al., 2010; Malusà and Balestrieri, 2012; Carlini et al., 2013)."**

**Only at this stage do the authors present their methodological approach, despite the results based on such a compilation are already presented in a previous section of the manuscript. Tables 1 to 4 should be placed in the Supplementary Material. Which kind of ages is indicated in those tables?**

At the beginning of the Section 2.2: Erosion rate analysis section (now lines 144-149), we list the number of samples that we use in the study from each thermochronometer, for the purposes of the AGE2EDOT erosion rate analysis. We believe this is the most appropriate place for the information, since this is part of our methods. We also provide a short review of the main interpretations from published thermochronologic studies in Section 1.2 as part of our introduction to the field area. The title of this section "1.2 Thermochronology data compilation" is perhaps misleading, since it is not part of the Methods, so we have removed the title for section 1.2 and include these lines within section 1.1, under the new title "Geologic and thermo-tectonic evolution".

We have moved Tables 1-4 to a new Supplement. We are not clear on reviewer 2's question "What kind of ages are indicated in those tables?" Each table is organized by the thermochronometer used to calculate the sample cooling age (e.g. bedrock AFT or detrital AFT). In each table, we provide the cooling age and errors, in addition to sample information and the original manuscript reference for the sample.

**Lines 149-154: "We converted ages to erosion rates using a half-space cooling model and a closure temperature concept (Willett and Brandon, 2013). This model has the advantage of including an**

accurate representation of the transience associated with whole lithosphere geotherms. Reset ages were converted to erosion rates using the closure temperature concept (Dodson, 1979), with closure temperatures specific to each thermochronometer, although this is a simplification of diffusional daughter product loss that neglects effects associated with complex cooling histories. For monotonic cooling histories, the measured age of the sample is represented by the time needed for a rock to move from the closure depth to the surface (e.g. Reiners and Brandon, 2006)."

**The authors should me more specific about this point and describe the strategy they have adopted to check that their results are not compromised by cooling histories that are potentially not monotonical.**

The literature on the Northern Apennines provides evidence for cooling histories (age-elevation profiles in several locations) that are not monotonical (e.g., Thomson et al. 2010; Balestrieri et al., 1996). Besides age-elevation profiles, the other way to check for non-monotonic cooling ages is to do a paired-ages analysis with AGE2EDOT. For AFT samples, track lengths provide constraints on non-monotonic cooling paths, but these can be modeled on single samples and to derive cooling histories only. This precludes the inversion of a large dataset and also does not allow taking into account the effect of heat advection to derive exhumation rates along a cooling path.

In this manuscript, we adopted two methods for calculating erosion rates through time. In the first method, we calculate time-integrated erosion rates for individual samples, where we cannot check for non-monotonic cooling ages. In the second method, we adopt the paired ages approach detailed in our manuscript (Lines 200-225). Regardless of the method that we use, we find the same overall results, namely that erosion rates have decreased through time on the Ligurian side and increased through time on the Adriatic side.

**Line 159-160: "In addition, the thermal initial and boundary conditions, as well as thermal parameters, must be specified for each sample site" How did the authors evaluate the initial thermal conditions? Please, better explain to the reader also underlining potential pitfalls.**

Constraining the thermal initial conditions is an important part of this manuscript and for the erosion rates analysis, and we discuss this topic at length. In Lines 176-182, we introduce our approach for constraining the geothermal gradient. We evaluated thermal conditions in two ways. In the first method, we assume initial conditions, namely an initial geothermal gradient of 25 C/km. This value should represent the geothermal gradient before the onset of erosion, so we chose a low geothermal gradient in the range of typical values for a foreland basin. In the second method, we impose only a final geothermal gradient that we derive from measured heat fluxes (della Vedova et al. 2001; Pauselli et al., 2019). This methodology is detailed in lines 180-190, and we dedicate an entire section (Section 4.2) to discussing possible uncertainties in these thermal conditions and how they may have affected the calculated erosion rates.

**Table 5: "initial geothermal gradient: 25°C/km"**

**The authors adopted the same initial geothermal gradient for samples located in the frontal and in the rear part of the orogen. This choice is particularly difficult to understand, given the remarkably different tectonic settings (contractional in the frontal part vs extensional in the rear part).**

The initial geothermal gradient of 25 C/km is only one of the methods that we use to constrain the geothermal gradient. In the AGE2EDOT inversion, the initial geothermal should represent the geothermal gradient before the onset of erosion, which in our case was fixed at 10 Ma. Thus, the initial geothermal gradient does not necessarily relate to the current position of the sample in the frontal or rear part of the wedge, but it should reflect the possible geothermal gradient within the wedge before and until the onset of erosion and heat advection. As we discuss in the text, there is little evidence of erosion, therefore heat advection, in the Northern Apennines before 10 Ma ago, and this justifies the assumption of a relatively low initial geothermal gradient. Because we are aware that this assumption is intrinsically simplistic, we also used the alternative approach of imposing a final geothermal gradient that we derive from the observed heat flux. We then compare how these different approaches affect our results, which we discussed at length in Section 3.2.

**Lines 190-191: "The modelling procedure described above was applied to all ages, assuming that erosion initiated over the entire region at 10 Ma." This is another assumption that is difficult to understand, because samples belong to tectonic units that were accreted during different and well-constrained time intervals.**

Given that there is no erosion while a sedimentary rock is buried in the foreland, heat advection only starts as the accretionary wedge emerges above sea level. Thus, the time of accretion of foreland sediments constrains only the maximum age for the onset of erosion. As we introduce in section 1.1 and discuss in section 4.2, there is little evidence of erosion before 10 Ma in the Northern Apennines. Thus, for all the samples from Cenozoic foredeep units with depositional ages much older than 10 Ma, an onset time for erosion at 10 Ma is likely a reasonable assumption. For the Cenozoic foredeep units that were deposited in the middle-late Miocene, this assumption may not hold. However, among the dated rocks compiled here, the youngest ones are the foredeep sediments of the Marnoso Arenacea. The depositional age of this unit can extend until the late Miocene. However, except for possibly a couple of samples, most ages from the Marnoso Arenacea come from portions of this unit that likely have depositional ages significantly older than or close to 10 Ma. Given that accretion in the Northern Apennines occurs very shortly after deposition (e.g., Argnani and Ricci Lucchi, 2001), it is likely that accretion and erosion started close to 10 Ma, even for the youngest dated Marnoso Arenacea rocks. Assuming a later onset of erosion, for instance 8 Ma, would result in a slightly higher erosion rate. Finally, we do present the case where the initiation of erosion may have been earlier (Lines 440-445), but note that changing the initiation age would proportionally decrease all erosion rates in the Northern Apennines in response to a longer period of erosion.

**Lines 191-197: "The resulting erosion rate applies from the onset of exhumation at 10 Ma to the present and reflects the time-averaged erosion rate constrained to pass through the closure temperature at the age and with a cooling rate commensurate with the average erosion rate. Thus, this method is limited to a single, average erosion rate. However, changes in exhumation rates through time in the Northern Apennines are supported by several lines of evidence, particularly by age-elevation transects (AETs). In fact, AETs from the existing literature illustrate differences along the age-elevation slope for a single thermochronometer (as in Balestrieri et al., 1999) or among age-elevation slopes for multiple thermochronometers (as in Thomson et al., 2010)."**

**As previously stated, how the authors can be confident that their results are not compromised, given that there is evidence that cooling histories are not monotonical?**

Reviewer2 mentions this point in a previous comment, which we address above.

**Lines 230-231: "The kinematic model presented here approximates the Northern Apennines as a doubly tapering, asymmetric wedge, given the geometric parameters illustrated in Fig. 5."**

**Note that the evolution of the Northern Apennines in the past 10 Ma includes a major strike-slip component not considered by the authors. This point should be discussed in detail also evaluating the impact on the model results.**

Reviewer 2 mentions a major strike-slip component here and in the summary as an important point that we have not considered. Reviewer 2 has not provided any reference studies that discuss this strike-slip component, and it's also not clear to us which faults they are referring to in the Northern Apennines. Without more details on this point, we are not able to address whether such a fault or such movement would affect our model results.

**Line 248: "Slab rollback rates are on the order of 6–10 km/My in this region of the Apennines"**

**What is the impact if slab rollback rates are much higher (ca 20 km/Ma), as shown for example by Malusa, Faccenna et al 2015 ?**

As suggested by reviewer 2, we reran our model with slab rollback rates of 20 km/Ma. Increasing the slab rollback rate to 20 km/Ma significantly alters the material paths, so that no particles accreted below ~2.5 km reach the surface, and particles accreted at depths of < ~1.5 km are not reset for any thermochronometer. Thus, only a very narrow range of rock particles are reset for AHe and AFT, and no rock particles are reset for ZHe. The reset AHe ages range from 3.7 Ma near the prowedge drainage divide to 4.5 Ma near the Tyrrhenian coast. AFT ages are only reset for the Ligurian side and range from 6.2 to 7.1 Ma. While these ages are reasonable, the high rate of slab rollback does not produce a realistic pattern of cooling ages across the orogen.

[Figure]

The Malusà et al. (2015) "Contrasting styles of (U)HP rock exhumation along the Cenozoic Adria-Europe plate boundary (Western Alps, Calabria, Corsica)" study mentioned by reviewer 2 includes a transect through Corsica and the Northern Apennines, where slab rollback and hinge retreat rates are estimated (see Figure 10). We could find no quantitative estimates from the paper beyond Figure 10, where it is difficult to confirm whether the rate of rollback suggested by reviewer 2 is consistent with this study. Rosenbaum and Piana Agostinetti (2015) constrain slab rollback to be a maximum of 10.6 km/Ma in the Sillaro area at the southern extent of our field area, based on the age of sedimentary basins on the Ligurian side of the orogen, and rates derived from tomographic data are similar (Faccenna et al. 2014). While we agree that our estimates of slab rollback could extend to ~11 km/Ma, to account for the southern portion of the study area, 20 km/Ma is higher than the estimates from this study and from the independent methods for constraining the slab rollback rate given in the literature.

**Tables 8 to 11 should be placed in the Supplementary Material.**

We have placed these tables in a new Supplement.

**Lines 414-415: "Fertility analysis of sediment from sampled Adriatic catchments also confirm that the detrital samples are representative of the eroded bedrock (Malusà et al., 2016)" However, no fertility analysis was performed by the authors on their samples draining the Ligurian side, which implies that this argument cannot be used to support their conclusion.**

While we have admittedly not performed a fertility analysis in this study, the catchments we sampled on the Ligurian side drain the same lithologies as the Adriatic catchments included in the Malusà et al., (2016) study. This suggests that the detrital samples on the Ligurian side should also be representative of eroded bedrock. We added a sentence (Lines 426-429) to convey this point:

*"Since the Ligurian catchments sampled in this study generally expose the same lithologies as the Adriatic catchments studied by Malusà et al. (2016), this suggests that detrital samples on the Ligurian side are also representative of eroded bedrock."*

---

## Author Response (AR2)

*The paper is much improved but there are still points that I think should be addressed:*

*-Different labs use different procedures, and thus introduce different sources of potential bias. The authors should either refer to a previous study where their specific procedure is described in full or provide more details about their separation procedure in the main text.*

As requested, we have added details to the Methods section (Line 125-131), which address our separation procedure for the apatite grains.

Lines 125-131 *"Samples were processed according to the external detector method for AFT dating, using standard methods. Bulk samples were first sieved with a 1.5 mm mesh, and heavy minerals were concentrated using standard techniques, involving the use of the Wilfley table and of heavy liquids. We separated the apatites from lighter minerals using a heavy liquid with a density of 3 g/cm³, and subsequently separated apatites from heavier minerals (e.g. zircon, rutile, and monazite) using a heavy liquid with a density of 3.3 g/cm³. A magnetic separator was used to further concentrate apatites. Apatites were then poured onto glass slides, carefully avoiding any potential selection of grains due to differences in size and shape. The grains were subsequently embedded in cold epoxy and polished to expose the internal surfaces of the apatite grains."*

*-I disagree that there is little evidence of erosion in the Northern Apennines before 10 Ma ago. For example, major unconformities are recorded within the wedge-top Apenninic successions in the Burdigalian (c. 18 Ma). This should be considered in the model set up.*

Within the wedge-top Apenninic successions, a major erosional unconformity at the top of an Aquitanian-lower Burdigalian formation has been attributed to an important thrusting phase (Papani et al., 1987). However, this is followed by shelf deposits in the wedge-top basin, rather than by the deposition of thick siliciclastic wedges of Apenninic provenance (Papani et al. 1987; Mancini et al. 2006). This pattern of deposition reflects a shallowing trend of the Epiligurian basin and rules out the emergence of the entire Apenninic accretionary wedge. Papani et al. (1987) suggest this unconformity and later ones can also be partly explained by sea level oscillations. We believe that these unconformities do not represent an onset of regional exhumation, which is the time period in Apenninic history that we are modeling. Other lines of evidence about the emergence above sea level of the Apenninic wedge come from many petrographic studies (e.g., Valloni et al., 2002) on the foredeep sandstones, indicating increasing Apenninic supplies through the middle Miocene. However, these studies find a clear shift to dominant Apenninic provenance only at the end of the Miocene, and the first important clastic wedge with Apenninnic provenance in Serravalle at ~12 Ma.  In our manuscript, we have noted that some ages older than 12 Ma exist, but that it would proportionally decrease all erosion rates in our analysis. We add text to the manuscript Section 4.2 to this effect and to address the Reviewer's point.

Lines 458-466. *"An erosional unconformity (ca. 18 Ma) is recorded in the Epiligurian deposits, which has been attributed to a phase of major thrusting and sea level oscillations (Papani et al., 1987). This unconformity is succeeded by shelf deposits that record an overall shallowing trend in the Epiligurian basins and is thus not related to the emergence of the entire Apennines wedge, which is the time period in Apenninic history that we are modelling. In fact, stratigraphic and petrographic data indicate that the Northern Apennines increasingly provided detrital sediment to the foredeep through the middle-late Miocene, while the wedge was still submerged (Valloni et al., 2002). The first important clastic supply from the Northern Apennines date to the Serravallian at ~12 Ma (Caprara et al., 1985). Thus, in our erosion rate analysis we used an onset time of 10 Ma as a reflection of the onset of heat advection due to erosion."*

Our kinematic model does not include a parameter for the onset of erosion age. Since we incorporate erosion rates consistent with our analysis of AHe ages, this model should reflect the more recent <5 Ma evolution of the Northern Apennines, when the Northern Apennines experienced major uplift and erosional exhumation (i.e. early Pliocene). We add a statement to this effect in the discussion.

Lines 569-572. *"Since we constrain the range of possible prowedge and retrowedge erosional velocities from AHe erosion rates, the kinematic model can be assumed to reflect the more recent <5 Ma evolution of the Northern Apennines, when the Northern Apennines experienced major uplift and erosional exhumation (i.e. early Pliocene)."*

***-The amount of strike slip through time is explicitly plotted in Malusà, Faccenna et al 2015, their Fig. 10b, and should be considered in the model.***

In Malusa et al. (2015), the authors suggest that strike-slip motion in the Northern Apennines has exceeded 300 km over the last 35 Ma and that strike-slip motion is on the order of ~ 50 km in the last 10 Ma (Figure 10b). This strike-slip motion is attributed to lateral translation of the Adria Slab beneath the Alpine Wedge that began in the Eocene.

Our kinematic model is 1D, and can be considered as a cross-section through the Apennines. The kinematics are only considered within the Apenninic wedge itself. The Adriatic slab is incorporated into the model only in the sense that it contributes underplating material to the wedge and for the slab rollback velocity. Thus, this model cannot resolve alternate orientations of material into the wedge, or how the alternate orientations might change the velocity of material entering the wedge. We have added text to the Methods (Lines 264-265) and to the Discussion  to further emphasize that the model is 1-D, and that it only considers a flux of material and slab rollback velocity in the direction of the wedge (i.e. perpendicular to the strike of the orogen).

Lines 285-286. *"Since the model represents a 1D cross-section through the Northern Apennines, all velocities and rates specified in this model are assumed to reflect motion within the direction of the wedge (i.e. perpendicular to the strike of the orogen)."*

Lines 617-620. *"We allow for a range of rollback rates that are consistent with rates for the field area, although the kinematic model is not able to resolve variability in rollback rates in either space or time, nor can it resolve how rotation or motion external to the wedge (e.g. lateral translation of the Adriatic slab relative to the Apennines wedge) would alter the flux or orientation of material entering the wedge."*

**-If the high rate of slab rollback does not produce a realistic pattern of cooling ages across the orogen, this does not necessarily imply that the model is right, and the slab rollback rates are wrong. The authors should address this point in a more open way.**

In Methods (Line 260) and Discussion (see below), we have added text to acknowledge that other studies have found higher rates of slab rollback for the Northern Apennines during the last 10 Ma. We note that a much higher rate of slab rollback would significantly affect the maximum burial depths for the particle paths, such that the model would produce shallower maximum burial depths (<2 km at the drainage divide and <3 km at the retrowedge model boundary) than values constrained from vitrinite reflectance. Thus, a lower rate of slab rollback on the order of 9-10 km/My will produce both the pattern of cooling ages and maximum burial depths in the Northern Apennines.

Lines 580-583. *"Slab rollback rates vary over an order of magnitude in the Northern Apennines (6–20 km/My). Higher values of slab rollback can reproduce the pattern of cooling ages across the Northern Apennines, although predicted maximum burial depths decrease with higher slab rollback rates."*